# SPEAR: A Unified SSL Framework for Learning Speech and Audio Representations

## Abstract

Self-Supervised Learning (SSL) excels at learning generic representations of acoustic signals, yet prevailing methods remain domain-specific, tailored to either speech or general audio, hindering the development of a unified representation model with a comprehensive capability over both domains. To address this, we present SPEAR (SPEech and Audio Representations), the first SSL framework to successfully learn unified speech and audio representations from a mixture of speech and audio data. SPEAR proposes a unified pre-training objective based on masked prediction of fine-grained discrete tokens for both speech and general audio. These tokens are derived from continuous speech and audio representations using a Multi-codebook Vector Quantisation (MVQ) method, retaining rich acoustic detail essential for modelling both speech and complex audio events. SPEAR is applied to pre-train both single-domain and unified speech-and-audio SSL models. Our speech-domain model establishes a new state-of-the-art on the SUPERB benchmark, a speech processing benchmark for SSL models, matching or surpassing the highly competitive WavLM Large on 12 out of 15 tasks with the same pre-training corpora and a similar model size. Crucially, our unified model learns complementary features and demonstrates comprehensive capabilities across two major benchmarks, SUPERB and HEAR, for evaluating audio representations. By further scaling up the model size and pre-training data, we present a unified model with 600M parameters that excels in both domains, establishing it as one of the most powerful and versatile open-source SSL models for auditory understanding. The inference code and pre-trained models will be made publicly available.

## 1 Introduction

The drive towards foundation models trained on broad data with generic features that can be adapted to many tasks is one of the most significant trends in AI, having been demonstrably successful in natural language processing (Brown et al., 2020) and computer vision (Kirillov et al., 2023). In the auditory domain, speech conveys linguistic and paralinguistic information, while general audio provides environmental context and sound events crucial for situational awareness. Critically, they are rarely isolated in real-world scenarios (Bregman, 1994). Therefore, developing a unified encoder with comprehensive capabilities across both domains is important for modern AI systems with holistic auditory perception function (Comanici et al., 2025).

Self-supervised learning (SSL) has emerged as an effective paradigm for learning generic representations in the field of speech and audio processing (Baevski et al., 2020; 2022; Huang et al., 2022; Dinkel et al., 2024). By leveraging a large amount of unlabelled data during pre-training, SSL models can achieve very high performance on tasks with limited supervised data (Chen et al., 2020b; Baevski et al., 2020). In speech processing, one dominant approach is masked token prediction (Hsu et al., 2021; Chung et al., 2021; Chiu et al., 2022). This approach involves quantisation techniques, such as k-means clustering (Hsu et al., 2021) or random projection quantisation (Chiu et al., 2022), to generate coarse-grained discrete tokens from the raw speech signal or intermediate SSL representations. Although the quantisation process discards some acoustic details, such as prosody and paralinguistic information (Xin et al., 2024), these tokens still effectively capture the phonetic content of speech (Baevski et al., 2020; Hsu et al., 2021). This makes the pre-training strategy highly suitable for the speech domain and has enabled models to achieve excellent performance on a wide range of downstream speech processing tasks (Chen et al., 2022; Yang et al., 2021).

However, such coarse-grained tokens are inadequate for general audio. In contrast to speech, general audio exhibits more irregular and complex spectro-temporal patterns (Attias & Schreiner, 1996) that are difficult to capture with coarse-grained discrete units (e.g., k-means). While BEATs (Chen et al., 2023) adapts discrete token prediction for audio, it requires a complex iterative process to train a dedicated audio tokeniser. EncodecMAE (Pepino et al., 2025) instead uses the off-the-shelf Encodec (Défossez et al., 2023) tokens as pre-training targets and shows promising results in general audio representation learning. However, when applied to speech data, the model exhibits only limited speech-related capability. Other audio SSL models forgo token prediction in favour of other objectives, such as masked autoencoder (MAE) (Huang et al., 2022; Dinkel et al., 2024) or bootstrapping-based methods (Li et al., 2024a). This divergence in performance and objectives has prevented the emergence of a unified SSL approach for both the speech and general audio domains.

To address this lack of unification, we propose **SPEAR** (SPEech and Audio Representations), a unified SSL framework for both speech and general audio. We hypothesise that masked prediction of *fine-grained discrete tokens* could be a unified SSL pre-training task for both domains. Our approach applies multi-codebook vector quantisation (MVQ) (Guo et al., 2023) to the intermediate representations of existing speech and audio SSL models to obtain fine-grained discrete tokens, which then serve as the targets for a masked prediction objective. MVQ decomposes the representation space into multiple subspaces spanned by parallel codebooks. This multi-codebook design enables MVQ tokens to retain far more detail than coarse-grained discrete units, making SPEAR suitable for both speech and general audio. The pre-training objective can be viewed as performing multiple masked language modelling (Devlin et al., 2019) tasks simultaneously, one for each MVQ codebook. A key aspect of SPEAR is the joint pre-training on speech and audio, where the model learns to predict two sets of MVQ tokens derived from separate expert models from two domains. Together with a specially designed asymmetrical pre-training strategy, the dual-target objective enables SPEAR to learn a single, unified representation space bridging both domains. Finally, to enhance the model's versatility for tasks requiring different temporal granularities (Shi et al., 2024a), we integrate a multi-temporal resolution encoder (Yao et al., 2024), allowing the model to process the input signal at variable frame rates in intermediate layers.

Our contributions can be highlighted as follows:

- We propose SPEAR, the first unified SSL framework for both speech and general audio that successfully learns high-quality unified representations for both domains.
- We conduct extensive experiments and validate the effectiveness of SPEAR in both single-domain and unified settings. Notably, our speech-domain model achieves the same or better performance than the competitive WavLM Large (Chen et al., 2022) on 12 out of 15 SU-PERB (Yang et al., 2021; Tsai et al., 2022) tasks under a fair comparison. Our audio model approaches the best-performing SSL models on the HEAR (Turian et al., 2022) benchmark, and even outperforms them on environment-related tasks while using far less data.
- We demonstrate that SPEAR unifies joint speech and audio pre-training, resulting in a model with a comprehensive capability over both domains. Furthermore, we show that this synergy is enhanced when scaling up model parameters and pre-training data.

Inference code and pre-trained models will be made open-source to facilitate future research.

## 2 RELATED WORK

**SSL for Speech and Audio**    As introduced in Section 1, SSL has been widely adopted for learning generic representations in both the speech domain (Baevski et al., 2020; Hsu et al., 2021; Chen et al., 2022) and audio domain (Huang et al., 2022; Dinkel et al., 2024). Yet, existing SSL methods are typically domain-specific, and a unified framework for both domains remains absent. Bootstrapping approaches (Grill et al., 2020; Caron et al., 2021; Niizumi et al., 2021) have shown promise, achieving strong performance in speech (Baevski et al., 2022; 2023) and audio (Chen et al., 2024; Li et al., 2024a) independently, but have not been successfully applied to joint SSL on both domains. While Gong et al. (2022) explored using both speech and audio data for SSL, their main focus was to improve general audio capability. To the best of our knowledge, our proposed SPEAR framework is the first to establish a unified SSL pre-training framework for both speech and general audio.

**Masked-Token-Prediction-based SSL**    Masked-token prediction is a widely used pretext task in SSL (Devlin et al., 2019). In the field of speech and audio, prevailing methods rely on coarse-grained

acoustic tokens generated via k-means clustering (Hsu et al., 2021; Chung et al., 2021) or random-projection quantisation (Chiu et al., 2022; Chen et al., 2023). In the music domain, MERT (Li et al., 2024b) integrates fine-grained Encodec tokens (Défossez et al., 2023) as its pre-training targets. Similarly, EncodecMAE (Pepino et al., 2025) uses an MAE (He et al., 2022) structure to predict the Encodec tokens for learning audio representations. Our framework SPEAR likewise utilises fine-grained tokens as the pre-training target, but derives them via a non-hierarchical multi-codebook vector quantisation method, while also extending the domains to both speech and audio.

**Knowledge Distillation**   Knowledge distillation (KD) (Hinton et al., 2014) is frequently employed as a model compression technique (Jiao et al., 2020; Chang et al., 2022; Yang et al., 2022). SPEAR is related to multi-teacher KD, since its pre-training targets are obtained from two domain-specific teacher models. Multi-teacher KD has been explored to combine knowledge from multiple domains within a single model (Ranzinger et al., 2024). In the speech and audio domain, Yang et al. (2025) proposes to train a single model by performing KD on three supervised teachers specialising in speech, speaker, and audio event, respectively. USAD (Chang et al., 2025), a contemporaneous work closely related to SPEAR, distils knowledge from two separate SSL models for speech and audio modalities into a single model to learn unified speech audio representations. However, USAD primarily focuses on feature matching, using objectives such as the L1 loss or cosine similarity to align the student representation space with that of the teacher. Our method differs from them by coupling KD with a well-established masked-token prediction SSL objective for representation learning, rather than explicitly matching teacher representations.

## 3   SPEAR

In this section, we introduce the proposed SPEAR framework. We hypothesise that a masked prediction objective can serve as a unified SSL solution for both speech and general audio, provided the discrete tokens are sufficiently fine-grained to retain critical acoustic detail from both domains. This motivates our choice of a powerful quantisation method, which is described below.

### 3.1   MULTI-CODEBOOK VECTOR QUANTISATION

To generate fine-grained discrete targets for our masked prediction SSL objective, we employ multi-codebook vector quantisation (MVQ) (Guo et al., 2023), a trainable quantisation method originally proposed to compress high-dimensional feature vectors for storage optimisation. To the best of our knowledge, the application of MVQ in the context of SSL pre-training has never been explored. MVQ utilises $N$ parallel codebooks, each containing $K$ trainable code vectors. Given an input feature vector $\boldsymbol{x} \in \mathbb{R}^d$, MVQ encodes it into a tuple of $N$ discrete tokens, i.e. $\boldsymbol{z} = \mathrm{Encode}(\boldsymbol{x}; \mathcal{Q}) = (z_1, \ldots, z_N)$. Each token $z_n$ is an integer index in the range $[0, K-1]$ specifying which code vector to select from the $n$-th codebook. These selected vectors can then be used to approximate the original feature vector $\boldsymbol{x}$ via a direct-sum scheme (Barnes & Watkins, 1995).

Intuitively, this process partitions the feature space into $N$ distinct subspaces, each governed by a corresponding codebook. The multi-codebook structure produces significantly more fine-grained representations than coarse methods like k-means, as the number of representable states grows exponentially as $K^N$. Compared to other multi-codebook quantisation methods like RVQ (Défossez et al., 2023), the codebooks in MVQ are non-hierarchical, reducing inter-codebook correlation and making each codebook equally important. The MVQ quantiser is trained by minimising the reconstruction error, supplemented by a diversity loss to encourage uniform code usage within each codebook. For a complete description of MVQ encoding and training mechanisms, we refer readers to the original paper (Guo et al., 2023) and the extended summary in Appendix B.

### 3.2   MULTI-CODEBOOK FINE-GRAINED MASKED TOKEN PREDICTION

#### 3.2.1   SINGLE DOMAIN PRE-TRAINING

The pre-training objective is to train a single student encoder $\mathcal{S}$, to predict fine-grained discrete tokens extracted from a pre-trained SSL teacher model $\mathcal{T}$ in a masked-token prediction manner. Since the teacher used for generating the pre-training targets was trained without any labelled data, SPEAR is treated as an SSL approach. An illustration of the overall framework is shown in Figure 1.

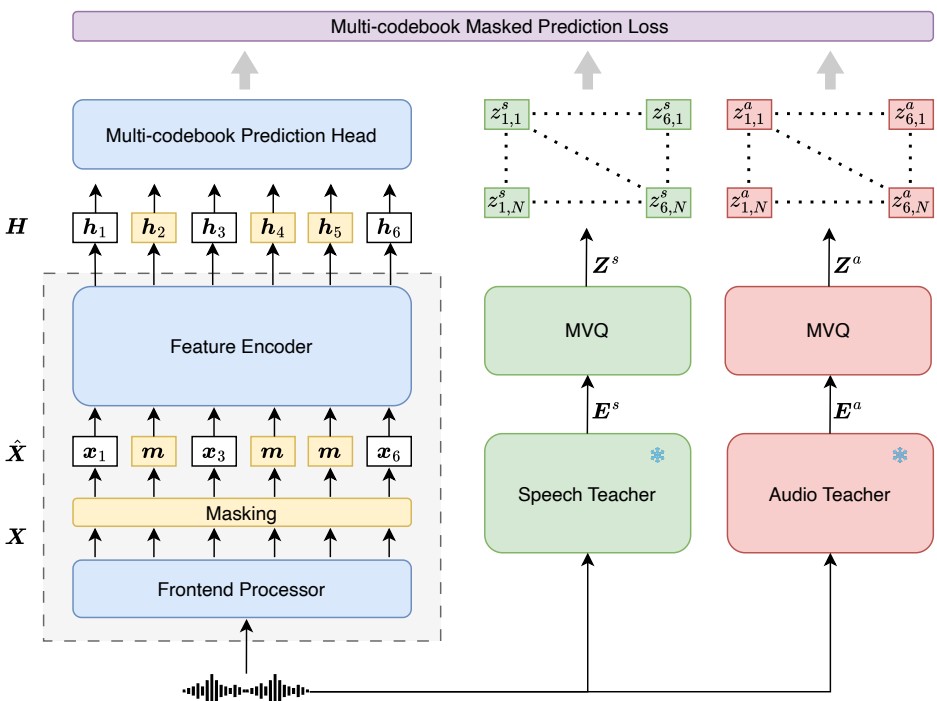

Figure 1: The SPEAR framework for dual-domain pre-training. Teacher models are frozen. For single-domain pre-training, only one teacher from the corresponding field is employed for generating pre-training targets. After pre-training, the components in the grey box are retained as the encoder.

The student encoder $\mathcal{S}$ consists of a frontend processor and a feature encoder $\mathcal{F}$ (specifically, a Zipformer (Yao et al., 2024)). The frontend processor converts the raw input waveform $\boldsymbol{w}$ into frame-level acoustical representations $\boldsymbol{X} = \{\boldsymbol{x}_1, \ldots, \boldsymbol{x}_T\}$ of length $T$. A masking operation is applied to $\boldsymbol{X}$ by randomly sampling a set of frames $\mathcal{M}$ and replacing $\{\boldsymbol{x}_t | t \in \mathcal{M}\}$ with a learnable mask embedding $\boldsymbol{m}$, creating the masked input $\hat{\boldsymbol{X}}$. The feature encoder $\mathcal{F}$ then processes $\hat{\boldsymbol{X}}$ to produce a sequence of contextualised representations $\boldsymbol{H} = \{\boldsymbol{h}_1, \ldots, \boldsymbol{h}_T\}$, where $\boldsymbol{h}_t \in \mathbb{R}^d$.

To generate the prediction targets, the same raw audio waveform $\boldsymbol{w}$ is fed into the teacher model $\mathcal{T}$, producing a sequence of frame-level representations $\boldsymbol{E} = \mathcal{T}(\boldsymbol{w}) = \{\boldsymbol{e}_1, \ldots, \boldsymbol{e}_T\}$. We assume the teacher and student models share the same frame rate[1]. These representations $\boldsymbol{E}$ are then quantised frame-by-frame using a pre-trained MVQ quantiser $\mathcal{Q}$ to produce a sequence of fine-grained discrete tokens $\boldsymbol{Z} = \{\boldsymbol{z}_1, \ldots, \boldsymbol{z}_T\}$ as the pre-training targets, where $\boldsymbol{z}_t = \mathrm{Encode}(\boldsymbol{e}_t; \mathcal{Q})$.

The student model is trained to predict the target tokens $\boldsymbol{Z}$ from the contextualised representations $\boldsymbol{H}$. The multi-codebook masked prediction loss is formulated as the sum of $N$ independent prediction losses, one for each codebook in the MVQ quantiser. Each of these losses is a cross-entropy objective calculated over all frames, with an adjustable weight $\alpha$ for masked and unmasked frames[2]:

$$\mathcal{L}_{\mathrm{single}}(\boldsymbol{H}, \boldsymbol{Z}) = \frac{1}{N} \sum_{n=1}^{N} [\alpha \mathcal{L}_m^n(\boldsymbol{H}, \boldsymbol{Z}) + (1-\alpha)\mathcal{L}_u^n(\boldsymbol{H}, \boldsymbol{Z})] \tag{1}$$

$$= \frac{1}{N} \sum_{n=1}^{N} \left[ \alpha \sum_{t \in \mathcal{M}} -\log p_n(z_{t,n} \mid \boldsymbol{h}_t) + (1-\alpha) \sum_{t \notin \mathcal{M}} -\log p_n(z_{t,n} \mid \boldsymbol{h}_t) \right], \tag{2}$$

where $\mathcal{L}_m^n$ and $\mathcal{L}_u^n$ are the loss on masked and unmasked frames for the $n$-th codebook, respectively. $p_n(z_{t,n}|\boldsymbol{h}_t)$ is the predicted probability of the correct token $z_{t,n}$ at time $t$ for the $n$-th codebook,

---

[1]This can be achieved by interpolating the teacher representations if the frame rates differ.
[2]The effect of $\alpha$ is investigated in Appendix G.2

which is computed via a softmax function over the logits from a projection matrix $\boldsymbol{W}_n$:

$$p_n(\cdot \mid \boldsymbol{h}_t) = \mathrm{softmax}(\boldsymbol{W}_n \boldsymbol{h}_t), \tag{3}$$

where $\boldsymbol{W}_n \in \mathbb{R}^{K \times d}$ is the projection matrix of the prediction head for the $n$-th codebook.

### 3.2.2 Unified Dual-Domain Pre-training

The framework is extended to dual-domain pre-training on a mixture of speech and general audio data for learning unified representations of both domains. Specifically, we employ two expert teacher models, $\mathcal{T}^s$ (speech) and $\mathcal{T}^a$ (general audio), along with their corresponding pre-trained MVQ quantisers, $\mathcal{Q}^s$ and $\mathcal{Q}^a$. Note that the number of codebooks could be different for $\mathcal{Q}^s$ and $\mathcal{Q}^a$.

For each input waveform, the teacher representations $\boldsymbol{E}^s$ and $\boldsymbol{E}^a$ are extracted. Two sets of fine-grained target tokens are obtained by applying the corresponding quantiser on $\boldsymbol{E}^s$ and $\boldsymbol{E}^a$:

$$\boldsymbol{Z}^s = \{\mathrm{Encode}(\boldsymbol{e}_1^s; \mathcal{Q}^s), \ldots, \mathrm{Encode}(\boldsymbol{e}_T^s; \mathcal{Q}^s)\} \tag{4}$$

$$\boldsymbol{Z}^a = \{\mathrm{Encode}(\boldsymbol{e}_1^a; \mathcal{Q}^a), \ldots, \mathrm{Encode}(\boldsymbol{e}_T^a; \mathcal{Q}^a)\}. \tag{5}$$

During dual-domain pre-training, the speech tokens $\boldsymbol{Z}^s$ are used as universal prediction targets for all input data, whereas the audio tokens $\boldsymbol{Z}^a$ are only used for loss computation when the input is general audio. This asymmetric approach helps the model achieve balanced performance across both domains (see Appendix G.7.1 for a comparison of three dual-domain pre-training strategies). The dual-domain training objective is formulated as follows:

$$\mathcal{L}_{\mathrm{dual}}(\boldsymbol{H}, \boldsymbol{Z}^s, \boldsymbol{Z}^a) = \mathcal{L}_{\mathrm{single}}(\boldsymbol{H}, \boldsymbol{Z}^s) + \mathbf{1}_{\mathrm{is\_audio}} \cdot \lambda \cdot \mathcal{L}_{\mathrm{single}}(\boldsymbol{H}, \boldsymbol{Z}^a), \tag{6}$$

where $\mathcal{L}_{\mathrm{single}}$ is the single-domain masked prediction loss defined in Equation 2. The term $\mathbf{1}_{\mathrm{is\_audio}}$ is an indicator function that returns 1 if the input is general audio and 0 otherwise. $\lambda$ is a hyperparameter for balancing the contribution of the general-audio-specific loss. By learning to predict both $\boldsymbol{Z}^s$ and $\boldsymbol{Z}_a$, the student model can learn a joint feature space for both domains.

## 4 Experimental Setup

**Data** Pre-training is performed on a mixture of public unlabelled English speech datasets and general audio datasets, as shown in Table 1. Due to the limited amount of public general audio datasets, we incorporate two music datasets, Music4all (Santana et al., 2020) and MTG-Jamendo (Bogdanov et al., 2019), to enrich the general audio data.

Table 1: Pre-training corpora used in SPEAR. Left: Speech datasets; Right: Audio datasets.

| Speech Dataset | Hours |
| --- | --- |
| Libriheavy (Kang et al., 2024) | $\sim 50$k |
| GigaSpeech (Chen et al., 2021) | $\sim 10$k |
| VoxPopuli (en) (Wang et al., 2021) | $\sim 24$k |
| Yodas-granary (en) (Koluguri et al., 2025) | $\sim 100$k |

| Audio Dataset | Hours |
| --- | --- |
| AudioSet (Gemmeke et al., 2017) | $\sim 5$k |
| VGGsound (Chen et al., 2020a) | $\sim 0.5$k |
| Freesound (Wu et al., 2023) | $\sim 2.8$k |
| Music4all (Santana et al., 2020) | $\sim 1$k |
| MTG-Jamendo (Bogdanov et al., 2019) | $\sim 3.8$k |

**Model Architecture** As shown in (Shi et al., 2024a), modelling speech representations at different time resolutions is beneficial to the comprehensive capabilities of speech SSL models. Therefore, Zipformer (Yao et al., 2024) is selected as the feature encoder in SPEAR due to its dynamic downsampling mechanism in the intermediate layers. The model receives 128-dimensional filter-bank features as input and produces frame-level representations at a 50 Hz frame rate.

**Pre-training Configuration** We pre-train three model scales under the SPEAR framework: Base (94M), Large (327M), and XLarge (600M). At the Base and Large scales, we pre-train both single-domain and dual-domain models. To further explore the benefits of scaling, we additionally train an XLarge dual-domain model on a larger dataset. The pre-training data used for different scales of SPEAR models are given in Table 2. The data mixtures are defined as follows: Speech-84k comprises Libriheavy, GigaSpeech, and VoxPopuli (en); Audio-13k includes all general audio datasets listed in Table 1; Mix-97k combines Speech-84k and Audio-13k; and Mix-197k additionally includes the Yodas-granary dataset. Detail regarding the encoder configurations is provided in Appendix C.1, and hyperparameters for pre-training are presented in Appendix C.2.

Table 2: Pre-training configurations for different SPEAR settings.

| Model | Domain(s) | Data Mixture | Total Hours |
|---|---|---|---|
| SPEAR$_s$-{Base, Large} | Speech | Speech-84k | ∼84k |
| SPEAR$_a$-{Base, Large} | Audio | Audio-13k | ∼13k |
| SPEAR$_{s+a}$-{Base, Large} | Speech & Audio | Mix-97k | ∼97k |
| SPEAR$_{s+a}$ XLarge | Speech & Audio | Mix-197k | ∼197k |

Table 3: Teacher models and MVQ-quantiser configurations for generating pre-training targets.

| Teacher Model | # Params | Pre-train Data | Model Config | | | MVQ Config | |
|---|---|---|---|---|---|---|---|
| | | | Domain | Model Dim | Frame Rate | $N$ | $K$ |
| WavLM Large | 317M | 94k | Speech | 1024 | 50 Hz | 16 | 256 |
| Dasheng 1.2B | 1.2B | 272k | Audio | 1536 | 25 Hz | 8 | 256 |

**Teacher Models and MVQ quantiser** WavLM Large (Chen et al., 2022) and Dasheng 1.2B (Dinkel et al., 2024) are utilised to generate pre-training targets for speech and audio domains, respectively. Model details and corresponding MVQ quantiser configurations are provided in Table 3. WavLM Large is pre-trained on 94k hours of unlabelled English speech data, including Libri-light (Kahn et al., 2020), GigaSpeech, and Voxpopuli (en). It can be fairly contrasted with SPEAR models trained on Speech-84k since Libriheavy is the segmented version of Libri-light. The speech MVQ quantiser with 16 codebooks is trained using the 21st layer representations of WavLM Large on 100 hours of data sampled from LibriSpeech (Panayotov et al., 2015). Dasheng 1.2B is an audio SSL model pre-trained with an MAE (Huang et al., 2022) objective on an exceptionally large amount of general audio data of over 272k hours. The audio MVQ quantiser is trained on the last-layer representations with 8 codebooks on 50 hours of AudioSet balanced set. Ablation studies on different choices of teacher models for pre-training target generation and MVQ quantiser configurations are presented in Appendix G.1.1 and Appendix G.3.

## 5 RESULTS

To validate the effectiveness of the SPEAR framework, we assess its performance through both full fine-tuning and frozen representation evaluations on two major benchmarks for evaluating speech and audio representations: SUPERB Yang et al. (2021); Tsai et al. (2022) and HEAR (Turian et al., 2022). Finally, ablation studies on core components of SPEAR can be found in Section 5.4.

### 5.1 DOWNSTREAM FINE-TUNING

We evaluate the performance of the pre-trained models on two key downstream fine-tuning tasks: **automatic speech recognition** (ASR) for speech capabilities and **audio tagging** (AT) for general audio understanding capabilities. The downstream fine-tuning results for the single-domain and dual-domain models are presented in Table 4 and configurations are shown in Appendix C.3.

**ASR** ASR performance is evaluated on LibriSpeech (Panayotov et al., 2015), where the model is fine-tuned on the train-clean-100 subset (LS-100) or the full 960 hours (LS-960) of LibriSpeech. A lightweight, stateless RNN-T decoder (Graves, 2012; Ghodsi et al., 2020) with fewer than 3M parameters using an output vocabulary of 500-class byte-pair-encoding (Sennrich et al., 2016) units is attached to the pre-trained models unless otherwise noted. During fine-tuning, the pre-trained SSL model is also updated. The projection heads for predicting MVQ tokens are discarded. Performance is measured by the Word Error Rate (WER) on the test-clean and test-other splits of LibriSpeech, using beam search decoding with no external language model. We mainly compare with the WavLM models pre-trained on similar corpora at similar model sizes. Additional results of ASR fine-tuning with a CTC (Graves et al., 2006) decoder can be found in Appendix D.

**Audio Tagging** To evaluate audio capabilities, our models are fine-tuned on AudioSet for AT following the procedure in Gong et al. (2021). We perform fine-tuning on both the balanced subset (AS-20k) and the full dataset (AS-2M). A linear projection layer is added on top of the encoder to

Table 4: Fine-tuning results on LibriSpeech ASR task and AudioSet AT task. For ASR, WERs under "clean" and "other" denote the WERs on test-clean and test-other sets. $^{\triangle}$: Model fine-tuned from the public checkpoint. Best results in **bold**, 2nd best results underlined.

| Model | # Params | Pre-train data | LS-100 | | LS-960 | | AS-20k | AS-2M |
|---|---|---|---|---|---|---|---|---|
| | | | clean | other | clean | other | | |
| ***Speech SSL Models*** | | | | | | | | |
| WavLM Base + $^{\triangle}$ (Chen et al., 2022) | 95M | 94k | 4.0 | 8.4 | 2.9 | 5.4 | - | - |
| HuBERT Large$^{\triangle}$ (Hsu et al., 2021) | 317M | 60k | - | - | 1.8 | 3.9 | - | - |
| WavLM Large$^{\triangle}$ (Chen et al., 2022) | 317M | 94k | 3.0 | 6.1 | 1.8 | 3.8 | - | - |
| Ours, SPEAR$_s$ Base | 94M | 84k | 3.0 | 5.8 | 1.9 | 4.0 | 26.9 | 43.6 |
| Ours, SPEAR$_s$ Large | 327M | 84k | 2.6 | 4.7 | 1.7 | 3.3 | 26.4 | 43.9 |
| ***Audio SSL Models*** | | | | | | | | |
| BEATs (Chen et al., 2023) | 90M | 5k | - | - | - | - | 38.9 | 48.6 |
| EAT (Chen et al., 2024) | 88M | 5k | - | - | - | - | **40.2** | 48.6 |
| ATST Frame (Li et al., 2024a) | 86M | 5k | - | - | - | - | 39.0 | 48.0 |
| Dasheng-Base$^{\triangle}$ (Dinkel et al., 2024) | 86M | 272k | - | - | - | - | - | 49.7 |
| Dasheng-1.2B$^{\triangle}$ (Dinkel et al., 2024) | 1.2B | 272k | 7.7 | 20.0 | 3.4 | 8.7 | - | **50.0** |
| Ours, SPEAR$_a$ Base | 94M | 13k | 11.2 | 23.0 | - | - | 39.2 | 49.3 |
| Ours, SPEAR$_a$ Large | 327M | 13k | 7.4 | 18.6 | - | - | 39.3 | 49.8 |
| ***Speech & Audio SSL Models*** | | | | | | | | |
| Ours, SPEAR$_{s+a}$ Base | 94M | 97k | 3.1 | 6.1 | 1.9 | 4.2 | 39.1 | 48.4 |
| Ours, SPEAR$_{s+a}$ Large | 327M | 97k | 2.6 | 4.8 | 1.7 | 3.4 | 39.2 | 49.6 |
| Ours, SPEAR$_{s+a}$ XLarge | 600M | 197k | **2.5** | **4.6** | **1.6** | **2.9** | 39.4 | **50.0** |

predict the class probability of 527 sound event classes. Binary cross-entropy is used as the training objective, and the mean average precision (mAP) is measured on the AudioSet evaluation set. We compare SPEAR models with existing state-of-the-art audio SSL models.

**Results** The results presented in Table 4 demonstrate that SPEAR learns high-quality speech and audio representations that transfer well to the downstream tasks. For speech, our SPEAR$_s$ Base and Large achieve relative WER reductions of 25.9% and 13.2% on the test-other set in LS-960 compared to their WavLM counterparts, with similar model size and pre-training data. For general audio, the SPEAR$_a$ Base model achieves an mAP of 49.3 on AS-2M, surpassing all other audio SSL models with similar sizes[3] (except Dasheng Base, which is pre-trained with 20 times more general audio data), and SPEAR$_a$ Large improves further to 49.8 on AS-2M. These results highlight the effectiveness of using fine-grained MVQ tokens as pre-training targets in SPEAR.

Moreover, the dual-domain models successfully learn a unified representation space capable of handling both tasks with minimal performance loss, achieving ASR and AT performance comparable to their single-domain counterparts, and this performance gap diminishes as model capacity increases. Specifically, the WER for SPEAR$_{s+a}$ Large model on test-other is only 0.1 higher than the speech-domain specialist SPEAR$_s$ Large, while its mAP is only 0.2 lower than the SPEAR$_a$ Large. This demonstrates that with sufficient model capacity, our dual-domain pre-training scheme enables a single model to learn a unified representation space with strong capability for both domains. It should be noted that this versatility is particularly important, since the single-domain models yield poor cross-domain capability (see SPEAR$_s$ on AT or SPEAR$_a$ and Dasheng on ASR). Finally, our largest model SPEAR$_{s+a}$ XLarge further improves the performance on ASR and AT, setting a new state-of-the-art for SSL models on AS-2M AT task by achieving an mAP of 50.0.

In summary, the experiments in Table 4 suggest that the representations learnt through SPEAR adapt well to both domains after fine-tuning, proving its strong capability of learning both domain-specific and unified representations that excel in both speech and general audio domains.

## 5.2 SUPERB EVALUATION

**Setup** Experiments are carried out on SUPERB (Yang et al., 2021; Tsai et al., 2022), a benchmark for evaluating SSL models on a wide range of speech processing tasks. We follow the standard SUPERB evaluation protocol, using a weighted sum of the frozen intermediate representations from

---

[3]A strict comparable AT fine-tuning setup with all models pre-trained with the same dataset is shown in Appendix G.1.1, where our Base-scaled SPEAR$_a$ model consistently outperforms SOTA audio SSL models.

Table 5: Results on SUPERB. Best results in **bold**, 2nd best underlined. F1 reported for SF and PESQ for SE. Other task metrics described in Appendix E. USAD models from Chang et al. (2025).

| Model | # Param | Pre-train data | Understanding | | | | | | Paralinguistic | | | Enhancement | |
|---|---|---|---|---|---|---|---|---|---|---|---|---|---|
| | | | PR↓ | ASR↓ | IC↑ | KS↑ | SF↑ | ST↑ | SID↑ | SV↓ | ER↑ | SE↑ | SS↑ |
| **Speech SSL models** | | | | | | | | | | | | | |
| WavLM Base+ | 95M | 94k | 3.5 | 3.92 | 99.00 | 97.37 | 90.6 | 24.3 | 89.4 | 4.07 | 68.7 | 2.63 | 10.85 |
| WavLM Large | 317M | 94k | 3.1 | 3.44 | 99.31 | 97.86 | 92.2 | 26.6 | 95.5 | 3.77 | 70.6 | 2.70 | 11.19 |
| Ours, SPEAR$_s$ Base | 94M | 84k | 3.4 | 3.46 | 99.17 | 97.50 | 91.0 | 24.4 | 90.5 | 3.75 | 69.2 | 2.64 | 10.84 |
| Ours, SPEAR$_s$ Large | 327M | 84k | **2.6** | 3.27 | 99.47 | 97.89 | 92.8 | 26.2 | 95.5 | 3.14 | 72.1 | 2.71 | 11.20 |
| **Audio SSL models** | | | | | | | | | | | | | |
| BEATs | 90M | 5k | 36.4 | 36.4 | 97.70 | 53.40 | - | - | 57.1 | - | 64.5 | - | - |
| EAT | 88M | 5k | 55.0 | 25.9 | 92.80 | 62.50 | - | - | 45.0 | - | 62.5 | - | - |
| ATST Frame | 86M | 5k | 20.4 | 18.8 | 95.10 | 85.40 | - | - | 69.8 | - | 64.4 | - | - |
| Dasheng 1.2B | 1.2B | 272k | 14.3 | 13.8 | 98.13 | 97.73 | - | - | 92.4 | - | 68.7 | - | - |
| **Speech+Audio Models** | | | | | | | | | | | | | |
| USAD Base | 94M | 126k | 5.1 | 7.70 | 98.30 | 97.10 | - | - | 88.6 | - | 68.0 | - | - |
| USAD Large | 330M | 126k | 4.0 | 6.50 | 98.40 | 97.10 | - | - | 91.2 | - | 68.4 | - | - |
| Ours, SPEAR$_{s+a}$ Base | 94M | 97k | 3.9 | 3.76 | 98.05 | 97.58 | 90.5 | 24.1 | 90.0 | 3.85 | 69.4 | 2.66 | 10.89 |
| Ours, SPEAR$_{s+a}$ Large | 327M | 97k | 3.1 | 3.39 | 99.40 | 97.92 | 92.1 | 25.6 | 95.0 | 3.30 | 71.6 | **2.72** | 11.12 |
| Ours, SPEAR$_{s+a}$ XLarge | 600M | 97k | 2.9 | **3.19** | **99.61** | **98.12** | **92.9** | **26.7** | **96.3** | **2.86** | **73.3** | **2.72** | **11.24** |

the SSL models. For better readability, we group the SUPERB tasks into three categories: Understanding, Paralinguistic, and Enhancement. We select representative tasks within each category and report their results in Table 5. The primary comparison is made against the WavLM Large, which is the current SOTA on SUPERB. Further details on the SUPERB evaluation, including the individual task information and complete results on SUPERB, can be found in Appendix E.

**Results**   As can be seen from Table 5, SPEAR improves across the range of speech tasks, achieving notable gains across all three task categories. With the same model size and pre-training corpora, our speech-domain SPEAR$_s$ Large model outperforms the current state-of-the-art WavLM Large on nearly every task. The improvement in paralinguistic capabilities is particularly noteworthy. For instance, SPEAR$_s$ Large achieves a 16.7% relative reduction of equal-error-rate on speaker verification (SV) and a 1.48% absolute accuracy improvement on emotion recognition (ER). This suggests that our fine-grained masked-token prediction objective helps the model learn richer paralinguistic information beyond speech content alone, than by using k-means clustered tokens. An analysis of the feature subspaces learned through the fine-grained MVQ tokens is conducted in Appendix G.4.

It is evident that audio-only SSL models generally underperform on the SUPERB benchmark. Even with a very large pre-training corpus of 272k hours, Dasheng 1.2B consistently performs more poorly than our much smaller SPEAR$_s$ Large, especially on tasks requiring higher-level semantic or phonetic understanding. This gap could be attributed to the MAE (He et al., 2022) objective used by Dasheng, which attempts to reconstruct the input acoustic features, making the model focus more on low-level acoustic details rather than high-level semantic structures necessary for speech understanding. In contrast, the fine-grained masked-token prediction objective in SPEAR allows the model to learn semantic structures while retaining acoustic details. Therefore, our dual-domain SPEAR models maintain strong performance on SUPERB. Despite a slight degradation in some understanding and paralinguistic tasks, the SPEAR$_{s+a}$ Large notably outperforms its speech-only counterpart on keyword spotting (KS) and speech enhancement (SE), indicating a positive synergy from the dual-domain pre-training on these tasks. SPEAR$_{s+a}$ Large also outperforms USAD Large (Chang et al., 2025) comprehensively, another unified speech and audio model trained via matching the representations of two teacher models, demonstrating the advantage of the SSL objective defined by SPEAR. Finally, by scaling up the model size and training data, SPEAR$_{s+a}$ XLarge, our largest dual-domain model, pushes the performance boundary even further, establishing new state-of-the-art on multiple SUPERB tasks, while being more versatile than speech-only models.

## 5.3   HEAR EVALUATION

**Setup**   To assess the general audio capabilities of our models, experiments are conducted on the HEAR benchmark (Turian et al., 2022), which evaluates audio representations across 19 diverse tasks. The final-layer representations are used for evaluation unless otherwise specified. For clarity, the average scores for each of the three task categories are reported: Environment, Speech and

Table 6: Results on the HEAR benchmark. The group-wise average score and the overall average score are reported. Rows with grey background: results obtained by using concatenation of all layers' features. Best results in **bold**, 2nd best results underlined.

| Model | # Params | Pre-train Data | Env | Speech | Music | Average |
|---|---|---|---|---|---|---|
| ***Speech Models*** | | | | | | |
| WavLM Base+ (Chen et al., 2022) | 95M | 94k | 57.28 | 68.14 | 61.31 | 62.69 |
| WavLM Large (Chen et al., 2022) | 317M | 94k | 72.86 | 72.69 | 65.77 | 69.65 |
| Ours, SPEAR$_s$ Base | 94M | 84k | 73.09 | 73.41 | 70.66 | 72.12 |
| Ours, SPEAR$_s$ Large | 327M | 84k | 72.74 | 74.80 | 71.68 | 72.96 |
| ***Audio Models*** | | | | | | |
| BEATs (Chen et al., 2023) | 90M | 5k | 73.23 | 62.40 | 77.52 | 71.05 |
| Dasheng-Base (Dinkel et al., 2024) | 86M | 272k | 80.18 | 72.48 | 84.00 | 79.31 |
| Dasheng 0.6B (Dinkel et al., 2024) | 600M | 272k | 82.95 | 74.82 | 84.73 | 81.03 |
| Dasheng 1.2B (Dinkel et al., 2024) | 1.2B | 272k | 83.20 | 75.72 | **84.86** | 81.44 |
| Ours, SPEAR$_a$ Base (5k) | 94M | 5k | 77.83 | 69.74 | 80.61 | 76.37 |
| Ours, SPEAR$_a$ Large (5k) | 327M | 5k | 78.16 | 72.94 | 81.80 | 78.08 |
| Ours, SPEAR$_a$ Base | 94M | 13k | 80.33 | 69.87 | 80.33 | 77.01 |
| Ours, SPEAR$_a$ Large | 327M | 13k | 83.58 | 72.70 | 81.85 | 79.18 |
| Ours, SPEAR$_a$ Base | 94M | 13k | 83.61 | 71.98 | 83.26 | 79.85 |
| Ours, SPEAR$_a$ Large | 327M | 13k | **84.97** | 73.01 | 84.62 | 80.83 |
| ***Speech & Audio Models*** | | | | | | |
| USAD Base (Chang et al., 2025) | 94M | 126k | 80.67 | 73.72 | 79.31 | 77.75 |
| USAD Large (Chang et al., 2025) | 330M | 126k | 81.97 | 74.48 | 81.7 | 79.36 |
| Ours, SPEAR$_{s+a}$ Base | 94M | 97k | 80.66 | 73.73 | 79.29 | 77.75 |
| Ours, SPEAR$_{s+a}$ Large | 327M | 97k | 81.10 | 76.47 | 80.42 | 79.26 |
| Ours, SPEAR$_{s+a}$ XLarge | 600M | 197k | 81.74 | 76.76 | 80.92 | 79.72 |
| Ours, SPEAR$_{s+a}$ Base | 94M | 97k | 82.58 | 77.3 | 81.97 | 80.55 |
| Ours, SPEAR$_{s+a}$ Large | 327M | 97k | 84.38 | 78.84 | 82.69 | 81.78 |
| Ours, SPEAR$_{s+a}$ XLarge | 600M | 197k | 84.69 | **79.72** | 83.69 | **82.33** |

Music, along with the overall average score in Table 6. More information regarding the tasks in HEAR and detailed results on individual tasks are provided in Appendix F. Apart from the SPEAR models in Table 2, two additional SPEAR$_a$ models in Base and Large architectures are pre-trained using only AudioSet 5k hours, denoted as SPEAR$_a$ {Base, Large} (5k).

**Results**    As shown in Table 6, all speech-domain models show very limited overall performance on the general-audio-focused HEAR benchmark, highlighting the need to incorporate general audio data during pre-training. Nonetheless, our SPEAR$_s$ models, even the SPEAR$_s$ Base, yield a higher overall score than the much bigger WavLM Large pre-trained with coarse k-means tokens, indicating that the fine-grained discrete tokens manage to capture general-audio-related information despite being extracted from a speech SSL model.

Our audio-domain SPEAR$_a$ models demonstrate strong performance on environment-related tasks. Notably, SPEAR$_a$ Large achieves 83.58 on the environment category, surpassing the best performing audio SSL model Dasheng 1.2B (83.2), with only a quarter of the model parameters and far less pre-training data. However, the performance of SPEAR$_a$ Large on speech and music tasks still trails that of Dasheng 1.2B, a gap we attribute to the significantly smaller scale of pre-training data and model size[4]. Despite this gap, it is noteworthy that SPEAR$_a$ benefits from scaling up the audio training data: increasing the audio data from 5k to 13k hours leads to a substantial improvement in the overall HEAR score, highlighting the potential of SPEAR for audio SSL at larger scales.

Finally, our dual-domain SPEAR$_{s+a}$ models consistently outperform their single-domain counterparts, highlighting the benefits of our unified pre-training framework. The SPEAR$_{s+a}$ Large achieves an average score of 79.26, surpassing both SPEAR$_s$ Large (72.96) and SPEAR$_{s+a}$ Large (79.18), indicating that SPEAR successfully unifies the representation space of both domains through joint pre-training on both speech and audio data, making it a more versatile model. Compared to USAD Large trained with multi-teacher knowledge distillation, our SPEAR$_{s+a}$ Large pre-

---

[4]A comparison of Dasheng and SPEAR using the same amount (5k hours) of training data is in Appendix I, where we show SPEAR$_a$ models outperforms Dasheng by a large margin with this amount of training data.

trained on a smaller corpora achieves a significant absolute improvement of 2.42 on the average score under the same evaluation setup of using intermediate representations. This again highlights the strength of SPEAR as a unified SSL framework for learning generic speech and audio representations jointly through fine-grained masked-token prediction. Another controlled comparison between SPEAR and USAD (same teachers, similar model size and training data) is presented in Appendix H, where SPEAR still consistently outperforms USAD. Finally, compared to the much larger Dasheng 1.2B, SPEAR$_{s+a}$ XLarge achieves stronger performance on speech-related tasks while trailing on environment and music tasks, which can be attributed to its smaller model size and imbalanced pre-training data composition (only 13k hours from the 197k hours are general-audio data). However, this performance gap can be reversed by leveraging all intermediate layer representations.

### 5.4 Ablation Studies

In order to provide an in-depth understanding of the core components of SPEAR, we performed extensive ablation studies. Here, we summarise the key aspects investigated and their corresponding findings. Detailed experimental setups and analyses are presented in Appendix G.

**Different Teacher Models**   We investigated the impact of utilising teacher models with varying sizes and training data scales in Appendix G.1.1. The key findings include:

- *Stronger teacher models generally lead to a stronger student model.* This suggests that it is necessary to select powerful teacher models for the optimal performance of SPEAR.
- *Student models are not upper-bounded by their teacher models.* Given the same training data and model size, the models trained with SPEAR are capable of outperforming their teachers, suggesting that the fine-grained masked prediction objective defined by SPEAR helps discover better features than the original teacher.

**MVQ vs k-means**   We conducted a controlled experiment to compare pre-training performance using MVQ tokens versus k-means tokens (see Appendix G.5). We observe that the model trained with MVQ tokens consistently achieves better performance across all tasks. This confirms that *the fine-grained nature of MVQ tokens conveys richer information from the teacher model than coarse k-means tokens.* Additionally, a visualisation of the embedding space (Appendix G.4) reveals that the MVQ quantiser exhibits a much clearer separation of different speakers compared to k-means.

**Dual-domain Training Strategy**   Two extra dual-domain pre-training strategies are compared against the asymmetrical strategy adopted by SPEAR (see Section 3.2.2), and results are shown in Appendix G.7. We find that *the asymmetrical design adopted by SPEAR achieves a more balanced performance across speech and audio domains due to the dominance of speech data in the mixed speech audio training set.* This motivates us to enlarge the proportion of general audio data (e.g., by curating a larger general audio dataset) in the training corpora for our future work. Furthermore, we experimented with varying $\lambda$ (Equation 6) to control the contribution of audio-specific loss, finding that $\lambda = 0.1$ yields the optimal result.

Beyond the studies above, we also investigated the effect of the weighting factor $\lambda$ in the masked-prediction pre-training loss (Appendix G.2), the number of codebooks in MVQ (Appendix G.3), and encoder architectures (Zipformer vs Transformer) for SPEAR (Appendix G.6).

## 6 Conclusions

In this work, we propose SPEAR, a unified SSL framework for both speech and general audio domains, learning unified and generic representations across both domains. By leveraging multi-codebook vector quantisation to generate fine-grained discrete speech and audio tokens, SPEAR performs fine-grained masked-token prediction as the pre-training task for representation learning. Based on this, an asymmetrical dual-domain pre-training pipeline is designed to balance the performance across both domains. To the best of our knowledge, SPEAR is the first SSL framework to successfully learn unified speech and audio representations from a mixture of speech and audio. The downstream fine-tuning experiments, along with the evaluation of frozen representations on two major benchmarks for evaluating speech and general-audio representations (SUPERB and HEAR), demonstrate the effectiveness of SPEAR in learning unified and generic speech and audio representations. Our dual-domain model with 600M parameters excels in both domains, making it one of the most powerful and versatile open-source SSL models for auditory understanding.

## 7 ETHICS STATEMENT

We recognise that powerful audio representation models could be misused. The technology presented could serve as a foundation for applications we do not endorse, such as non-consensual speaker identification, mass surveillance, or the generation of synthetic audio for disinformation. Our goal in releasing these models is to enable transparency and accelerate positive academic innovation. We strongly condemn any application of our research to unethical ends.

## 8 REPRODUCIBILITY STATEMENT

To ensure the reproducibility of our research and to support further advancements in the field, we will make our resources publicly available, including the inference code and the model checkpoints. All essential training details, including model configurations and hyperparameters, have been thoroughly documented in the main paper as well as Appendix C.2 and Appendix C.3. We hope that by providing these resources, more researchers can contribute to the development of this exciting research area. Details regarding access and implementation will be updated after the double-blind review. We invite the community to build upon our work to further advance the research field.

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

## A    LLM USAGE

We used LLMs in paper presentation for the purpose of correcting grammatical errors and spelling.

## B    MULTI-CODEBOOK VECTOR QUANTISATION

Here, we present more details regarding the MVQ quantiser to supplement Section 3.1.

### B.1    ENCODE AND DECODE

A Multi-codebook Vector quantisation (MVQ) module consists of $N$ codebooks, each containing $K$ codebook vectors. Given a $d$-dimensional representation $\boldsymbol{x} \in \mathbb{R}^d$, the MVQ quantiser $\mathcal{Q}$ encodes it to a sequence of integers (i.e, tokens) from a finite discrete value space $[0, \ldots, K-1]$.[5] The encoded integers are denoted as MVQ tokens, which can be used for reconstructing the original input through a $\mathrm{Decode}(\cdot)$ operation:

$$\boldsymbol{z} = \mathrm{Encode}(\boldsymbol{x}; \mathcal{Q}), \tag{7}$$
$$\hat{\boldsymbol{x}} = \mathrm{Decode}(\boldsymbol{z}; \mathcal{Q}). \tag{8}$$

The MVQ quantiser performs a mapping $f : \mathbb{R}^d \to \mathbb{C}^N$, where $\mathbb{C}$ denotes a fixed-sized discrete value space $\{0, \ldots, K-1\}$. $\boldsymbol{z} = \{z_1, \ldots, z_N\} \in \mathbb{C}^N$ is the MVQ tokens with $z_n \in \{0, \ldots, K-1\}$ and $\hat{\boldsymbol{x}}$ is the reconstructed input. The reconstruction operation follows a direct-sum scheme, where one codebook vector is selected from each codebook, resulting in a summation over $N$ codebook vectors:

$$\hat{x} = \sum_{n=1}^{N} \boldsymbol{C}_{z_n}^n, \tag{9}$$

where $\boldsymbol{C}_{z_n}^n \in \mathbb{R}^d$ is the $z_n$-th entry code vector in the $n$-th codebook. Each codebook $\boldsymbol{C}^n = \{\boldsymbol{c}_0^n, \ldots, \boldsymbol{c}_{K-1}^n\}$ is a matrix consisting of $K$ code vectors. As can be seen, $z_n$ denotes the encoded index of the code vector in the $n$-th codebook, i.e., which code vector to choose from the $n$-th codebook for reconstruction.

The encoding process aims to find $\boldsymbol{z}$ that leads to the lowest reconstruction error: $E[||\hat{\boldsymbol{x}} - \boldsymbol{x}||_2^2]$. Naively enumerating all combinations of $z_n$ is impractical, so a heuristic encoding algorithm is utilised to reduce the search space while maintaining a relatively low reconstruction error. The

---

[5]For storage efficiency, we always use $K = 256$ since the indices can be stored with uint8 format. However, it is theoretically possible to increase $K$ to a bigger number.

MVQ quantiser employs $N$ neural classifiers $\mathcal{G}_n$ to first generate an initial estimation of the encoded index for each codebook, denoted as $z_{\text{init}}$, and iteratively refines $z_{\text{init}}$ for a fixed number of steps, e.g., 5. The mechanism of the refinement algorithm is out of the scope of our work, and we direct readers to the original MVQ paper (Guo et al., 2023) for more details.

### B.2 MVQ Training

The trainable parameters in the MVQ quantiser are the codebooks $C^n$ and $N$ neural classifiers $\mathcal{G}_n$. For each input float vector $x$ and its encoding $z = \text{Encode}(x)$, the training loss for MVQ is formulated as follows:

$$\mathcal{L} = \mathcal{L}_{\text{residual}} + \mathcal{L}_{\text{prediction}} + \beta \mathcal{L}_{\text{reg}} \tag{10}$$

$$= ||x - \text{Decode}(z; \mathcal{Q})||_2^2 + \sum_{n=1}^{N} -\log \mathcal{G}_n(x)_{z_n} + \beta \mathcal{L}_{\text{reg}}, \tag{11}$$

where $\mathcal{G}_n(x)_{z_n}$ is the predicted probability of choosing $z_n$. The first term $\mathcal{L}_{\text{residual}}$ is the L2-squared reconstruction loss and optimises the code vectors. The second term $\mathcal{L}_{\text{prediction}}$ encourages the neural classifiers to select the encoded indexes $z$ obtained through the refinement algorithm. By doing so, the initial estimate $z_{\text{init}}$ predicted by $\mathcal{G}_n$ is expected to be close to the actual encodings $z$, most likely with a lower reconstruction error. The last term $\mathcal{L}_{\text{reg}}$ is an auxiliary regularisation loss to encourage a balanced code usage within each codebook, and $\beta$ is the scale for this auxiliary loss.

## C MODEL SPECIFICATION AND TRAINING SETTINGS

### C.1 MODEL SPECIFICATION

The model specifications of Base, Large, and XLarge variants of SPEAR are presented in Table 7. The model configuration is determined by the configuration of the Zipformer (Yao et al., 2024) encoder, which adopts a stack-wise design, with each stack consisting of multiple layers operating at a specific downsampling factor. The Zipformer Encoder is characterised by the following attributes:

- **Model Dimension**: the dimensionality of the output representations.
- **Feedforward Dimension**: the dimensionality of the feedforward module.
- **Attention Heads**: the number of attention heads.
- **Encoder Layers**: the number of Zipformer layers per stack.
- **Downsampling Ratio**: the relative temporal downsampling factor to the input representations (i.e., 100 Hz filterbank features).
- **CNN Kernel Size**: the kernel size of the convolutional module in each layer.

Table 7: The configurations of the Zipformer encoder in different versions of SPEAR.

|  | Base | Large | XLarge |
|---|---|---|---|
| Number of parameters | 94M | 327M | 600M |
| Model dimension | 512 | 1024 | 1280 |
| Feedforward dimension | 1536 | 3072 | 3840 |
| Attention heads | 8 | 8 | 8 |
| Encoder layers | 1,2,3,3,1,1,1 | 1,2,2,3,1,1,1 | 1,2,3,4,1,1,1 |
| Downsampling ratio | | 1,2,4,8,4,2,1 | |
| CNN kernel size | | 31,31,15,15,15,31,31 | |

### C.2 PRE-TRAINING RESOURCES AND CONFIGURATIONS

The training hyperparameters and the required computing resources for training SPEAR are presented in Table 8. All models in SPEAR are trained using the NVIDIA A800 (80GB) GPUs. As

shown in WavLM (Chen et al., 2022), applying data augmentations to the input audio can improve the pre-training performance. Therefore, we apply both in-batch utterance mixing and noise mixing during training. MUSAN (Snyder et al., 2015) is used as the noise dataset. The optimiser and scheduler settings follow Yao et al. (2024), where the ScaledAdam optimiser and Eden scheduler are used.

Table 8: Hyperparameters and computing resources required for pre-training. Batch size denotes the total duration of speech (audio) in seconds. Approximate total GPU hours (not elapsed time) also reported.

| | Speech Pre-train | | Audio Pre-train | | Speech & Audio Pre-train | | |
| --- | --- | --- | --- | --- | --- | --- | --- |
| | Base | Large | Base | Large | Base | Large | XLarge |
| **Hyperparameters** | | | | | | | |
| Learning rate | | | | 0.045 | | | |
| Total steps | 400k | 500k | 250k | 250k | 400k | 500k | 500k |
| Batch size | 4.8k | 4.8k | 4.8k | 4.8k | 6.4k | 6.4k | 6.4k |
| Utterance mix prob | 0.1 | 0.1 | - | - | - | - | - |
| Noise mix prob | 0.1 | 0.2 | 0.5 | 0.5 | 0.5 | 0.5 | 0.5 |
| $\alpha$ (see Equation 2) | | | | 0.5 | | | |
| $\lambda$ (see Equation 6) | - | - | - | - | 0.1 | 0.1 | 0.1 |
| **Computing Resources** | | | | | | | |
| Num GPUs | 8 | 8 | 8 | 8 | 8 | 16 | 32 |
| GPU hours (approx.) | 460 | 900 | 290 | 560 | 660 | 2,000 | 3,800 |

### C.3 FINE-TUNING CONFIGURATIONS

The fine-tuning configurations for the downstream ASR tasks and AT tasks presented in Section 5.1 are shown in Table 9. We used Pruned RNN-T (Kuang et al., 2022), a memory-efficient variant of RNN-T for optimisation. An asynchronous learning rate policy is adopted during fine-tuning by setting a smaller learning rate for the pre-trained encoder parameters. In ASR experiments, 3-fold speed perturbation is applied to the training data. In AS-2M, we follow prior work (Gong et al., 2021) to adopt a weighted sampler to cope with the imbalanced label distribution in the full AudioSet. Mixup (Zhang et al., 2018) with a probability of 0.5 is used in AT fine-tuning.

Table 9: Fine-tuning configurations. "Encoder LR scale" denotes the relative ratio of the encoder learning rate. Batch size is measured in seconds.

| | ASR | | AT | |
| --- | --- | --- | --- | --- |
| | LS-100 | LS-960 | AS-20k | AS-2M |
| Learning rate | | 0.045 | | |
| Encoder LR scale | 0.1 | 0.1 | 0.2 | 0.1 |
| Num epochs | 90 | 90 | 20 | 40 |
| Batch size | 2000 | 4800 | 2000 | 4000 |
| MUSAN (Snyder et al., 2015) | | ✓ | | |
| SpecAugment (Park et al., 2019) | | ✓ | | |
| Weighted sampling (Gong et al., 2021) | - | - | ✓ | ✓ |
| MixUp (Zhang et al., 2018) | - | - | 0.5 | 0.5 |

## D  LIBRISPEECH FINE-TUNING EXPERIMENTS

To evaluate the adaptation capability of SPEAR under limited supervision, we fine-tune the models on 10h and 100h subsets of the LibriSpeech corpus. Following prior work (Hsu et al., 2021; Chen et al., 2022), we use the same CTC decoder with graphemes as modelling units and the same decoding process for fair comparison. The CTC vocabulary consists of the 26 English letters, a space, an

apostrophe, and a blank symbol. Decoding with an external language model is performed using the wav2letter++ beam search decoder (Pratap et al., 2019), formulated as:

$$\log p_{\text{CTC}}(\boldsymbol{y} \mid \boldsymbol{x}) + w_1 \log p_{\text{LM}}(\boldsymbol{y}) + w_2 \, |\boldsymbol{y}|, \tag{12}$$

where $\boldsymbol{x}$ is the input audio, $\boldsymbol{y}$ is the predicted text sequence, $|\boldsymbol{y}|$ denotes its length, and $w_1, w_2$ are the language model and word score coefficients, respectively.

Table 10: LibriSpeech fine-tuning results with limited supervised data. Best results in **bold**, and second-best results are underlined in each section.

| Model | # Params | LM | LS-100 | | LS-10 | |
|---|---|---|---|---|---|---|
| | | | test-clean | test-other | test-clean | test-other |
| WavLM Base (Chen et al., 2022) | 95M | None | 5.7 | 12.0 | 9.8 | 16.0 |
| WavLM Base+ (Chen et al., 2022) | 95M | None | 4.6 | 10.1 | 9.0 | 14.7 |
| Ours, SPEAR$_s$ Base | 94M | None | 3.1 | 6.0 | 5.2 | 8.2 |
| Ours, SPEAR$_{s+a}$ Base | 94M | None | 3.3 | 6.6 | 5.6 | 9.2 |
| Ours, SPEAR$_s$ Large | 327M | None | **2.6** | **4.8** | **4.6** | **6.9** |
| Ours, SPEAR$_{s+a}$ Large | 327M | None | **2.6** | 4.9 | 4.9 | 7.3 |
| Ours, SPEAR$_{s+a}$ XLarge | 600M | None | **2.6** | 4.9 | 4.8 | **6.9** |
| HuBERT Base (Hsu et al., 2021) | 95M | 4-gram | 3.4 | 8.1 | 4.3 | 9.4 |
| WavLM Base (Chen et al., 2022) | 95M | 4-gram | 3.4 | 7.7 | 4.3 | 9.2 |
| WavLM Base+ (Chen et al., 2022) | 95M | 4-gram | 2.9 | 6.8 | 4.2 | 8.8 |
| WavLM Large (Chen et al., 2022) | 317M | 4-gram | **2.3** | 4.6 | **2.9** | 5.5 |
| Ours, SPEAR$_s$ Base | 94M | 4-gram | 2.4 | 5.0 | 3.2 | 6.0 |
| Ours, SPEAR$_{s+a}$ Base | 94M | 4-gram | 2.7 | 5.3 | 3.6 | 6.9 |
| Ours, SPEAR$_s$ Large | 327M | 4-gram | **2.3** | **4.2** | **2.9** | **5.1** |
| Ours, SPEAR$_{s+a}$ Large | 327M | 4-gram | 2.4 | 4.4 | 3.1 | 5.6 |
| Ours, SPEAR$_{s+a}$ XLarge | 600M | 4-gram | **2.3** | 4.3 | **2.9** | 5.2 |

**Result Interpretation**   As shown in Table 10, the speech-domain SPEAR$_s$ models consistently outperform their WavLM counterparts under both Base and Large scales, regardless of decoding with an external 4-gram language model. Our SPEAR$_s$ Large yields the lowest WERs on both LS-10 and LS-100 setups, implying that the MVQ tokens used during pre-training transfer rich semantic information to the student model. Interestingly, the speech-domain SPEAR$_s$ models slightly outperform their dual-domain SPEAR$_{s+a}$ counterparts in this CTC setting, especially when supervision is scarce. We hypothesise this stems from the nature of the representation space: a unified representation space for speech and general audio is inherently more complex than one specialised for speech. Consequently, adapting the unified representation space for the ASR task becomes more challenging with insufficient supervised data, particularly when using a simple, letter-based CTC decoder.

## E   SUPERB EVALUATION

In this section, we provide more detail about the SUPERB benchmark (Yang et al., 2021; Tsai et al., 2022) as a supplement to Section 5.2 and present the complete results on SUPERB benchmark. A summary of the SUPERB tasks is shown in Table 11. Complete results of SPEAR models along with other existing speech SSL models are shown in Table 12 and Table 13.

**Result Interpretation**   As shown in Table 12 and Table 13, our speech-domain model SPEAR$_s$ demonstrates very high performance on understanding, paralinguistics, and enhancement tasks on SUPERB. Our speech-domain model SPEAR$_s$ Large achieves the same or better performance on 12 out of 15 tasks on SUPERB (except for ST, QbE, and VC) compared to WavLM Large, the previous state-of-the-art model on SUPERB, with the same pre-training corpora and similar model size. This again suggests that the performance of the SPEAR framework is not constrained by the teacher model, since SPEAR$_{s+a}$ Large is pre-trained with fine-grained targets generated by WavLM Large. It has also been observed that performing dual-domain training leads to slight performance degradation on understanding and paralinguistic tasks compared to the speech-only models. However, improvement on KWS and enhancement tasks is also observed, suggesting a positive task

Table 11: Detailed task information in SUPERB.

| Task Name | Metric(s) |
|---|---|
| Speaker Identification (SID) | Accuracy |
| Automatic Speaker Verification (ASV) | Equal Error Rate (EER) |
| Speaker Diarization (SD) | Diarization error rate (DER) |
| Emotion Recognition (ER) | Accuracy |
| Phoneme Recognition (PR) | Phone Error Rate (PER) |
| Automatic Speech Recognition (ASR) | Word Error Rate (WER) |
| Out-of-domain ASR (ar/es/zh) | Word (character) Error Rate |
| Keyword Spotting (KS) | Accuracy |
| Query by Example Spoken Term Detection (QbE) | Maximum Term Weighted Value (MTWV) |
| Speech Translation (ST) | BLEU |
| Intent Classification (IC) | Accuracy |
| Slot Filling (SF) | F1, Character Error Rate (CER) |
| Speech Enhancement (SE) | Perceptual Evaluation of Speech Quality (PESQ) Short-Time Objective Intelligibility (STOI) |
| Speech Separation (SS) | Scale-invariant Signal-to-distortion Ratio improvement (SI-SDRi) |
| Voice Conversion (VC) | MCD (Mel Cepstral Distortion), WER, EER |

Table 12: Full results of understanding tasks on the SUPERB benchmark. Best results in **bold**, the 2nd best results are underlined.

| Model | # Params | Pre-train Data | Understanding | | | | | | | |
|---|---|---|---|---|---|---|---|---|---|---|
| | | | PR | ASR | OOD-ASR | KS | QbE | ST | IC | SF |
| | | | PER ↓ | WER ↓ | WER ↓ | Acc ↑ | MTWV ↑ | BLEU ↑ | Acc ↑ | F1 ↑ CER ↑ |
| FBANK | 0 | - | 82.01 | 23.18 | 63.58 | 8.63 | 0.0058 | 2.32 | 9.10 | 69.64 52.94 |
| *Existing Speech SSL models* | | | | | | | | | | |
| WavLM Base+ (Chen et al., 2022) | 95M | 94k | 3.92 | 5.59 | 38.32 | 97.37 | **0.0988** | 24.25 | 99.00 | 90.58 21.20 |
| wav2vec 2.0 Large (Baevski et al., 2020) | 317M | 60k | 4.25 | 3.75 | 44.89 | 96.66 | 0.0480 | 12.48 | 95.28 | 87.11 27.31 |
| HuBERT Large (Hsu et al., 2021) | 317M | 60k | 3.53 | 3.62 | 44.08 | 95.29 | 0.0353 | 20.10 | 98.76 | 89.81 21.76 |
| WavLM Large (Chen et al., 2022) | 317M | 94k | 3.06 | 3.44 | 32.27 | 97.86 | 0.0886 | 26.57 | 99.31 | 92.21 18.36 |
| *Ours, Speech SSL models* | | | | | | | | | | |
| SPEAR$_s$ Base | 94M | 84k | 3.44 | 3.46 | 34.35 | 97.50 | 0.0772 | 24.37 | 99.17 | 90.96 19.22 |
| SPEAR$_s$ Large | 327M | 84k | **2.56** | 3.27 | 31.70 | 97.89 | 0.0768 | 26.20 | 99.47 | 92.25 17.86 |
| *Ours, Speech & Audio SSL models* | | | | | | | | | | |
| SPEAR$_{s+a}$ Base | 94M | 97k | 3.89 | 3.76 | 35.48 | 97.58 | 0.0801 | 24.07 | 98.05 | 90.54 20.14 |
| SPEAR$_{s+a}$ Large | 327M | 97k | 3.08 | 3.39 | 31.22 | 97.92 | 0.0712 | 25.64 | 99.40 | 92.07 18.04 |
| SPEAR$_{s+a}$ XLarge | 600M | 197k | 2.94 | **3.19** | **30.69** | **98.12** | 0.0745 | **26.66** | **99.61** | **92.86 17.23** |

synergy in these tasks. Finally, our largest dual-domain model, SPEAR$_{s+a}$ XLarge, improves upon SPEAR$_{s+a}$ Large, and further improves the best results of 12 SUPERB tasks, confirming that the SPEAR framework scales effectively with both model and data size.

It is worth noting that our results on the Voice Conversion (VC) task are not directly comparable to previous SSL models due to a change in the VC recipe within the SUPERB codebase, as pointed out by Shi et al. (2024a). Within our own experiments, we observe that our Large and XLarge variants underperform the Base model on VC. This is potentially due to overfitting on the small training dataset, an issue also observed in MR-HUBERT (Shi et al., 2024a). We leave the investigation of using our models for generation tasks as an important direction for future work, since a unified capability for both understanding and generation is highly desirable.

Table 13: Full results of paralinguistics, enhancement, and Generation tasks on the SUPERB benchmark. Best results in **bold**, the 2nd best results are underlined.

| Model | # Params | Pre-train Data | Paralinguistics | | | | Enhancement | | | | Generation | | |
| | | | SID | ASV | SD | ER | SE | | SS | | VC | | |
| | | | Acc ↑ | EER ↓ | DER ↓ | Acc ↑ | PESQ ↑ | STOI ↑ | SI-SDRi ↑ | | MCD ↓ | WER ↓ | ASV ↑ |
| FBANK | 0 | - | 0 | 9.56 | 10.05 | 35.39 | 2.55 | 93.6 | 9.23 | | 8.47 | 38.3 | 77.25 |
| *Existing Speech SSL models* | | | | | | | | | | | | | |
| WavLM Base+ (Chen et al., 2022) | 95M | 94k | 89.42 | 4.07 | 3.50 | 68.65 | 2.63 | 94.3 | 10.85 | | 7.40 | 8.1 | 99.00 |
| wav2vec 2.0 Large (Baevski et al., 2020) | 317M | 60k | 86.14 | 5.65 | 5.62 | 65.64 | 2.52 | 94.0 | 10.02 | | 7.63 | 15.8 | 97.25 |
| HuBERT Large (Hsu et al., 2021) | 317M | 60k | 90.33 | 5.98 | 5.75 | 67.62 | 2.64 | 94.2 | 10.45 | | **7.22** | **9.0** | 99.25 |
| WavLM Large (Chen et al., 2022) | 317M | 94k | 95.49 | 3.77 | 3.24 | 70.62 | 2.70 | 94.5 | 11.19 | | 7.30 | 9.9 | 99.00 |
| *Ours, Speech SSL models* | | | | | | | | | | | | | |
| SPEAR$_s$ Base | 94M | 84k | 90.5 | 3.75 | 3.57 | 69.21 | 2.64 | 94.3 | 10.84 | | 7.40 | 10.1 | 99.00 |
| SPEAR$_s$ Large | 327M | 84k | 95.49 | 3.14 | 3.20 | 72.10 | 2.71 | 94.5 | 11.20 | | 7.33 | 10.4 | 99.00 |
| *Ours, Speech & Audio SSL models* | | | | | | | | | | | | | |
| SPEAR$_{s+a}$ Base | 94M | 97k | 90.02 | 3.85 | 4.13 | 69.40 | 2.66 | 94.5 | 10.89 | | 7.34 | 10.2 | 99.00 |
| SPEAR$_{s+a}$ Large | 327M | 97k | 95.01 | 3.30 | 3.80 | 71.57 | 2.72 | 94.6 | 11.12 | | 7.42 | 10.7 | 99.00 |
| SPEAR$_{s+a}$ XLarge | 600M | 197k | **96.34** | **2.86** | **3.17** | **73.29** | 2.72 | 94.6 | 11.24 | | 7.44 | 10.9 | 99.00 |

## F   HEAR EVALUATION

Here, we provide further details on Holistic Evaluation of Audio Representations (HEAR) (Turian et al., 2022), a benchmark for evaluating audio representations as a supplement to Section 5.3. HEAR encompasses 19 tasks, which can be categorised into 3 groups: environment, speech, and music. Following prior work (Anton et al., 2023; Dinkel et al., 2024), we discard the Beehive task due to its overly long utterances and small sample size, leading to inconsistent results. The tasks can also be divided into frame-level tasks and clip-level tasks. The detailed task information is shown in Table 14 and the complete results on the HEAR benchmark are shown in Table 15.

Table 14: Individual task information in HEAR. *: frame-level task. Otherwise, clip-level task.

| Task Name | Group | Description | Metric |
|---|---|---|---|
| Beijing Opera Percussion (BJ) | Music | Classification of 6 Beijing Opera percussion instruments | Accuracy |
| CREMA-D (CD) | Speech | Speech emotion recognition | Accuracy |
| DCASE 2016 Task2 (D16)* | Environment | Office sound event detection in synthesized scenes | Onset FMS |
| ESC-50 (ESC) | Environment | Environmental sound classification | Accuracy |
| FSD50K (FSD) | Environment | Broad-domain audio multi-labeling | mAP |
| Gunshot Triangulation (Gun) | Environment | Identify location of microphone recording a gunshot | Accuracy |
| GTZAN Genre (GZ-Gen) | Music | Music genre classification. | Accuracy |
| GTZAN Music Speech (GZ-MS) | Music | Classification of audio into music or speech. | Accuracy |
| LibriCount (LC) | Speech | Multiclass speaker count identification. | Accuracy |
| MAESTRO 5h (MST)* | Music | Music transcription | Onset FMS |
| Mridingham Stroke (Mri-S) | Music | Non-Western pitched percussion, classification of stroke | Accuracy |
| Mridingham Tonic (Mri-T) | Music | Non-Western pitched percussion, classification of tonic | Accuracy |
| NSynth Pitch, 5h (NS-5) | Music | Pitch classification of synthesized sounds. | Pitch Acc |
| NSynth Pitch, 50h (NS-50) | Music | Pitch classification of synthesized sounds. | Pitch Acc |
| Speech Commands (v2), 5h (SC-5) | Speech | Spoken commands classification. | Accuracy |
| Speech Commands (v2), full (SC-F) | Speech | Spoken commands classification. | Accuracy |
| Vocal Imitations (VI) | Speech | Classification of vocal imitation to type of sound imitated | mAP |
| VoxLingua107 Top 10 (VL) | Speech | Spoken language identification. | Accuracy |

**Result Interpretation**   Our audio-domain models demonstrate strong performance on the environment-related tasks. Specifically, SPEAR$_{s+a}$ Large outperforms its teacher model, Dasheng 1.2B, on D16, ESC, and FSD, three well-known tasks for environmental audio understanding. It should be noted that this is achieved under the premise that Dasheng 1.2B is a much bigger model trained on over 20 times more data. The performance of SPEAR$_a$ Large is lower on speech and audio tasks compared to Dasheng 1.2B, due to the limited amount of general audio data in our setup. However, we anticipate SPEAR to outperform Dasheng 1.2B given a similar amount of general audio data for pre-training, which we leave as an important direction for future work.

By performing unified speech and audio pre-training that incorporates more speech data, the dual-domain model SPEAR$_{s+a}$ yields a notable improvement on speech-related tasks over the audio-

Table 15: Results on the HEAR benchmark. The last column is the average performance across all tasks. All results are the higher the better. Rows in grey: evaluated by concatenating the intermediate layers. Best results in **bold**, the 2nd best results are underlined.

| Model | # Params | BJ | CD | D16 | ESC | FSD | GZ-Gen | GZ-MS | Gun | LC | MST | Mri-S | Mri-T | NS-50 | NS-5 | SC-5 | SC-F | VI | VL | Avg |
|---|---|---|---|---|---|---|---|---|---|---|---|---|---|---|---|---|---|---|---|---|
| **Speech Model** | | | | | | | | | | | | | | | | | | | | |
| WavLM Base+ | 96M | 87.3 | 68.7 | 49.9 | 60.1 | 32.8 | 75.0 | 98.5 | 86.3 | 62.5 | 4.3 | 89.8 | 78.9 | 35.1 | 21.6 | 94.4 | 95.1 | 14.2 | 74.0 | 62.7 |
| HuBERT Large | 317M | 92.4 | 74.5 | 44.0 | 64.5 | 35.8 | 74.7 | 92.8 | 94.4 | 64.3 | 3.1 | 95.1 | 85.9 | 39.4 | 19.8 | 91.5 | 92.8 | 17.5 | 73.3 | 64.2 |
| WavLM Large | 317M | 91.5 | 75.5 | 85.1 | 68.6 | 40.1 | 80.0 | 94.4 | 97.6 | 70.3 | 8.8 | 96.0 | 88.8 | 43.8 | 23.0 | 94.8 | 96.1 | 19.5 | 79.9 | 69.7 |
| SPEAR$_s$ Base | 94M | 94.5 | 79.5 | 92.0 | 74.8 | 43.4 | 83.9 | 96.0 | 82.1 | 66.5 | 6.3 | 95.6 | 90.5 | 61.7 | 36.8 | 95.9 | 96.4 | 20.3 | 81.9 | 72.1 |
| SPEAR$_s$ Large | 327M | 91.5 | 80.9 | 84.2 | 73.9 | 43.3 | 82.5 | 96.1 | 89.6 | 71.1 | 8.6 | 95.8 | 89.7 | 67.7 | 41.6 | 95.5 | 95.7 | 20.7 | 85.0 | 73.0 |
| **Audio Model** | | | | | | | | | | | | | | | | | | | | |
| BEATs | 90M | 95.8 | 68.1 | 43.0 | 81.9 | 51.4 | 87.0 | 98.5 | 90.5 | 74.6 | 0.0 | 96.1 | 96.0 | 82.0 | 68.6 | 88.2 | 91.5 | 13.5 | 43.8 | 69.3 |
| ATST-Frame | 86M | 95.8 | 76.7 | 95.7 | 89.0 | 55.7 | 88.3 | 100.0 | 94.3 | 78.1 | 24.4 | 97.5 | 94.1 | - | 68.6 | 92.6 | 95.1 | 22.3 | 66.9 | - |
| Dasheng base | 86M | 93.6 | 78.7 | 93.9 | 82.9 | 51.0 | 89.2 | 99.2 | 92.9 | 76.6 | **43.9** | 96.1 | 94.9 | 83.3 | 71.8 | 95.9 | 97.1 | 16.7 | 69.9 | 79.3 |
| Dasheng 0.6B | 600M | 94.9 | 81.2 | 94.4 | 85.9 | 53.9 | 88.6 | 97.6 | 97.6 | 80.7 | 43.5 | 96.6 | 96.2 | 85.8 | 74.6 | 97.0 | 97.5 | 17.8 | 74.7 | 81.0 |
| Dasheng 1.2B | 1.2B | **96.2** | 81.6 | 94.2 | 85.3 | 54.2 | 88.8 | 97.7 | **99.1** | 79.6 | 43.3 | 96.8 | 96.1 | 85.6 | 74.4 | 97.1 | 97.9 | 19.4 | 78.7 | 81.4 |
| SPEAR$_a$ Base | 94M | 93.6 | 77.2 | 93.6 | 85.5 | 52.9 | 90.1 | 92.2 | 89.3 | 77.2 | 22.8 | 96.7 | 96.5 | 83.7 | 70.0 | 93.8 | 95.1 | 18.8 | 57.1 | 77.0 |
| SPEAR$_a$ Large | 327M | 94.9 | 79.8 | 94.5 | 86.6 | 54.4 | 89.1 | 96.8 | 98.8 | 79.6 | 23.6 | 96.9 | 96.3 | 86.4 | 70.8 | 95.3 | 96.3 | 19.0 | 66.2 | 79.2 |
| SPEAR$_a$ Base | 94M | **96.2** | 79.0 | 95.4 | 87.4 | 55.3 | 89.4 | **100.0** | 96.4 | 81.6 | 23.8 | 97.0 | 97.5 | 87.5 | 74.8 | 94.9 | 95.9 | 19.9 | 60.7 | 79.6 |
| SPEAR$_a$ Large | 327M | 95.8 | 79.9 | **96.5** | 89.6 | **57.4** | 90.7 | 98.5 | 96.4 | 82.4 | 25.6 | 97.4 | 98.1 | 89.6 | 81.4 | 95.8 | 96.9 | 20.5 | 62.7 | 80.8 |
| **Speech + Audio Model** | | | | | | | | | | | | | | | | | | | | |
| USAD Base | 94M | 95.8 | 80.0 | 93.6 | 82.2 | 52.2 | 94.0 | **100.0** | 86.3 | 78.7 | 26.7 | 97.3 | 95.7 | 81.6 | 57.0 | 96.6 | 97.6 | 19.5 | 76.0 | 78.4 |
| USAD Large | 330M | 94.1 | 79.5 | 93.9 | 83.4 | 53.0 | 87.4 | **100.0** | 97.6 | 79.1 | 38.4 | 97.4 | 96.1 | 83.2 | 57.0 | 97.0 | 97.5 | 18.5 | 75.3 | 79.4 |
| SPEAR$_{s+a}$ Base | 95M | 92.0 | 78.6 | 93.8 | 83.8 | 49.9 | 86.5 | 96.9 | 95.2 | 70.8 | 24.7 | 96.8 | 94.1 | 78.9 | 64.6 | 97.0 | 96.7 | 21.9 | 77.3 | 77.8 |
| SPEAR$_{s+a}$ Large | 327M | 94.9 | 81.4 | 93.8 | 85.1 | 51.4 | 87.6 | 96.4 | 94.1 | 76.2 | 26.9 | 96.8 | 96.0 | 80.0 | 64.8 | 97.5 | 97.2 | 22.6 | 83.9 | 79.3 |
| SPEAR$_{s+a}$ XLarge | 600M | 94.5 | 81.6 | 95.5 | 84.8 | 52.4 | 88.5 | 98.5 | 94.4 | 77.7 | 27.7 | 97.0 | 96.5 | 81.5 | 63.4 | 97.2 | 98.1 | 22.6 | 83.6 | 79.7 |
| SPEAR$_{s+a}$ Base | 95M | 95.3 | 82.0 | 95.1 | 85.9 | 54.2 | 88.8 | **100.0** | 95.2 | 76.2 | 26.8 | 97.2 | 96.0 | 82.2 | 69.4 | 97.3 | 98.2 | 24.6 | 85.6 | 80.6 |
| SPEAR$_{s+a}$ Large | 327M | 94.9 | **83.8** | 95.9 | 87.6 | 56.4 | 89.2 | 99.2 | 97.6 | 78.7 | 27.9 | 97.4 | 97.5 | 85.3 | 70.2 | 98.1 | 98.3 | 25.7 | 88.5 | 81.8 |
| SPEAR$_{s+a}$ XLarge | 600M | 95.3 | 83.6 | **96.0** | 89.4 | 57.1 | **91.0** | **100.0** | 96.3 | 80.7 | 27.7 | 97.4 | 97.9 | 86.0 | 74.2 | **98.4** | **98.6** | 26.6 | 90.4 | 82.3 |

domain model SPEAR$_a$, as evidenced by tasks such as CD, SF-5, VI, and VL. We also observe a sharp increase for MST, a music transcription task, from SPEAR$_a$ Large with 23.6 to SPEAR$_{s+a}$ Large with 27.7. This suggests that the joint pre-training on speech and audio data enhances the model's capability of performing fine-grained music tasks. Despite achieving a better overall score on HEAR, we do notice that the dual-domain model suffers from performance degradation in some environment and music-related tasks. This further motivates us to use a more balanced dataset containing more general audio data in future work.

Finally, our largest dual-domain model SPEAR$_{s+a}$ XLarge achieves further performance improvement over SPEAR$_{s+a}$ Large, demonstrating that scaling data and model size is effective for SPEAR.

# G ABLATION STUDIES

Ablation studies on the following components are performed to provide an in-depth understanding of SPEAR framework:

- **Teacher Model Selection**: We compared pre-training with MVQ tokens extracted from different SSL teacher models (see Appendix G.1.1) and different layers of teacher models (see Appendix G.1.2).

- **Masked Prediction Pre-training Loss**: We investigated how to balance the pre-training loss on the masked and unmasked positions for SPEAR (see Appendix G.2).

- **Number of Codebooks**: We studied the effect of varying the number of codebooks in the MVQ quantiser on the pre-training performance (see Appendix G.3).

- **Feature Subspaces**: We compared the feature subspaces reconstructed by the MVQ quantiser and a k-means clustering model (see Appendix G.4).

- **MVQ Tokens vs k-means Tokens**: We compared using fine-grained MVQ tokens and k-means tokens as pre-training targets (see Appendix G.5).

- **Encoder Architectures**: We compared using Zipformer and Transformer as the encoder backbone for SPEAR (see Appendix G.6).

- **Dual-domain Pre-training**: We investigated how the losses from speech and audio domains should be balanced in the dual-domain pre-training (see Appendix G.7).

## G.1 Teacher Models and Layers

### G.1.1 Teacher Model Selection

In this group of ablation studies, different choices of SSL models are compared for generating pre-training targets under the SPEAR framework. In addition to WavLM Large and Dasheng 1.2B as presented in Table 3, HuBERT-Large (Hsu et al., 2021) and ATST-frame (Li et al., 2024a), which are pre-trained with different SSL objectives and data scales, are used for generating fine-grained discrete targets for speech and audio data, respectively.

Table 16: Performance on SUPERB benchmark of speech-domain models trained with MVQ tokens extracted from different models. Best results in **bold**, 2nd best results are underlined.

| Model | # Params | Targets | Pre-train Data | SUPERB | | | | | | |
|---|---|---|---|---|---|---|---|---|---|---|
| | | | | ASR $\downarrow$ | KS $\uparrow$ | IC $\uparrow$ | ASV $\downarrow$ | SID $\uparrow$ | SD $\downarrow$ | ER $\uparrow$ |
| ***Speech SSL Models*** | | | | | | | | | | |
| HuBERT Large | 317M | - | 60k | 3.62 | 95.29 | 98.76 | 5.98 | 90.33 | 5.75 | 67.62 |
| WavLM Large | 317M | - | 94k | 3.44 | 97.86 | 99.31 | 3.77 | **95.49** | 3.24 | 70.62 |
| ***Ours*** | | | | | | | | | | |
| Large-H-1 | 327M | HuBERT Large | 50k | 3.24 | 97.05 | 99.47 | 4.11 | 91.52 | 3.84 | 69.86 |
| Large-H-2 | 327M | HuBERT Large | 84k | **3.17** | 97.79 | **99.51** | 3.49 | 94.42 | 3.24 | 70.91 |
| SPEAR$_s$ Large | 327M | WavLM Large | 84k | 3.27 | **97.89** | 99.47 | **3.14** | 95.49 | **3.20** | **71.88** |

**HuBERT** HuBERT (Hsu et al., 2021) is a speech SSL model pre-trained using masked language modelling (MLM) loss on 60k hours of speech from Libri-light Kahn et al. (2020). In contrast to WavLM Large, HuBERT-Large is pre-trained on less diverse data (only read speech) without augmentations, resulting in weaker overall performance, especially on speaker-related tasks. A comprehensive comparison of HuBERT Large and WavLM Large on SUPERB can be found in Table 12 and Table 13.

Following Table 3, the representation from the 21st layer of HuBERT Large is used to train an MVQ quantiser with 16 codebooks for generating the pre-training targets. We train the following two Large models with pre-training targets generated from the HuBERT Large:

- Large-H-1: Pre-trained on Libriheavy **without** data augmentation. This model is used to contrast with HuBERT Large.

- Large-H-2: Pre-trained on Speech-84k using the same data augmentation as SPEAR$_s$ Large, enabling a fair comparison with SPEAR$_s$ Large and WavLM Large.

The pre-training performance is evaluated on SUPERB, and the results are shown in Table 16. As can be seen, MVQ tokens extracted from HuBERT Large are also effective pre-training targets. Large-H-1 outperforms its teacher HuBERT Large on all SUPERB tasks by a large margin on the premise of using the same amount of pre-training data. Notably, Large-H-1 demonstrates strong ASR performance, achieving the lowest WER on SUPERB ASR task, even surpassing SPEAR$_s$ Large trained with more data, suggesting that the MVQ tokens extracted from HuBERT Large could have a stronger focus on semantic information (Mousavi et al., 2025).

By increasing the amount of pre-training data, Large-H-2 further improves over Large-H-1. However, Large-H-2 yields a weaker overall performance on SUPERB compared to SPEAR$_s$ Large, which is pre-trained with MVQ tokens extracted from WavLM Large. This suggests that MVQ tokens extracted from stronger speech representations translate to a stronger per-training performance under SPEAR framework.

**ATST-frame** ATST-frame (Li et al., 2024a) is an audio SSL model pre-trained with BYOL (Grill et al., 2020) objective on 5k hours of AS-2M with 86M parameters. The model generates 768-d frame-level audio representations at a 25Hz frame rate. We train the following two Base models for comparison:

- Base-audio-1: Pre-trained on AS-2M with MVQ tokens extracted from the last layer of ATST-frame using 8 codebooks.

- BASE-AUDIO-2: Pre-trained on AS-2M with MVQ tokens extracted from the last layer of Dasheng 1.2B using 8 codebooks.

The results of AT fine-tuning on AudioSet and HEAR benchmark are presented in Table 17.

Table 17: Performance of audio-domain models pre-trained with MVQ tokens extracted from different teacher models on AudioSet AT tasks and HEAR. All results are the higher the better. *: For fair comparison with ATST-frame, HEAR evaluation is performed using the concatenation of all layers' representations. Best results in **bold**.

| Model | # Params | Targets | Pre-train Data | AudioSet | | HEAR* | | | | | |
|---|---|---|---|---|---|---|---|---|---|---|---|
| | | | | AS-20k | AS-2M | ESC | FSD | GZ-Gen | NS-5 | LC | SC-5 |
| *Audio SSL Models* | | | | | | | | | | | |
| EAT (Chen et al., 2024) | 86M | - | 5k | 40.2 | 48.6 | - | - | - | - | - | - |
| ATST-frame | 88M | - | 5k | 39.0 | 48.0 | 89.0 | 55.7 | 88.3 | 68.6 | 78.1 | 92.6 |
| *Ours* | | | | | | | | | | | |
| BASE-AUDIO-1 | 94M | ATST-frame | 5k | **40.3** | **49.6** | **89.4** | **57.2** | 89.5 | 64.4 | 79.4 | **94.3** |
| BASE-AUDIO-2 | 94M | Dasheng 1.2B | 5k | 39.1 | 49.3 | 88.9 | 56.6 | **90.1** | **72.2** | **81.2** | **94.3** |

As can be seen in Table 17, BASE-AUDIO-1 exhibits very strong performance on AudioSet AT tasks, achieving higher mAP than its teacher ATST-frame. By yielding an mAP of 40.3 on AS-20k and 49.6 on AS-2M, BASE-AUDIO-1 outperforms EAT (Chen et al., 2024), setting a new state-of-the-art for audio SSL models pre-trained only on AS-2M. This validates the effectiveness of SPEAR as an audio SSL approach, as it shows that the student model can consistently outperform its teacher model used for generating the pre-training targets.

However, despite achieving a higher mAP on AudioSet, BASE-AUDIO-1 shows weaker generalisation capability than BASE-AUDIO-2, the model pre-trained with MVQ tokens from Dasheng-1.2B. This is shown by its lower performance on HEAR tasks from the speech and music domains (e.g., GZ-Gen, NS-5, LC, and SC-5). We attribute this to the fact that the Dasheng 1.2B model produces more generic audio representations due to the vast amount of pre-training data and enormous model size, and this quality is encapsulated in the MVQ tokens derived from the MVQ quantiser. This suggests that the choice of teacher model plays a critical role in our framework's pre-training quality, as high-quality, generic features can be transferred to the student via the MVQ tokens, even when the student is trained on significantly less data.

**Conclusion** From Table 16 and Table 17, we conclude that the performance of SPEAR depends on the choices of teacher models for generating the pre-training targets. Under both speech-domain and audio-domain experiments, using a more powerful and generic teacher for pre-training targets generation leads to a better student model, indicating the necessity of using better teacher models for optimal performance. We also show that the performance of SPEAR framework is not upper-bounded by the teacher model, as our student models are always capable of outperforming their corresponding teacher model for generating the pre-training targets.

### G.1.2 TEACHER LAYER SELECTION

In this ablation study, experiments are carried out to compare different teacher layers for extracting pre-training targets. In our initial setuo, the teacher layer is selected based on the downstream fine-tuning performance, i.e. ASR for speech teacher and AT for audio teacher. All experiments are conducted using the Base architecture and the results are shown in Table 18.

For the speech teacher, we evaluate the 18th, 21st, and 24th (last) layers of WavLM Large and report the WERs on LS-100 fine-tuning tasks. The 21st layer is selected since it achieves the lowest WERs. Note that this choice aligns with the findings of Shi et al. (2024b), who also report the discrete tokens derived from the 21st layer of WavLM Large to yield strong ASR performance. For the audio teacher, we compare using the 25th, 35th, and 40th (last) layers of Dasheng-1.2B and report the mAP on the AS-20k fine-tuning task. The 40th layer is selected since it yields the highest mAP.

Table 18: Layer selection for speech teacher (WavLM) and audio teacher (Dasheng).

(a) WavLM layer comparison on LS-100 (WER).

| WavLM layer | test-clean ↓ | test-other ↓ |
|---|---|---|
| 18 | 3.1 | 6.1 |
| 21 (adopted) | **3.0** | **5.8** |
| 24 | 3.1 | 6.0 |

(b) Dasheng layer comparison on AS-20k (mAP).

| Dasheng layer | mAP ↑ |
|---|---|
| 25 | 38.4 |
| 35 | 38.7 |
| 40 (adopted) | **39.2** |

## G.2 PRE-TRAINING LOSS

The hyperparameter $\alpha$ in Equation 2 controls the contribution of the prediction loss on the masked and unmasked frames. To investigate its influence w.r.t SPEAR, we conducted experiments on single-domain models, varying $\alpha$ from 0.0 (predicting only unmasked frames) to 1.0 (predicting only masked frames). The speech-domain models are pre-trained on LibriSpeech, while the audio models are pre-trained on AS-2M. The same MVQ tokens from Table 3 are used. We evaluate the downstream fine-tuning performance on LS-100 fine-tuning and AT, results presented in Figure 2.

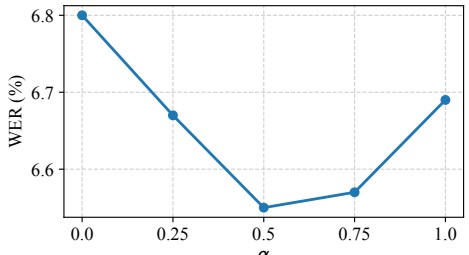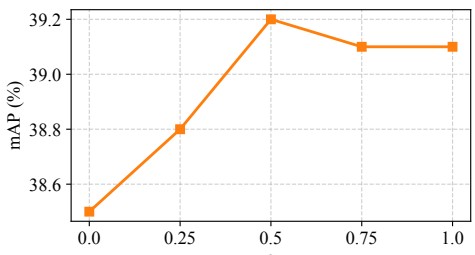

Figure 2: Effect of $\alpha$ on two downstream fine-tuning tasks. Left: WERs of test-other on LS-100 ASR fine-tuning task; Right: mAP on AudioSet evaluation set on AS-balanced fine-tuning task.

As shown in Figure 2, a balanced contribution of prediction loss on masked and unmasked frames with $\alpha = 0.5$ yields the best downstream performance. This observation diverges from the findings in HuBERT (Hsu et al., 2021), where computing prediction loss merely on masked frames (i.e. $\alpha = 1.0$) was optimal. We hypothesise this difference stems from the fine-grained nature of the MVQ tokens. Compared to the coarse units generated from k-means clustering, predicting fine-grained MVQ tokens is a significantly more challenging pretext task. Including the easier objective of predicting tokens at unmasked positions helps to regularize the model and stabilize the learning process. However, the pretext task must remain sufficiently challenging: setting $\alpha = 0.0$ makes the objective too simple, degrading it to a non-contextual prediction task that is ineffective for learning powerful representations. Thus, a balanced $\alpha$ is crucial for the success of the SPEAR framework.

## G.3 NUMBER OF CODEBOOKS

The relationship between the number of codebooks $N$ and the pre-training performance is investigated here. Experiments are carried out under single-domain settings with $N$ varying from 4 to 16, using the Base size model.

For speech-domain experiments, the same representations from the 21st layer of WavLM Large are used to train the MVQ quantiser. We pre-train the models on LS-960 for 300k updates. The performances on the following three tasks are evaluated: ASR fine-tuning on LS-100, speaker identification (SID), and emotion recognition (ER) from SUPERB, which serve as indicators of the model's understanding and paralinguistic capabilities. The results are shown in Table 19. As can be seen, increasing the number of codebooks for speech pre-training consistently enhances the model performance. The WER on the test-other set is reduced by 6.4% with $N$ increasing from 4 to 16. Moreover, models trained with a larger $N$ also exhibit stronger paralinguistic capabilities, which are

Table 19: Results of SPEAR speech-domain pre-training with different numbers of codebooks $N$. Best results in **bold**.

| $N$ | LS-100 | | SUPERB | |
|---|---|---|---|---|
| | test-clean↓ | test-other↓ | SID↑ | ER↑ |
| 4 | 3.34 | 7.01 | 83.12 | 67.24 |
| 8 | 3.19 | 6.77 | 84.83 | 67.76 |
| 16 | **3.08** | **6.55** | **86.35** | **68.29** |

Table 20: Results of audio-domain pre-training with different numbers of codebooks $N$. All results are the higher the better. Best results in **bold**.

| $N$ | AudioSet | | HEAR | | | |
|---|---|---|---|---|---|---|
| | AS-20k | AS-2M | Environment | Speech | Music | Average |
| 4 | **39.2** | 49.1 | 77.63 | 68.09 | 80.30 | 75.63 |
| 8 | **39.2** | **49.3** | **80.33** | 69.87 | **80.70** | **77.01** |
| 16 | 38.9 | 49.0 | 80.25 | **69.92** | 80.64 | 76.97 |

manifested through their performance on SID and ER. This implies that increasing $N$ to 32 could lead to further performance improvement for speech-domain models.

Similar experiments are conducted for audio-domain pre-training. Following Table 3, the last layer of Dasheng 1.2B is used to train the MVQ quantiser with 4, 8, and 16 codebooks. The models are evaluated on the AudioSet fine-tuning task, and the results are shown in Table 20. As can be seen, increasing $N$ from 4 to 8 improves the downstream AT fine-tuning performance and the HEAR scores. However, further increasing $N$ to 16 degrades the pre-training performance, leading to a lower mAP and average HEAR score compared to $N = 8$. We hypothesise that representations of audio SSL models encapsulate less information compared to speech representations in general. Therefore, using a moderate number of codebooks seems to be enough for audio-domain pre-training. A too large $N$ might force some codebooks to capture the nuances in the audio teacher representations and introduce noise to the pre-training.

### G.4 FEATURE SUBSPACES OF MVQ TOKENS

In order to investigate if the codebooks in the MVQ quantiser have captured useful characteristics from the speech and audio representations, we visualise the reconstructed embedding space of the MVQ quantiser on a 2-D plane using UMAP (McInnes et al., 2018). Specifically, we visualise the speaker embeddings encoded by the MVQ quantiser of 10 speakers randomly drawn from LibriSpeech dev-clean sets, with each speaker having 25 utterances. The speech MVQ quantiser from Table 3 is used. The speaker embeddings are computed with the procedure described below. First, we use the speech MVQ quantiser (see Table 3) to encode the frame-level embeddings generated by WavLM Large into the MVQ tokens. Then, we compute the reconstructed frame-level embeddings by summing over the encoded code vector from each codebook. The speaker embedding for each utterance is obtained by calculating the mean embedding vector over all frames. As a comparison, we also visualize the speaker embedding space represented through a k-means clustering model. The 500-cluster k-means model used for generating the pre-training targets for WavLM Large is adopted, which is trained on the 9th layer representations of HuBERT Base. We use the cluster centroid to represent each frame and average the frame-level embeddings along the temporal dimension to obtain a single speaker embedding for each utterance.

The visualisations are shown in Figure 3c. As can be seen, the MVQ quantiser successfully retains speaker characteristics, showing a clear separation between different speakers. It is noteworthy that a single codebook with 256 codes is capable of capturing certain levels of the speaker characteristics, showing reasonable separation between different speakers in Figure 3b. However, the k-means centroids fail to distinguish different speakers, showing poor separation for different speakers. This observation aligns with the fact that our speech-domain model SPEAR$_S$ Large achieves far better

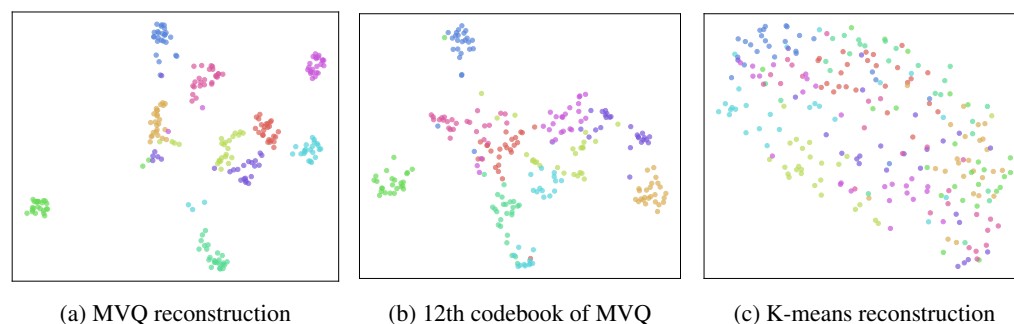

|            |            |            |
|:----------:|:----------:|:----------:|
| (a) MVQ reconstruction | (b) 12th codebook of MVQ | (c) K-means reconstruction |

Figure 3: Comparing the reconstructed embedding space obtained through different MVQ quantisation and k-means. The speaker embeddings of 10 speakers drawn from LibriSpeech dev-clean are visualized using UMAP on a 2D plane, with each colour representing a single speaker. (a): Reconstruction using all codebooks of MVQ; (b): Reconstruction using only the 12th codebook of MVQ; (c): Reconstruction using k-means centroids.

performance on ASV compared to WavLM Large, a task requiring distinguishing different speakers by comparing the speaker embedding similarity.

## G.5 MVQ TOKENS AND K-MEANS TOKENS

To isolate the impact of the quantization target, we conduct a controlled ablation comparing our MVQ tokens against standard k-means tokens. Both target types are derived from the same layer (the 21st layer) of WavLM Large, with the k-means baseline utilising 2000 clusters. Using the Base architecture, both models are pre-trained for 300k updates on LibriSpeech under identical augmentation strategies (noise mixing, utterance mixing, and masking). Notably, for the k-means experiment, we adopt the standard configuration by computing the loss solely on masked positions (Hsu et al., 2021). The results on the LS-100 ASR fine-tuning task and the SUPERB benchmark are shown in Table 21.

Table 21: Comparison of MVQ tokens and k-means as pre-training target.

| Target | LibriSpeech finetune | | SUPERB | | | | |
|--------|:---------:|:---------:|:------:|:------:|:------:|:------:|:------:|
|        | test-clean | test-other | PR $\downarrow$ | IC $\uparrow$ | KWS $\uparrow$ | SID $\uparrow$ | ER $\uparrow$ |
| k-means, 2000 clusters | 3.5 | 7.2 | 4.0 | 97.92 | 96.79 | 86.6 | 67.56 |
| MVQ, 16 codebooks | **3.1** | **7.0** | **3.4** | **98.37** | **96.83** | **88.4** | **68.29** |

It can be seen that the model trained with MVQ tokens achieves better performance on all tasks, especially the two paralinguistic tasks: SID and ER. This aligns with our findings in Appendix G.4, where it is shown that the fine-grained MVQ tokens retain richer paralinguistic information than the coarse k-means tokens.

## G.6 ENCODER ARCHITECTURE

This ablation study investigates the effect of different encoder architectures. Specifically, the Zipformer architecture (with filterbank input) is compared with the Transformer model (with waveform input), a commonly used architecture for SSL models (Baevski et al., 2020; Chen et al., 2022) in the speech domain. The transformer implementation follows the WavLM, which is an improved version of wav2vec 2.0. We pre-trained the two models using the same MVQ pre-training loss under the SPEAR framework on LS-960 data for 300k updates. The results on the LS-100 ASR fine-tuning task and the SUPERB benchmark are reported in Table 22.

To decouple the influence of the encoder architecture from our pre-training framework, this ablation compares the Zipformer (utilising filterbank inputs) against a standard Transformer backbone (utilising waveform inputs). The Transformer implementation adopts the architecture adopted in

WavLM (Chen et al., 2022). We pre-train both models on the full LibriSpeech 960h corpus (LS-960) for 300k updates, maintaining the same MVQ-based pre-training objective. The results on LS-100 ASR fine-tuning and the SUPERB benchmark are presented in Table 22.

Table 22: Comparison of Zipformer and Transformer as encoder backbone.

| Encoder Backbone | LibriSpeech finetune | | SUPERB | | | | |
|---|---|---|---|---|---|---|---|
| | test-clean | test-other | PR $\downarrow$ | IC $\uparrow$ | KWS $\uparrow$ | SID $\uparrow$ | ER $\uparrow$ |
| Transformer | 4.1 | 9.0 | **3.4** | 97.79 | **97.27** | **89.8** | 66.27 |
| Zipformer | **3.1** | **7.0** | **3.4** | **98.37** | 96.83 | 88.4 | **68.29** |

On the SUPERB tasks, which evaluate frozen representations, both architectures exhibit comparable performance: the Transformer proves stronger on SID, whilst the Zipformer excels on ER. However, the Zipformer demonstrates a distinct advantage in downstream ASR fine-tuning, a result consistent with its ASR-centric design (Yao et al., 2024). Moreover, the Zipformer is more computationally efficient due to its intermediate downsampling operations. For instance, pre-training the Zipformer requires approximately 350 GPU hours, roughly 60% of the 600 hours required for the Transformer. Consequently, the stronger ASR fine-tuning performance and computational efficiency motivate our selection of the Zipformer as the backbone architecture for SPEAR.

### G.7 DUAL-DOMAIN PRE-TRAINING

#### G.7.1 ASYMMETRICAL PRE-TRAINING LOSS

As mentioned in Section 3.2.2, we adopt an asymmetrical pre-training loss in Equation 6 for dual-domain pre-training. The following strategies for dual-domain pre-training are investigated:

- JOINT: Each training input $w$ induces two losses, computed against the $Z^s$ and $Z^a$, regardless of the domain of $w$.
- DISJOINT: Each training input $w$ only induces one loss, computed against the targets generated by the teacher from the same domain as $w$.
- ASYMMETRICAL: For speech data, losses are computed against both $Z^s$ and $Z^a$. For audio data, loss is only computed against $Z^a$. This approach is adopted by SPEAR.

Table 23: Results of three strategies for dual-domain pre-training. Best results in **bold**.

| Strategy | LS-100 | | AS-20K $\uparrow$ | SUPERB | | HEAR |
|---|---|---|---|---|---|---|
| | test-clean $\downarrow$ | test-other $\downarrow$ | | SID $\uparrow$ | PR $\downarrow$ | Avg $\uparrow$ |
| JOINT | **2.9** | 5.9 | 36.8 | 90.6 | 3.24 | 78.7 |
| DISJOINT | 3.0 | **5.8** | **37.0** | 87.4 | 3.40 | 78.3 |
| ASYMMETRICAL | **2.9** | **5.8** | 36.9 | **90.7** | **3.12** | **79.0** |

We perform dual-domain pre-training using the Large size model on the Mix-97k data with the aforementioned three strategies. The models are evaluated after 100k training steps for quicker system verification, where we compare the results of LS-100 ASR fine-tuning, AS-20K AT fine-tuning, two SUPERB tasks (SID and PR), and the average score on HEAR. The results are shown in Table 23. Among the three strategies, ASYMMETRICAL achieves a balanced performance across both domains. Computing the loss against the speech MVQ tokens for audio data is a useful regularisation to bridge the domain mismatch between speech and audio, preventing significant performance degradation on both domains (ASYMMETRICAL vs DISJOINT).

On the other hand, computing pre-training loss against the audio MVQ tokens for speech data in the JOINT strategy is less useful. Compared to the ASYMMETRICAL strategy, an increase of 0.1 absolute WER and a 0.3 absolute lower average score on HEAR are observed. We suspect that this is caused by the imbalanced data distribution between speech and audio in Mix-97k, where speech data makes up 87% of the total data. The dominance of the speech data hinders the effective learning

of generic audio representations for the JOINT strategy. This motivates us to enlarge the proportion of general audio data in the total training corpora for our future work.

### G.7.2 LOSS WEIGHTING

As shown in Equation 6, the hyperparameter $\lambda$ controls the contribution of the general-audio masked-prediction loss during joint training. To determine the optimal balance, we conduct an ablation study with 3 values of $\lambda$ using our SPEAR Large architecture and compare the fine-tuning performance on LS-100 and AS-20k after 100k pre-training steps. The experimental results are shown in Table 24. We observed that reducing $\lambda$ from 0.3 to 0.1 yields a 0.3 absolute WER improvement on test-other, while the mAP is only reduced by 0.1 absolute. Consequently, we adopted $\lambda$=0.1 in our dual-domain experiments for a balanced performance across both domains.

Table 24: Effect of $\lambda$ in dual-domain pre-training.

| $\lambda$ | LS-100 | | AS-20k |
|---|---|---|---|
| | test-clean ↓ | test-other ↓ | mAP ↑ |
| 0.3 | 3.0 | 5.9 | **37.0** |
| 0.2 | 3.0 | 5.7 | 36.9 |
| 0.1 | **2.9** | **5.6** | 36.9 |

## H  COMPARISON WITH USAD

In this section, we performed a controlled comparison between SPEAR and USAD (Chang et al., 2025), another framework for joint speech and audio representation learning also leveraging multiple domain-specific teachers. Specifically, we trained a new dual-domain SPEAR model with the Base architecture, named SPEAR (USAD-aligned), mirroring the USAD settings to isolate the impact of the pre-training objective. We used the same teacher models as used in USAD, namely WavLM Base+ (speech) and ATST-Frame (Li et al., 2024a) (audio) to extract the MVQ tokens as pre-training targets in SPEAR. We also used a subset of the USAD training corpora, excluding Fisher and VoxLingua for speech and SoundNet for audio due to availability issues, a summary of the model configurations and data usage for both models is shown in Table 25

Table 25: Model configurations of SPEAR (USAD-aligned) and USAD. Approximate data amount in hours.

| Model | # Params | Speech data | Audio data | Total data |
|---|---|---|---|---|
| SPEAR (USAD-aligned) | 94M | 86k | 9.3k | 95.3k |
| USAD Base | 94M | 91k | 35k | 126k |

The comparison between SPEAR (USAD aligned) and USAD on SUPERB and HEAR is shown in Table 26. As can be seen, SPEAR (USAD-aligned) consistently outperforms USAD Base, despite only using a subset of the training data used by USAD, suggesting that SPEAR is more effective than USAD for learning unified speech and audio representations. We attribute this performance gap to the following two reasons:

- **Training objectives**: In SPEAR, the student is trained to predict the discrete tokens extracted from teacher models given a masked input, which is a frequently used pretext task for SSL. This combination of KD and SSL in SPEAR enables the student to learn generic representations while benefiting from the knowledge of the two domain-specific teachers, creating student model even with the capability of surpassing teacher models (e.g. SPEAR$_s$ Large outperforms WavLM Large). On the other hand, USAD enforces the student to mimic the teacher representations through L1 and cosine distance loss. Consequently, the student performance is theoretically upper-bounded by the teacher performance.
- **Joint feature matching is ill-defined for disparate domains**: In USAD, the student effectively minimises the distance to two embedding spaces (speech and audio) simultaneously.

The L1 losses induced by two teachers encourage the student model to find a "mean" of two representation spaces. Since the feature spaces could be distinct, this "mean" representation may lie in a region of the manifold that lacks semantic meaning for either domain. SPEAR avoids this risk by quantising representations into discrete tokens via MVQ, where the tokens exhibit the capability of representing a certain characteristic of the input speech/audio data (see Appendix G.4 where we found a single codebook to contain rich speaker information). This allows the model to retain distinct, high-fidelity details by learning to predict the discrete tokens for both domains simultaneously, with lower risk of destructive interference.

Table 26: Comparison between SPEAR (USAD-aligned) and USAD on SUPERB and HEAR. HEAR results for both SPEAR (USAD-aligned) and USAD Base are obtained with feature concatenation.

| Model | # Params | Data | SUPERB | | | | | | HEAR | | | |
|-------|----------|------|--------|------|------|------|-------|------|-------|---------|--------|-------|
| | | | PR ↓ | ASR ↓ | IC ↑ | KS ↑ | SID ↑ | ER ↑ | Env ↑ | Speech ↑ | Music ↑ | Avg ↑ |
| SPEAR (USAD-aligned) | 94M | 95.3k | **4.6** | **5.1** | **98.7** | **97.4** | **89.2** | **69.4** | **81.1** | **76.9** | **80.5** | **79.4** |
| USAD Base | 94M | 126k | 5.1 | 7.7 | 98.3 | 97.1 | 88.6 | 68.0 | 80.7 | 73.7 | 79.3 | 77.8 |

## I  COMPARISON WITH DASHENG

As discussed in Section 5.3, the performance of $\text{SPEAR}_a$ models lags behind Dasheng on the HEAR benchmark, mainly due to the large difference in the amount of general audio training data (smaller by a factor of 20). In this section, to validate the effectiveness of SPEAR for audio SSL under a more comparable setup, we compare SPEAR and Dasheng under the constraint of using the same pre-training dataset. Specifically, both models are pre-trained on AudioSet (5k hours) and their performances on HEAR (average score) are reported in Table 27. The results of the Dasheng models are taken from the original paper (Dinkel et al., 2024), while the results of $\text{SPEAR}_a$ models are taken from Table 6. As can be seen, $\text{SPEAR}_a$ Large achieves 78.08 on HEAR, 3.21 points higher than the 4 times bigger Dasheng 1.2B model pre-trained on the same data. Although $\text{SPEAR}_a$ Large uses the original Dasheng 1.2B pre-trained on larger amount of data to extract the MVQ tokens for SSL pre-training, this significant gap between $\text{SPEAR}_a$ Large and Dasheng 1.2B in Table 27 still suggests that our SPEAR framework is highly effective for audio SSL.

Table 27: Performance comparison between Dasheng and SPEAR on HEAR benchmark (average score). All models are pre-trained on AudioSet with 5k hours data. HEAR results for Dasheng from Dinkel et al. (2024)

| Model | Pre-train Data | # Params | HEAR score ↑ |
|-------|----------------|----------|--------------|
| Dasheng Base | AudioSet 5k | 86M | 70.43 |
| Dasheng 1.2B | AudioSet 5k | 1.2B | 74.87 |
| $\text{SPEAR}_a$ Base | AudioSet 5k | 94M | 76.37 |
| $\text{SPEAR}_a$ Large | AudioSet 5k | 327M | 78.08 |

