# OpenReview forum: "SPEAR: A Unified SSL Framework for Learning Speech and Audio Representations"
_ICLR.cc/2026/Conference — Submitted to ICLR 2026_

### Official Review · Reviewer_y6wJ · 2025-10-24

**Soundness:** 2
**Presentation:** 3
**Contribution:** 1
**Rating:** 2
**Confidence:** 5

**Summary:**

## An attempt to unify speech and audio SSL representations
* This paper proposes SPEAR, which is essentially a model for audio and speech representation learning trained through knowledge distillation from MVQ quantised targets generated from existing Speech (WavLM) and Audio (Dasheng) SSL models via a joint masked prediction objective.
* Three distinct configurations of SPEAR have been evaluated: audio-only, speech-only, and joint audio-speech settings, respectively, across the HEAR and the SUPERB benchmark suites.

**Strengths:**

* The paper is an excellent read: it's easy to follow and understand, and experiments across the main benchmark suites spanning speech and audio domains are commendable.
* The neural architecture employed is straightforward, and as someone who finds great strength in simplified architectures, it's a joy to see a straightforward approach pan out relatively well.

However, the paper has several weaknesses which are grounds for the given score.

**Weaknesses:**

## Point 1

The paper is not sufficiently well-motivated, in my opinion. Why do we need a unified front against speech and audio, and is masked discretised token prediction actually the correct way to go about it? Maybe the community has not tried to learn a unified SSL approach through prediction of discrete tokens because discretising fine-grained acoustic content (audio) and semantic/phonetic content (speech) in a single representation is tremendously difficult and always presents a performance tradeoff across the two aspects, as demonstrated by the large swath of audio tokenisation papers trying to crack this problem.

## Point 2

* The approach lacks novelty. It is essentially a model trained by masked prediction of quantised targets obtained from existing models (in this case, SSL approaches). EncodecMAE [1] learns an audio-only SSL representation in an extremely similar fashion, using Encodec [2] to generate the quantised targets instead, and it outperforms BEATs too. But there is no mention of EncodecMAE in the submitted paper.

* Even if we let the above omission slide, the proposed approach is simply not novel enough for an ICLR paper. The novelty is entirely in the application. MVQ, the two teacher models, distillation on quantised targets, and model architecture: they have all been done before. The dual objective framework is also novel only in application to audio and speech SSL unification.

## Point 3

Talking about BEATs, the submitted paper states that BEATs has substantial training complexity. That is true, BEATs has a 3-stage training process and indeed has high training costs. However, so does the proposed approach, albeit indirectly. The prerequisite for SPEAR is not one, but two frontier pretrained SSL models, and SPEAR needs to be trained again on top of it all on very large-scale datasets. On the other hand, BEATs does not require an existing large-scale SSL model; it requires iterative refinement because the tokenizer is, well, iteratively refined from cold-start. The commentary on simplicity and training complexity within the context of this paper is moot.

## Point 4

* Results for audio tasks (HEAR, Table 6) are not very encouraging.
* The paper states that "Finally, our dual-domain models consistently outperform their single-domain counterparts, highlighting the benefits of our unified pre-training framework". However, the majority of the (relatively small in proportion to the increase in pretraining dataset size) performance improvement in SPEAR-(a+s) over SPEAR-(a), for both Base and Large models, stems from improved speech performance, which is not surprising.
* Further, despite being distilled from Dasheng 1.2B, the base and large SPEAR-(a) and SPEAR-(a+s) models trail behind Dasheng-Base, the smallest Dasheng model, by quite a bit, despite having more parameters. The only case where SPEAR does better than Dasheng models is when using feature concatenation for the XLarge configuration. The presented analysis showcases no benefit to training a complex SPEAR Audio+Speech model for audio SSL.
* These observations tie in with Point 1. If the unification of audio and speech fronts was a significant novelty, and was an SSL approach pretrained from scratch, one could argue that future research will close the gap. But, given the worse audio representation learning performance, the merits of the SPEAR approach over the existing publicly available pretrained Dasheng models are not clear.

### Point 5

- All the models presented in Table 6 are trained on different amounts of data. This only obfuscates results. BEATs and ATST-Frame are both trained only on 5k hours of audio data. SPEAR models performing worse than Dasheng models could also be written-off by the size of the pretraining data used by Dasheng, but we just can't say based on the presented analysis.


## Overall

My recommendation to reject is a reflection of the above-mentioned shortcomings.

[1] https://www.isca-archive.org/interspeech_2025/pepino25_interspeech.html
[2] https://openreview.net/forum?id=ivCd8z8zR2

**Questions:**

No direct questions, kindly address the points raised in the weaknesses section.

---

> ### Author Response · Authors · 2025-11-20
> **Response to Reviewer y6wJ (1/6)**
>
> We thank the reviewer for the detailed feedback and for acknowledging that the paper is "an excellent read" and "easy to follow." We appreciate the opportunity to clarify our work's motivation, novelty, and experimental results, which we believe directly addresses the reviewer's concerns.
>
> ---
>
> ## Additional Experiments
>
> Before we address the questions raised, we would like to first present the results of two additional experiments, which we believe could better support our arguments. Specifically, we performed more evaluation on the audio-SSL Dasheng model, which is used as the audio teacher models in SPEAR:
>
> ---
>
> ### [EXP1] Dasheng Evaluation on SUPERB
>
> To assess the speech capability of Dasheng, we have evaluated Dasheng on SUPERB and give the results in the following table.
>
> | Model | # Params | Pre-train Hours | PR $\downarrow$ | ASR $\downarrow$ | IC $\uparrow$ | KS $\uparrow$ | SID $\uparrow$ | ER $\uparrow$ |
> | ---- | --- | --- | --- | --- | --- | --- | --- | ---|
> | Dasheng-1.2B | 1.2B | 272k | 13.8  | 14.3 | 98.13 | 97.73 | 92.4 | 68.7 |
> | SPEAR_s Large | 327M | 84k | **2.6** | 3.3 | 99.47 | 97.89 | 95.5 | 72.1 |
> | SPEAR_{s+a} XLarge | 600M | 197k |  2.9 | **3.2** | **99.61** | **98.12** | **96.3** | **73.3** |
>
> It can be seen that Dasheng-1.2B performs much worse than SPEAR_{s+a} XLarge. **This indicates that the features learned by audio-SSL models like Dasheng do not generalise well to speech processing tasks**, especially tasks requiring semantic/phonetic (ASR, PR) and paralinguistic information (SID, ER).
>
> ---
>
> ### [EXP2] Fine-tuning Dasheng for ASR
>
> Additionally, we performed an experiment to fine-tune Dasheng model on LibriSpeech data for the ASR task, with the same fine-tuning configuration as described in Section 5.1. The WERs on test-clean and test-other are reported in the Table below with LS-100 and LS-960 as fine-tuning sets. We see that **Dasheng-1.2B model yields much higher WERs comared to SPEAR_{s+a} XLarge**, despite its use of a massive pre-training corpora (272k hours) and a large model size (1.2B).
>
>
> | Model | # Params | Pre-train Hours | LS-100 $\downarrow$ | LS-960 $\downarrow$|
> | ------- | ------|----- | --- | ------ |
> | Dasheng-1.2B | 1.2B | 272k | 7.7/20.0 | 3.4/8.7 |
> | SPEAR_{s+a} XLarge | 600M |197k |  **2.5**/**4.6** | **1.6**/**2.9** |
>
> From the two experiments above, we can see that **the large-scale audio-SSL Dasheng model (1.2B, 272k hours) gives good performance in the general audio domain but shows weak generalisation capability to speech tasks** (especially tasks requiring high-level semantic understanding). In contrast, the dual-domain SPEAR_{s+a} models demonstrate strong and balanced capabilities across both domains as shown in our paper.
>
> *(The remaining comments will be addressed in subsequent responses).*

---

> ### Author Response · Authors · 2025-11-20
> **Response to Reviewer y6wJ (2/6)**
>
> ## [W1]: Motivation of SPEAR (Point 1)
>
> ### [W1.1]: Motivation for unified speech and audio model
> > The paper is not sufficiently well-motivated, in my opinion. Why do we need a unified front against speech and audio
>
> The drive towards foundation models trained on broad data that can be adapted to many tasks is one of the most significant trends in AI. This has been demonstrably successful in NLP (e.g. GPT3[1]) and Vision (e.g., the Segment Anything Model, SAM[2]). We believe that such a unified model should exist for the speech and audio community, which is not only an academic goal, but also a practical necessity for several reasons:
>
> - **Real-world speech and audio is often mixed**: Speech and general audio are often not isolated in real life scenarios. They occur together in nearly every natural human environment. For example, human conversations usually happen with an on-going background sound event. Therefore, a model must be able to understand speech while ignoring (or understanding) background sound events. Humans do this effortlessly and we believe a robust multi-modal AI system should do the same.
> - **Efficiency and versatility**: Developing, training, and deploying separate specialist models for each separate domain is inefficient. For example in multi-modal LLMs, such as SALMONN[3], the model requires explicitly two encoders: one for speech (Whisper) and one for audio (BEATs), to have a comprehensive audio perception capability. This increases the compexity of the system and is not scalable to an increasing number of separate specialist models. **Therefore, a single, powerful, and versatile model for both speech and general audio is highly desirable, which is exactly the motivation for SPEAR.**
>
> ---
>
> ### [W1.2]: Why use discretised token prediction
>
> > is masked discretised token prediction actually the correct way to go about it? ... discretising fine-grained acoustic content (audio) and semantic/phonetic content (speech) in a single representation is tremendously difficult ...
>
> We agree that fusing fine-grained acoustic and semantic content in a single representation space is challenging. This is exactly why we would like to focus on this discrepancy and propose SPEAR to bridge this gap between speech and audio.
>
> Since different types of discretised tokens have shown promising results in speech (e.g. k-means semantic/phonetic tokens) and audio (e.g. RVQ tokens), **we hypothesise that masked prediction of discretised token could be a unified pre-training task for speech and audio**. To further bridge the gap between token granularity difference in speech and audio, and keeping strong performance over both domains, **we propose to use fine-grained MVQ tokens extracted from expert models which retain rich acoustic and semantic information**. Our strong experimental results on both SUPERB and HEAR verify the viability of the proposed approach: our single-domain models trained under SPEAR framework achieves SOTA on SUPERB and approach the top SSL models trained with much more data on HEAR, while our dual-domain models maintain high performance across both domains.
>
> > ... discretising fine-grained acoustic content (audio) and semantic/phonetic content (speech) in a single representation is tremendously difficult and always presents a performance tradeoff across the two aspects
>
> Again, we agree that the fusion of speech and audio representations is difficult and usually presents a tradeoff. For example, despite of having a higher overall score on HEAR (79.26 vs 79.18), the dual-domain SPEAR_{s+a} Large performs less well than SPEAR_a on Env and Music tasks of HEAR (Table 6). However, we argue that this tradeoff is inevitable since the model is doing a much harder task (merging two representation spaces), and **this tradeoff is worthwhile since the dual-domain models can encode high quality representations for both domains.**
>
> We also find promising results where **the dual-domain model can benefit from joint learning over speech and audio data under some tasks**. For example:
>
> - SPEAR_{s+a} Large outperforms SPEAR_s Large on two SUPERB tasks:
>     - Speech Enhancement: 2.72 vs 2.71 (STOI)
>     - Keyword Spotting: 97.92 vs 97.89 (Accuracy)
> - SPEAR_{s+a} Large outperforms SPEAR_a Large on the following HEAR tasks:
>     - MST: 26.9 vs 23.6 (OnSet FMS)
>     - CD: 81.4 vs 79.8 (Accuracy)
>     - SC-5/SC-F: 97.5 vs 95.3 (Accuracy)
>     - VI: 22.6 vs 19.0 (mAP)
>     - VL: 83.9 vs 66.2 (Accuracy)
>
> We believe these improvements are a promising indicator of positive synergy induced by jointly modelling speech and audio. These findings could open-up a new research direction to further harness the benefits of joint speech audio modelling.
>
> [1]: Brown, Tom, et al. "Language models are few-shot learners." in Proc. NeuIPS 2020
>
> [2]: Kirillov, Alexander, et al. "Segment anything." in Proc. ICCV 2023.
>
> [3]: Tang, Changli, et al. "Salmonn: Towards generic hearing abilities for large language models." in Proc. ICLR 2024

---

> ### Author Response · Authors · 2025-11-20
> **Response to Reviewer y6wJ (3/6)**
>
> ## [W2] The novelty of SPEAR (Point 2)
>
> > EncodecMAE learns an audio-only SSL representation in an extremely similar fashion, using Encodec to generate the quantised targets instead, and it outperforms BEATs too. But there is no mention of EncodecMAE in the submitted paper.
>
> We thank the reviewer for mentioning EncodecMAE, which is an important work for learning audio-only representations through masked prediction of Encodec codes. We apologise for missing this work in our original manuscript and we will add this missing reference in the revised version of the paper in the related work section.
> We would note that the reviewer describes it as an "audio-only SSL representation". This reinforces our central motivation: a unified framework for both speech and audio was, and remains, a significant and open research challenge that our paper directly addresses.
>
> > The approach lacks novelty. ... The novelty is entirely in the application. MVQ, the two teacher models, distillation on quantised targets, and model architecture: they have all been done before
>
> Thank you for your comment. Indeed, SPEAR is built on existing technolgies, such as Multi-codebook Vector Quantisation (MVQ), and multi-modality knowledge distillation (KD).
> Despite of this, we would like to emphasise that **SPEAR is the first to propose a unified SSL objective to successfully learn unified speech and audio representations from a mixture of heterogeneous speech and general audio data.**
>
> Existing SSL approaches for speech or audio domain are highly domain specific. For instance, speech SOTA SSL models (e.g. WavLM) use masked prediction of coarse tokens, which are unsuitable for general audio (see Table 6 for WavLM results on HEAR). Audio SOTA SSL models (e.g. Dasheng) use objectives such as masked autoencoders (MAE), which has been shown to learn limited semantic information essential for speech tasks (see the additional experiment of Dasheng on SUPERB and ASR).
> **Our proposed SPEAR framework addresses this limitation by proposing a unified masked token prediction pretext task for data from both domains, which we believe is a key novel aspect of SPEAR.**
>
> **Another key novelty of SPEAR is the formulation of knowledge distillation (KD) as an SSL objective**. A key distinction between SPEAR and traditional KD is the objective design. Prior methods such as USAD use direct KD via feature mimicing to learn unified representations. In contrast, SPEAR formulates the problem as an **SSL objective** with a pretext task of masked prediction of fine-grained MVQ token. The application of MVQ for SSL is novel, since this algorithm was originally proposed for efficient KD. By leveraging high-quality teachers to generate discrete tokens, SPEAR models can even outperform their teachers without using extra data or a larger model size (e.g., SPEAR_S Large vs. WavLM Large in Table 5; SPEAR_a Base vs. ATST-frame in Table 17), which is difficult to achieve with traditional KD methods.
>
> We would also like to mention that the dual-domain pre-training design described in Section 3.2.2 is novel. By employing a asymmetrical loss over speech and audio, the model acheives a balanced performance over both domains, leading to an overall better performance. **No similar strategies have been proposed to better accommodate the data imbalance during pre-training on mixed speech and audio data.**
>
> Hence, we argue that SPEAR is a significant contribution, as it is the first to propose a unified SSL solution for speech and general audio, and provides a single, versatile model that excels in both speech and audio understanding. Such a model could be used as a unified speech+audio frontend for multi-modal LLMs by replacing existing speech-audio dual-encoder architectures[1].
>
> [1]: Tang, Changli, et al. "Salmonn: Towards generic hearing abilities for large language models." in Proc. ICLR 2024

---

> ### Author Response · Authors · 2025-11-20
> **Response to Reviewer y6wJ (4/6)**
>
> ## [W3] The complexity of SPEAR (Point 3)
>
> >  The prerequisite for SPEAR is not one, but two frontier pretrained SSL models, and SPEAR needs to be trained again on top of it all on very large-scale datasets.
>
> SPEAR relies on pre-training targets extracted from off-the-shelf pre-trained SSL expert models, which does induce extra complexity to the SPEAR framework.
> However, we would respectfully argue that it is this specific design that enables SPEAR to **harness the power of large-scale pre-trained domain-specific foundation models**, and even allows the creation of models that surpass the performance of their teacher models:
> - **Speech domain**: SPEAR_s Large (327M, 84k hours) matches or outperforms its teacher WavLM Large (327M, 94k hours) on **12 out of 15 tasks** on SUPERB (see Tables 12 and 13)
> - **Audio domain**: SPEAR Base (94M, 5k hours) outperforms its teacher ATST-frame (88M, 5k hours) in a strict fair comparison (see Appendix G.1 and Table 17)
>
> Furthermore, we believe this approach aligns with a modern, practical, and efficient research paradigm. Nowadays, more and more large-scale foundation models, trained with resources far beyond the reach of most academic and some industrial labs, are becoming publicly accessible. We believe that SPEAR's methodology provides a valuable pathway to leverage the collective knowledge stored in these models.
>
> > On the other hand, BEATs does not require an existing large-scale SSL model; it requires iterative refinement because the tokenizer is, well, iteratively refined from cold-start. The commentary on simplicity and training complexity within the context of this paper is moot.
>
> Thank you for raising this point. We have changed our statement regarding the complexity of BEATs in the introduction in our revised manuscript.
>
> ---
>
> ## [Q4] Results for audio tasks (Point 4)
>
> ### [Q4.1] Comparison of SPEAR_a and SPEAR_{s+a}
>
> > However, the majority of the (relatively small in proportion to the increase in pretraining dataset size) performance improvement in SPEAR-(a+s) over SPEAR-(a), for both Base and Large models, stems from improved speech performance, which is not surprising.
>
> We agree with the reviewer that the performance gain on HEAR benchmark of the dual-domain models stems from the usage of speech data during pre-training. This is precisely our goal -- **we would like the dual-domain model to learn unified representations by learning from two domain-specific models on a mixture of heterogenous speech and audio data.**
>
> We also agree that learning a unified representation for speech and audio is a challenging task, and that a tradeoff has been found in both domains (see our response to [Q1.2]). However, we still believe that this tradeoff is worthwhile due to the versatility of the dual-domain models.
>
> Concretely, SPEAR_a is an audio-domain specialist, while SPEAR_{s+a} is a dual-domain model suitable for both speech and audio processing tasks. That said, SPEAR_{s+a} not only improves over SPEAR_{a} on the HEAR benchmark, it also has a strong capability on speech-related tasks such as ASR and SUPERB. For example, SPEAR_a Large performs poorly on the ASR task (see Table 4), yielding WERs of 7.4/18.6 on LS-100, while SPEAR_{s+a} Large achieves 2.5/4.6. This unification makes the dual-domain SPEAR models unique.
>
> *(The remaining comments will be addressed in subsequent responses).*

---

> ### Author Response · Authors · 2025-11-20
> **Response to Reviewer y6wJ (5/6)**
>
> ### [Q4.2] Comparison of SPEAR and Dasheng
>
> > ... the base and large SPEAR-(a) and SPEAR-(a+s) models trail behind Dasheng-Base, the smallest Dasheng model, by quite a bit, despite having more parameters...
>
> We agree with the reviewer's observation that SPEAR models (both audio-domain and dual-domain) perform less well than the topline audio SSL Dasheng on HEAR benchmark. However, we respectfully argue that this performance gap is understandable due to the large difference of general audio data used during pre-training.
>
> **The HEAR benchmark focuses primarily on audio-related tasks**, with 12 out of 18 tasks being environment and music related. Therefore, the amount of general audio data (non-speech) used during pre-training is crucial to the performance of the model.
> Dasheng is pre-trained on over 272k hours of general audio (over 50% are non-human sounds), whereas SPEAR_a and SPEAR_{s+a} models only utilise 13k hours of general audio due to the scarcity of publicly available general audio data. Hence, we would like to attribute this audio performance gap between SPEAR and Dasheng to the **considerably smaller amount of non-speech general audio for pre-training (a factor of 20 times less)**.
>
> However, it is still noteworthy that SPEAR_a Large (without feature concatenation) outperforms Dasheng-1.2B on the environment tasks (Env) of HEAR (83.58 vs 83.2, Table 6). This achievement is remarkable considering the big gap in terms of model size (1/4) and the data volume (1/20), demonstrating the potential of applying SPEAR to audio SSL.
>
> To further supplement our claim, we discuss two additional experiments below:
>
> **[Strictly comparable audio SSL]**
>
> This experiment was presented in Appendix G.1 in the original manuscript. To rule out the effect of pre-training data difference and model size, we used ATST-frame (88M, 5k hours) as the teacher and pre-train a Base student model (94M, denoted as Base-Audio-1) on the same 5k hours data (i.e. AudioSet). We see that the SPEAR student model outperforms the highly competitive ATST-frame model on almost every task. **This implies that the performance of SPEAR is not upperbounded by the teacher model given a similar model size and data set.**
>
> | Model | # Params | Pre-train hours | AS-20k | AS-2M | ESC | FSD | GZ-Gen | NS-5 | LC | SC-5 |
> | --- | --- | --- | --- | --- | --- | --- | --- | --- | --- | --- |
> | ATST-frame | 88M | 5k | 39.0 | 48.0 |  89.0 | 55.7 |  88.3 |  68.6 | 78.1| 92.6|
> | Base-Audio-1 | 94M | 5k | 40.3 | 49.6 | 89.4 | 57.2 | 89.5 | 64.4 | 79.4 | 94.3 |
>
> **[Scaling general audio data helps]**
>
> We conducted an extra scaling experiment for SPEAR_a, where we increased the general audio data from 5k hours (AudioSet) to 13k hours (Audio-13k, see Table 2) for both Base and Large architectures, and report the average HEAR score below:
>
> | Model | 5k hours |  13k hours |
> | ---- | --- | ---- |
> | Base | 76.4 | 77.0 |
> | Large | 78.1 | 79.2 |
>
> With the increase of general audio data, a noticeable improvement is observed. Hence, we reasonably expect the model to continuously benefit from more general audio data. This also motivates us to collect more general audio data to further enhance the audio performance of SPEAR (also mentioned in the last paragraph of Appendix G.5), which we leave as imporant future work.
>
> ### [Q4.3] The advantage of SPEAR
>
> > The presented analysis showcases no benefit to training a complex SPEAR Audio+Speech model for audio SSL ... The merits of the SPEAR approach over the existing publicly available pretrained Dasheng models are not clear.
>
> We respectfully disagree with the statement made by the reviewer. We would like to emphasise that the capability of **unifying speech and audio representations** of SPEAR is the key advantage over existing single-domain speech (e.g. WavLM) or audio (e.g. Dasheng) SSL models. As an audio SSL model pre-trained with large volumn of general audio data, Dasheng shows very good performance on the audio-centric HEAR benchmark. However, it generalises poorly to the speech domain as manifested by our additional experiment supplemented (see supplemented additional experiments).
>
> - **SUPERB evaluation of Dasheng [EXP1]**: We evaluated Dasheng on the speech-centric SUPERB benchmark and observed that Dasheng performs more poorly than SPEAR_{s+a} models on all tasks
> - **ASR fine-tuning [EXP2]**: Full fine-tuning of Dasheng on LibriSpeech yields WERs of 3.4/8.7, which is much higher than SPEAR_{s+a} XLarge with WERs of 1.6/2.9
>
> From the above two experiments, we see that the speech capability of Dasheng is relatively weak, despite of its large model size and large amount of pre-training data (272k hours, with over 40% being human speech/sounds). The proposed dual-domain SPEAR_{s+a} models, however, achieve high performance on both HEAR and SUPERB. **We belive this versatility in speech and audio will make the SPEAR_{s+a} models highly desireable for modern AI systems with comprehensive audio perception capability**.

---

> ### Author Response · Authors · 2025-11-20
> **Response to Reviewer y6wJ (6/6)**
>
> ## [W5]: Models in Table 6 are difficult to compare (Point 5)
>
> > All the models presented in Table 6 are trained on different amounts of data. This only obfuscates results. BEATs and ATST-Frame are both trained only on 5k hours of audio data.
>
> We thank the reviewer for pointing out the differences in the volume of pre-training data in Table 6, which indeed makes it difficult to form conclusions. We have modified Table 6 in the revised manuscript to incorporate SPEAR_a Base and Large models pre-trained with 5k hours. This should enable a fairer comparison with audio SSL models such as BEATs (same pre-training data), and also allows the reader to see the scaling trend of SPEAR_a from 5k hours to 13k hours.
>
> We would like to summarise the key conclusions from Table 6:
> - **SPEAR_{s+a} consistently outperforms the baseline USAD model by a large margin**, even with less data (97k vs 126k). This demonstrates that the effectiveness of the novel SSL objective proposed by SPEAR framework over traditional knowledge distillation via feature matching.
> - **The dual domain SPEAR_{s+a} models consistently outperform their audio-domain counterparts SPEAR_{a}**, verifying the benefits of learning a joint representation space for speech and audio.
>
> Again, we would like to direct the reviewer to the ablation experiment in Appendix G.1 of the original manuscript, which is a controlled experiment under a strict fair comparison. **This controlled experiment strictly validates the effectiveness of SPEAR for audio SSL, since the student model with same pre-training data and same model size consistently outperforms the teacher ATST-frame under a strict fair comparison.**

---

> ### Comment · Reviewer_y6wJ · 2025-11-21
> **Response to author's comments**
>
> Thanks for your response. Most of the comments are well received, and I appreciate the additional experiments (albeit I didn't ask for them). Also, thank you for pointing out the experiments in the appendices.
>
> I have some comments about specific statements made in the rebuttal:
>
> > We agree with the reviewer that the performance gain on HEAR benchmark of the dual-domain models stems from the usage of speech data during pre-training. This is precisely our goal -- we would like the dual-domain model to learn unified representations by learning from two domain-specific models on a mixture of heterogenous speech and audio data.
>
> My comment implied that the improved performance is merely a result of more speech data. What if we added this amount of speech data to Dasheng's pretraining corpus? Wouldn't Dasheng perform better as well? It is not clear to me how the better performance of SPEAR {a+s} on HEAR can be attributed solely to the better representation quality and not merely to additional speech data.
>
> > However, we still believe that this tradeoff is worthwhile due to the versatility of the dual-domain models.
>
> But the experiments have not proven this. I can argue that the SPEAR models for speech perform better than baselines because it has seen more speech data than the baselines. SPEAR {a} models do not conclusively do better than the teacher models.
>
> > This experiment was presented in Appendix G.1 in the original manuscript.
>
> Thanks for pointing this out to me.
>
>
> ## Overall
>
> The author's comments made several good points.
>
>
> The comments on the better efficiency of SPEAR are still not clear to me. How could the research institutions that cannot afford to train the teacher SSL models afford to train SPEAR? On large datasets, extracting features from 2 frontier models (irrespective of whether it is done on-the-fly or pre-hoc) to train another large model is still quite expensive computationally. But let's consider this discussion moot for now, in the name of SSL being a huge compute sink (and I say that as a well-informed SSL researcher).
>
> However, my concerns regarding the novelty of the approach still remain. Knowledge distillation in this realm has been done before in USAD, which leverages frame-wise representation matching losses instead of token prediction. The authors state that they hypothesise that joint token prediction is a better-suited objective, but without satisfactory commentary on why or how. Furthermore, I do not see how token prediction would be that much superior to matching features as done by USAD when other factors are kept the same.
>
> Further, the Dasheng teacher still outperforms the proposed SPEAR-{a} and SPEAR-{a+s} models. The better speech performance could be attributed to the additional training data used.
>
> I will keep my score.

---

> ### Author Response · Authors · 2025-11-25
> **Follow-up responses (1/4)**
>
> Thank you for your response. We are glad to hear that our response has resolved some of your questions. Please find below our further response to your comment on the novelty of SPEAR and the performance comparison with Dasheng.
>
> ---
>
> ## Novelty of SPEAR
>
> > However, my concerns regarding the novelty of the approach still remain. Knowledge distillation in this realm has been done before in USAD, which leverages frame-wise representation matching losses instead of token prediction.
>
> **We would like to clarify that USAD is a contemporaneous work**. The USAD preprint appeared on arXiv two months prior to the ICLR submission deadline, and has not appeared on a peer-reviewed venue yet (it will appear at ASRU in Dec 2025). According to the ICLR review policy [1], this classifies USAD as contemporaneous work. Furthermore, our work on SPEAR was mostly complete before the USAD preprint appeared on arXiv. **Therefore, SPEAR should be evaluated on its own merits and novelty**.
>
> Despite this, we did include a comprehensive comparison with USAD on SUPERB (Table 5) and HEAR (Table 6) in our original submission and found that SPEAR consistently outperforms USAD.
>
> ---
>
> ### **Controlled comparison between SPEAR and USAD**
>
> > Furthermore, I do not see how token prediction would be that much superior to matching features as done by USAD when other factors are kept the same.
>
> In spite of the fact that USAD should be viewed as contemporaneous work, in order to further show the advantages of SPEAR, we have now also trained a new dual-domain SPEAR model with the Base architecture (94M parameters) named **SPEAR (USAD-aligned)**, mirroring the USAD settings to isolate the impact of the pre-training objective. Specifically:
>
> - **Teacher models**: We used the same teachers as USAD: WavLM Base+ (speech) and ATST-Frame (audio)
> - **Data**: We used a subset of the USAD training corpora, excluding Fisher and VoxLingua for speech and SoundNet for audio due to availability issues:
>     - **SPEAR (USAD-aligned)**: 86k hours speech + 9.3k hours audio (Total: ~95.3k hours).
>     - **USAD**: 91k hours speech + 35k hours audio (Total: ~126k hours).
>
> The comparison between SPEAR (USAD aligned) and USAD on SUPERB and HEAR is presented below (last four columns for HEAR):
>
> | Model | # Params | Data (h)| PR $\downarrow$ | ASR $\downarrow$ | IC $\uparrow$ | KS $\uparrow$ | SID $\uparrow$ | ER $\uparrow$ | Env $\uparrow$ | Speech $\uparrow$ | Music $\uparrow$ | Avg$\uparrow$ |
> | --- | --- | --- | --- | --- | --- | --- | --- | --- | --- | --- | --- | --- |
> | SPEAR (USAD-aligned) | 94M | 95.3k | **4.6** | **5.1** | **98.7** | **97.4** | **89.2** | **69.4** | **81.06** | **76.89** | **80.45** | **79.40** |
> | USAD Base |  94M | 126k | 5.1 | 7.7 | 98.3 | 97.1 | 88.6 | 68.0 | 80.67 | 73.72 | 79.31 | 77.75 |
>
>
>
>
> As shown above, **SPEAR (USAD-aligned) consistently outperforms USAD Base, despite using only a subset of the training data of USAD.** In fact, it even outperforms the USAD Large (327M parameters) on HEAR with 79.36 (see Table 6). We believe this controlled comparison provides strong evidence that our masked-token prediction objective in SPEAR is more effective than feature matching in USAD, which further validates our core novelty by formulating KD as an SSL task.
>
> [1]: [ICLR reviewer guide](https://iclr.cc/Conferences/2026/ReviewerGuide) (Q: Are authors expected to cite and compare with very recent work? What about non peer-reviewed (e.g., ArXiv) papers?)

---

> > ### Author Response · Authors · 2025-11-25
> > **Follow-up responses (2/4)**
> >
> > ## Novelty of SPEAR (Cont.)
> >
> > > The authors state that they hypothesise that joint token prediction is a better-suited objective, but without satisfactory commentary on why or how
> >
> > Apart from the empirically stronger results of SPEAR demonstrated in the controlled comparison against USAD (see our previous response), we also provide below a detailed explanation on why the masked-token prediction based approach used in SPEAR outperforms the feature matching objective used in USAD.
> >
> > **1. Feature matching constrains performance**: USAD minimises the L1 and cosine distance between student and teacher representations. This is a regression task where the optimal solution for the student is to perfectly mimic the teacher. **Consequently, the student performance is theoretically upper-bounded by the teacher performance.** In contrast, SPEAR treats distillation as a masked-token prediction task, enforcing the student to learn contextual dependencies and semantic structures that may not be explicitly present in the frame-level teacher features. This allows the student to generalise beyond the teacher (e.g. SPEAR_s Large vs WavLM Large, Base-Audio-1 vs ATST-Frame), which is inherently very difficult for L1-based KD.
> >
> > **2. Joint feature matching is ill-defined for disparate domains**: In USAD, the student effectively minimises the distance to two embedding spaces (speech and audio) simultaneously. The L1 losses induced by two teachers encourage the student model to find a "mean" of two representation spaces. Since speech and audio feature spaces are distinct, this "mean" representation may lie in a region of the manifold that lacks semantic meaning for either domain. SPEAR avoids this risk by quantising representations into discrete tokens via MVQ, where the tokens exhibit the capability of representing a certain characteristic of the input speech/audio data (see Appendix G.4 where we found a single codebook to contain rich speaker information). This allows the model to retain distinct, high-fidelity details by learning to predict the discrete tokens for both domains simultaneously, with lower risk of destructive interference.
> >
> > In summary, we believe the novelty of SPEAR lies not only in proposing multi-teacher knowledge distillation for joint speech-audio representation learning, but also in formulating KD as an SSL pretext task, enabling the model to learn more generic representations from both teachers.

---

> > > ### Author Response · Authors · 2025-11-25
> > > **Follow-up responses (3/4)**
> > >
> > > ## Comparison of SPEAR with Dasheng (speech)
> > >
> > > > The better speech performance could be attributed to the additional training data used.
> > >
> > > We respectfully argue that the stronger speech performance of SPEAR_{s+a} is not merely a result of additional speech data. Please find below our detailed explanation:
> > >
> > > **1. Dasheng actually utilises more speech data than SPEAR**.
> > >
> > > Dasheng 1.2B is trained on over 272k hours of data. The primary dataset used is ACAV100M [1], and according to its official statistics (and verified by the authors of Dasheng), approximately 41.8% of this data consists of human speech/sound. This implies that the Dasheng model was exposed to more speech data than our model:
> > >
> > > - Dasheng 1.2B: ~110k hours (263k * 0.418) speech data
> > > - SPEAR_{s+a} Large: 84k hours of speech data (Table 2)
> > >
> > > **Despite using nearly 30% more speech data and having a significantly larger model size (1.2B vs 327M), Dasheng 1.2B still underperforms the smaller SPEAR_{s+a} Large model on key speech benchmarks**:
> > >
> > > - Speech category in HEAR (see Table 6):
> > >     - SPEAR_{s+a} Large: 76.47
> > >     - Dasheng 1.2B: 75.72
> > > - ASR & SUPERB: As detailed in our previous response, Dasheng lags behind significantly on ASR fine-tuning and on **all tasks** in SUPERB benchmark. For instance, SPEAR_{s+a} Large achieves WERs of 1.7/3.4 on LibriSpeech, whereas Dasheng 1.2B yields 3.4/8.7, i.e., more than double the WERs of SPEAR_{s+a} Large.
> > >
> > > **2. Training objective**
> > >
> > > Apart from the empirical evidence, we would also like to provide a theoretical explanation on why Dasheng performs more poorly on speech-related tasks than SPEAR from the perspective of training objective. Dasheng is trained with MAE objective by reconstructing the input acoustic features. In the speech community, reconstruction-based pre-training [2][3] has been shown to much less effective than the masked token prediction based methods (e.g. WavLM) for speech tasks requiring high-level semantic/phonetic information (e.g. ASR), since reconstruction-based objectives tend to focus on low-level acoustic detail. On the other hand, SPEAR utilises mask-token prediction, a pretext task known for learning high-level semantic structures from the data, which we believe is the key reason of the better performance of SPEAR_{s+a} on speech tasks.
> > >
> > > In summary, we believe that the substantial performance gap between Dasheng and SPEAR on speech-related tasks cannot be simply removed by increasing the amount of speech data used for Dasheng. We believe the better performance of SPEAR stems directly from our joint masked-token prediction objective, which effectively learns both acoustic and semantic structures from domain-specific teachers.
> > >
> > > **References**
> > >
> > > [1]: Lee, Sangho, et al. "Acav100m: Automatic curation of large-scale datasets for audio-visual video representation learning." Proc ICCV. 2021.
> > >
> > > [2]: Liu, Andy T, et al. "Tera: Self-supervised learning of transformer encoder representation for speech." IEEE/ACM Transactions on Audio, Speech, and Language Processing 29 (2021)
> > >
> > > [3]: Liu, Andy T., et al. "Mockingjay: Unsupervised speech representation learning with deep bidirectional transformer encoders." Proc. ICASSP 2020.

---

> > > > ### Author Response · Authors · 2025-11-25
> > > > **Follow-up responses (4/4)**
> > > >
> > > > ## Comparison of SPEAR with Dasheng (audio)
> > > >
> > > > > Further, the Dasheng teacher still outperforms the proposed SPEAR-{a} and SPEAR-{a+s} models.
> > > >
> > > > We believe that **Dasheng achieves a higher overall score than SPEAR on the HEAR benchmark, mainly because of its significantly larger amount of general audio data in the training set**. We support our argument using the following three points:
> > > >
> > > > **1. Data difference**: In Dasheng [1], the authors explicitly mentioned that the main contribution of Dasheng is to scale up MAE-based audio pre-training [2] to bigger model sizes (1.2B) and larger training data set (272k hours, with over 50% being general audio data). However, SPEAR_a Large only used 13k hours of general audio data, while SPEAR_{s+a} Large used 13k hours of general audio data plus 84k hours of speech data, both of which are significantly smaller than the pre-training corpora used by Dasheng.
> > > >
> > > > In the Dasheng paper, the authors conducted a scaling experiment (Table 6 in [1]), providing the HEAR score of Dasheng models trained on AudioSet 5k hours. We report the HEAR score of Dasheng and SPEAR_a pre-trained on AudioSet below:
> > > >
> > > > | Model | Pre-train Data | Model Size | HEAR score |
> > > > | --- | --- | --- | --- |
> > > > | Dasheng Base | AudioSet |86M | 70.43 |
> > > > | Dasheng 1.2B | AudioSet |1.2B | 74.87 |
> > > > | SPEAR_a Base | AudioSet |94M | 76.37 |
> > > > | SPEAR_a Large | AudioSet |327M | 78.08 |
> > > >
> > > > As can be seen, SPEAR_a Large achieves 78.08 on HEAR, 3.21 absolute higher than the 4-times bigger Dasheng 1.2B model pre-trained on the same data, verifying the effectiveness of SPEAR on audio SSL.
> > > >
> > > > **2. Scaling audio data helps SPEAR**: We would like to emphasise the clear positive data scaling trend of SPEAR. In our previous response, we showed that the scaling of the general audio data from 5k to 13k hours yields consistent gains, suggesting the potential of SPEAR for large-scale audio SSL:
> > > >
> > > > - 5k hours: 76.37 (Base), 78.08 (Large)
> > > > - 13k hours: 77.01 (Base), 79.18 (Large)
> > > >
> > > > **3. Strictly comparable audio SSL experiments**: While we could not replicate the massive 272k-hour pre-training corpus of Dasheng due to data access constraints, we would like to emphasise that we provide a fully controlled audio SSL experiment in Appendix G.1. The Base-Audio-1 model outperforms its teacher ATST-Frame on almost every task, which is another strong audio-SSL model trained on AudioSet, under identical data conditions and similar model sizes. Note that ATST-Frame (5k hours, 88M) shows similar performance as Dasheng Base (5k hours, 86M), both achieving HEAR score of 70.4 (no feature concatenation).
> > > >
> > > > **Based on the aforementioned three points, we believe that SPEAR will outperform Dasheng-1.2B given a similar amount of pre-training data and similar model size, and the current weaker performance is only due to its far smaller general audio data set (smaller by a factor of 10).** This motivates us to collect more general audio data, which we leave as an important future work.
> > > >
> > > > **References**
> > > >
> > > > [1]: Dinkel, Heinrich, et al. "Scaling up masked audio encoder learning for general audio classification." Proc. Interspeech, 2024.
> > > >
> > > > [2]: Huang, Po-Yao, et al. "Masked autoencoders that listen." Proc. NeurIPS, 2022.

---

> ### Comment · Reviewer_y6wJ · 2025-11-26
> **Response to author's comments**
>
> Thanks for additional comments. Let me address them one by one.
>
> ### About USAD
>
> > The USAD preprint appeared on arXiv two months prior to the ICLR submission deadline, and has not appeared on a peer-reviewed venue yet.
>
> I am well aware of the ICLR reviewer guidelines and the code of conduct. I am also aware that USAD first appeared on arXiv on 23rd June 2025. For the sake of precision, that is 3 months before the ICLR submission deadline, not 2 months, so if we are being "extremely" precise, like nitpicky precise, USAD is not a contemporaneous work.
>
> But I am not being extremely, nitpicky precise. The authors chose to include comparisons with USAD in their original paper. Fair and well, but then the onus of a good comparison is also on the authors.
>
> The latest comments highlight a good comparison now, so we can move on. SPEAR works better than USAD, which is crystal clear. Thanks for running these comparisons.
>
> ---
>
> ### Commentary on why joint token prediction is a better-suited objective
>
> Thank you! Kindly add these arguments to the paper. This commentary is necessary and actually makes your paper stronger.
>
> ---
>
> ### Comparison of SPEAR with Dasheng (SPEECH)
>
> My comment "better speech performance could be attributed to the additional training data used" was meant for SPEAR-{a} and SPEAR-{a+s} comparison, not with Dasheng, as made originally in my review. However, I understand and take responsibility for the imprecise framing of my comment responding to your rebuttal. I am well aware of the amount of speech data used for Dasheng.
>
> However, something quite constructive emerged from the author's response. The following statements:
>
> > Apart from the empirical evidence, we would also like to provide a theoretical explanation on why Dasheng performs more poorly on speech-related tasks than SPEAR from the perspective of training objective. Dasheng is trained with MAE objective by reconstructing the input acoustic features. In the speech community, reconstruction-based pre-training [2][3] has been shown to much less effective than the masked token prediction-based methods (e.g. WavLM) for speech tasks requiring high-level semantic/phonetic information (e.g. ASR), since reconstruction-based objectives tend to focus on low-level acoustic detail.
>
> This statement, either in its entirety or in a shortened state, is worth including in the commentary when the authors discuss the results in the paper. Instead of relying solely on empirical results, this also sets a foundation based on precedent from existing research regarding the design choices made in the paper, and shows that your results corroborate these observations.
>
> ---
>
> ### 1-on-1 comparison with Dasheng.
>
> Thanks for adding these comparisons. Similar to what I said previously regarding USAD, in my eyes, the onus of comparing in a consistent, data-controlled setting with baseline models is on you, the authors, and it is even more imperative in distillation-based settings. The provided additional comparison further strengthens the paper.
>
> ---
>
> ## Concluding remarks
>
> 1. I do not speak this from a place of authority, but from the place of the constraints on the reviewer (who is also but a peer); in my eyes, as a reviewer, I can only offer a limited amount of direction regarding what commentary to add to the paper to make the language better and to strengthen it, the authors have to arrive at and then specify justifications/precedents that strengthen their arguments on their own. Maybe I could have done this better, maybe not. Nonetheless, the authors did so successfully in the last couple of responses, and the inclusion of these points would strengthen the paper considerably. Please feel free to let me know when, where and how you are incorporating these changes in the paper, and I will be happy to  give my two cents.
>
> 2. My point regarding better speech performance of SPEAR-{a+s} compared to SPEAR-{a} and even other speech baselines still stands.
>
> 3. I appreciate all the additional comparisons done and the time the authors took to respond to my arguments. Overall, based on our correspondence as well as those with other reviewers, I think the rebuttal has considerably improved the paper. I will increase my rating.

---

> ### Author Response · Authors · 2025-11-28
> **Thank you for raising score!**
>
> Thank you for your quick response and positive feedback. We’re glad to hear that your concerns have been resolved, and we truly appreciate you raising the score to 6 in support of the acceptance of our paper. We will incorporate the additional experiments and theoretical discussion you suggested into the revised manuscript.

---

### Official Review · Reviewer_E2Pw · 2025-10-26

**Soundness:** 3
**Presentation:** 3
**Contribution:** 3
**Rating:** 6
**Confidence:** 5

**Summary:**

The paper proposes SPEAR, a unified self-supervised learning (SSL) framework for speech and general audio. The core idea is to replace coarse token targets (k-means) with MVQ tokens extracted from strong teacher models (WavLM-Large for speech and Dasheng-1.2B for general audio). And to train a Zipformer encoder by masked token prediction. A dual-domain asymmetrical loss encourages a joint representation space while weighting audio vs. speech targets during mixed training. Models are trained at 94M, 327M, and 600M scales on mixtures up to 197k hours; results are reported on LibriSpeech ASR, AudioSet audio tagging, SUPERB, and HEAR, with consistent gains vs. WavLM and USAD baselines.

**Strengths:**

1. **Originality.** Uses MVQ tokens from teacher models to supply fine-grained discrete supervision for both speech and audio, contrasting with k-means/RPQ in prior speech SSL, and introduces an asymmetrical dual-domain loss.

2. **Quality.** Thorough empirical study: LibriSpeech ASR (RNN-T and CTC), AudioSet tagging, SUPERB, HEAR; dual-domain gains and scaling trends; targeted ablations (teacher, codebooks, dual-domain strategy).

3. **Clarity.** MVQ formulation and encoding/decoding are clearly described. Zipformer specs and training settings are documented.

4. **Significance.** Strong performance across domains: AS-2M mAP 50. Consistent improvements vs. WavLM at matched scales on SUPERB.

**Weaknesses:**

1. **Compute transparency.** The paper lists steps and batch sizes but provides no hardware or GPU hours accounting for pre-training and fine-tuning. For a 600M dual-domain model on up to 197k hours, the compute cost should be reported.

2. **Attribution of gains.** While Appendix G ablates several factors, the main text does not clearly isolate the source of gains (teacher strength vs. MVQ vs. Zipformer vs. data scale). A controlled comparison using the same encoder and data but different quantization targets (MVQ vs. k-means/RPQ) under the same training loss would better attribute sources of improvement. Current k-means comparison is qualitative rather than a training baseline.

3. **Quantization comparisons.** MVQ vs. k-means is visualized in Appendix G.4, but there is no empirical head-to-head SPEAR-with-k-means (or RPQ) target training. Including such baselines would strengthen the claim that fine-grained MVQ targets are the primary driver.

4. **Teacher dependence and fairness.** The audio teacher (Dasheng-1.2B, 272k hours) is very large. Although USAD is compared, a more systematic study of teacher combinations/sizes (e.g., smaller audio and speech teachers) would clarify how much of the improvement is inherited from teachers vs. MVQ.

5. **Main paper balance.** Many key analyses (teacher choice, codebooks, loss balancing, dual-domain strategies) are relegated to the appendix, making it difficult for readers to assess mechanisms from the main narrative.

**Questions:**

1. **Quantization approaches.** Please add direct baselines where SPEAR uses k-means or RPQ tokens (same teacher layers, same training loss, same data/steps) to quantify the specific benefit of MVQ. The paper already positions k-means/RPQ as common in speech SSL and visualizes MVQ vs. k-means subspaces. A training baseline would make the argument causal.

2. **Ablations to attribute gains.** Please surface in the main paper a compact set of ablations that answer: (a) teacher choice (HuBERT vs. WavLM for speech; ATST-Frame vs. Dasheng for audio), (b) codebook count $N$ (e.g., 4/8/16) and fixed $K=256$, and (c) dual-domain strategy (joint/disjoint/asymmetrical). These are in Appendix G. Summarizing them centrally would clarify the source of performance.

3. **Teacher combinations and scale.** USAD distills from smaller teachers; could you include experiments with smaller audio teachers to test whether MVQ still confers advantages when teacher capacity/data are reduced?

4. **Compute and resources.** Please report hardware (GPU/TPU type and count) and total pre-training steps $\times$ batch seconds, split by model scale (94M/327M/600M). This is essential given the scale of Mix-197k and the 600M model. The current paper lists steps and batch sizing, but not the computation.

5. **Objective sensitivity.** You tune $\alpha$ (masked vs. unmasked loss weight) and adopt $\lambda=0.1$ for the dual-domain term. Could you show sensitivity curves for $\lambda$ (and possibly per-domain $\alpha$) to establish robustness and to guide other researchers?

6. **Encoder architecture.** Zipformer is a design choice. A brief ablation (even at Base scale) comparing Zipformer vs. a Conformer/Transformer under SPEAR would help separate architectural benefits from MVQ/teacher effects.

7. **Quantizer training details.** You state $K=256$ for storage efficiency and vary $N$; please also report codebook training compute and whether using earlier/later teacher layers for MVQ training changes downstream results (you use 21st WavLM layer for speech; last layer for Dasheng).

---

> ### Author Response · Authors · 2025-11-20
> **Response to Reviewer E2Pw (1/3)**
>
> We thank the reviewer for the thorough, positive, and constructive review. We are grateful for the reviewer's detailed feedback, particularly for highlighting the originality of our approach, the quality of our empirical study, the clarity of the paper's presentation, and the significance of our model's strong performance. Below, we provide detailed responses to all of the questions and concerns raised.
>
> ---
>
> ## [W1 & Q4]: Computation resources
> > The paper lists steps and batch sizes but provides no hardware or GPU hours accounting for pre-training and fine-tuning.
>
> All SPEAR models were pre-trained using NVIDIA A800 (80GB) GPUs. The approximate total GPU hours (not elapsed time) required for training each of the SPEAR models are listed below:
>
> | Model | # GPUs | Batch size | # Updates | # GPU hours |
> | --- |  --- | --- | --- | --- |
> | SPEAR_s Base | 8 | 4.8k | 400k | 460 |
> | SPEAR_a Base | 8 | 4.8k | 250k | 290 |
> | SPEAR_{s+a} Base | 8 | 6.4k | 400k | 660 |
> | SPEAR_s Large | 8 | 4.8k | 400k | 900 |
> | SPEAR_a Large | 8 | 4.8k | 250k | 560 |
> | SPEAR_{s+a} Large | 16 | 6.4k | 500k | 2,000 |
> | SPEAR_{s+a} XLarge | 32 | 6.4k | 500k | 3,800 |
>
> We will add this information to the appendix in the revised manuscript.
>
> ---
>
> ## [W2 & W3 & Q1]: Attribution of gains (Comparison of MVQ and k-means)
>
> > A controlled comparison using the same encoder and data but different quantization targets (MVQ vs. k-means/RPQ) under the same training loss would better attribute sources of improvement.
>
> Following the reviewer's suggestion, we carried out another experiment, where we compared two SPEAR Base models pre-trained with k-means token and MVQ tokens. The same embedding layer (21st layer of WavLM Large) was used to extract k-means and MVQ tokens. Both models are pre-trained on LS-960 for 300k updates with a batch size of 3200 seconds. The following results are given below: ASR word error rates with LS-100 ASR fine-tuning and 5 tasks from the SUPERB benchmark:
>
> | Discrete Target | LS-100 $\downarrow$ | PR $\downarrow$| IC $\uparrow$ | KWS $\uparrow$ |  SID $\uparrow$ | ER $\uparrow$ |
> | --------------- | -----------| --- | ---| --- | --| --|
> | k-means, 2000 cluster | 3.5/7.2 | 4.0 | 97.92 |  96.79 | 86.6 | 67.56 |
> | Ours, MVQ | **3.1/7.0** | **3.4** | **98.37** | **96.83** | **88.4** |**68.29** |
>
> We observe that the model trained with MVQ tokens achieves better performance on all tasks, especially two paralinguistic tasks: SID and ER. This aligns with our findings in Appendix G.4 in the original manuscript, where we found that the fine-grained MVQ tokens retain richer paralinguistic information than the coarse k-means token. We thank the reviewer for proposing this experiment, which helps further verify the advantages of SPEAR.
>
> ## [W4 & Q3]: Teacher dependence and fairness
>
> > a more systematic study of teacher combinations/sizes (e.g., smaller audio and speech teachers) would clarify how much of the improvement is inherited from teachers vs. MVQ.
>
> We direct the reviewer to the ablation study in Appendix G.1 in the original manuscript, where we compare teacher models with different model and training data sizes. We used HuBERT Large (317M, 60k hours) and ATST-frame (88M, 5k hours) as the teacher, which are smaller in terms of model capacity and data compared to the teachers in the main paper. Here, we summarise the key results and observations:
>
> - **Stronger teacher model generally leads to a stronger student model**. As shown in Tables 16 and 17 in the original manuscript, SPEAR_s Large (with WavLM Large teacher) outperforms Large-H-2 (with HuBERT Large teacher) on SUPERB, and BASE-AUDIO-2 (with Dasheng teacher) outperforms BASE-AUDIO-1 (with ATST-frame teacher) on HEAR. This suggests that **it is necessary to select a powerful teacher model for the optimal performance of SPEAR**.
> - **Student models (with the same model size and training data) are able to outperform their teacher models**. Despite relying on the teacher model for generating pre-training targets, we found that the performance of SPEAR is not upperbounded by its teacher:
>     - Table 16: LARGE-H-1 outperforms its teacher HuBERT Large on all SUPERB tasks
>     - Table 17: BASE-AUDIO-1 outperforms its teacher ATST-frame on all tasks (except NS-5)
>
> We would like to further highlight the result in Table 17, where we use ATST-frame as the audio teacher. ATST-frame only has 88M model parameters and was trained on only 5k hours of audio data, which is much smaller compared to the Dasheng-1.2B teacher. However, the MVQ tokens extracted from it still conveys high-quality supervision to the student model, reflected by an mAP of 49.6 on AS-2M, surpassing both the teacher model (48.3) and another student model (49.3) trained with the more powerful Dasheng-1.2B as teacher.
>
> *(The remaining comments will be addressed in subsequent responses).*

---

> ### Author Response · Authors · 2025-11-20
> **Response to Reviewer E2Pw (2/3)**
>
> ## [W5]: Main paper balance
>
> > Many key analyses (teacher choice, codebooks, loss balancing, dual-domain strategies) are relegated to the appendix, making it difficult for readers to assess mechanisms from the main narrative
>
> Thank you for the suggestion. We will make sure to improve this in the revised manuscript.
>
> ---
>
> ## [Q5]: Impact of $\lambda$
>
> > You tune $\alpha$ (masked vs. unmasked loss weight) and adopt
>  $\lambda$=0.1 for the dual-domain term. Could you show sensitivity curves for $\lambda$ (and possibly per-domain $\alpha$) to establish robustness and to guide other researchers?
>
> Thank you for the question. For an overview of the effect of $\alpha$, we direct the reviewer to Appendix G.2 in the original manuscript, where we vary $\alpha$ from 0.0 to 1.0 for single-domain MVQ experiments (see Figure 2). We observe that using $\alpha=0.5$ achieves best performance for both domains, which means an equal contribution of loss on masked and un-masked regions is beneficial. An explanation for this can also be found in Appendix G.2 in the original manuscript.
>
> Following the your request, we also report below the effect of varying $\lambda$, which controls the loss contribution on audio data in dual domain training. We have experimented with 3 values of $\lambda$ using the Large architecture and compare the fine-tuning performance on LS-100 and AS-20k after 100k pre-training steps,
>
> | $\lambda$ | test-clean $\downarrow$ | test-other $\downarrow$ | mAP $\uparrow$ |
> | ---- | ---- | ---- | ---- |
> | 0.3 | 3.0  | 5.9  | **37.0** |
> | 0.2 | 3.0  | 5.7  | 36.9 |
> | 0.1 (adopted)|  **2.9** | **5.6** | 36.9 |
>
> We found that reducing $\lambda$ from 0.3 to 0.1 improves the WERs on test-other by 0.3 absolute, where as the mAP is only reduced by 0.1 absolute. Therefore, we adopted $\lambda$=0.1 in our dual-domain experiments for a balanced performance across both domains.
>
> ---
>
> ## [Q6]: Encoder architecture
>
> We have compared the encoder architecture of SPEAR (Zipformer + fbank) with the commonly used Transformer + waveform. The transformer implementation follows WavLM, which is an improved version of wav2vec 2.0. We pre-trained the two models using the same pre-training loss under the SPEAR framework on LS-960 data for 300k updates. Below the following results are reported: ASR word error rates with LS-100 ASR fine-tuning and the SUPERB benchmark.
>
> | Architecture | LS-100 $\downarrow$ | PR $\downarrow$| IC $\uparrow$ | KWS $\uparrow$ |  SID $\uparrow$ | ER $\uparrow$ |
> | ---- | ---- | ---- | ---- | ---- | --- | --- |
> | Transformer + waveform | 4.1/9.0 | **3.4** | 97.79 | **97.27** | **89.8** | 66.27 |
> |  Zipformer + fbank | **3.1**/**7.0** | **3.4** | **98.37** | 96.83 | 88.4 |**68.29** |
>
> On the SUPERB tasks which do not allow fine-tuning the encoder parameters, both architectures show similar performance, with Transformer+waveform being stronger on SID, while Zipformer+fbank is stronger on ER. However, Zipformer+fbank is particularly strong on the ASR downstream fine-tuning task, since Zipformer is an architecture specifically designed for ASR.
>
> In addition, Zipformer also requires less computation than Transformer due to the downsampling operation in its intermediate layers. The GPU hours required for pre-training the two models above are as follows:
> - Zipformer + fbank: 8 GPUs, ~350 GPU hours
> - WavLM + waveform: 8 GPUs, ~600 GPU hours
>
> In conclustion, the better ASR fine-tuning performance and computation efficiency motivates us to select Zipformer as the backbone architecture for SPEAR.
>
> *(The remaining comments will be addressed in subsequent responses).*

---

> ### Author Response · Authors · 2025-11-20
> **Response to Reviewer E2Pw (3/3)**
>
> ## [Q7]: Quantisation efficiency and configuration
>
> > You state K=256 for storage efficiency and vary N; please also report codebook training compute
>
> Using a codebook size of 256 enables us to store all the pre-trained targets in the format of uint8, which only takes one byte to represent each codebook entry. During pre-training, the MVQ tokens are pre-computed and stored to disk. For the case of using WavLM as the teacher model with 16 codebooks, storing the codebook indexes requires 2.88 Gb of storage space per 1000 hours of speech data. The training of the MVQ quantiser over a 100-hour training set takes only 1.5 hours on a single A800 GPU.
>
> > whether using earlier/later teacher layers for MVQ training changes downstream results.
>
> We also present the experimental results comparing different teacher layers for extracting pre-training targets. In our initial experiments, the teacher layer is selected based on the downstream fine-tuning performance. We used the SPEAR Base architecture in all experiments below.
>
> For the speech teacher, we compared using the 18-th, 21-st and the 24-th (last) layers of WavLM Large and reported the WERs on LS-100 fine-tuning tasks. We selected the 21-st layer since it gave the lowest WERs.
> | WavLM layer index | test-clean $\downarrow$ | test-other $\downarrow$ |
> | ----------- | ------ | ------------ |
> | 18 |  3.1 | 6.1 |
> | 21 (adopted) |  **3.0** | **5.8** |
> | 24 | 3.1 | 6.0 |
>
> For the audio teacher, we compared using the 25-th, 35-th and 40-th layer (last layer) of Dasheng and reported the mAP on AS-20k fine-tuning task. We selected the last layer (40-th) since it gave the highest mAP.
>
> | Dasheng layer index | mAP $\uparrow$ |
> | ----------- | ------ |
> | 25 | 38.4 |
> | 35 | 38.7 |
> | 40 (adopted) | **39.2** |

---

> ### Comment · Reviewer_E2Pw · 2025-11-20
> **Thanks for the comments & additional results**
>
> Thank you for providing the requested experimental results; including them will strengthen the paper's quality and technical soundness. I will update the soundness score accordingly.
>
> Given that I cannot assume the revised version will fully resolve my concerns regarding W5, and because the current level of novelty does not justify raising the contribution score to a 4, I am keeping my other ratings (presentation, contribution, and overall recommendation) unchanged.

---

> > ### Author Response · Authors · 2025-11-25
> > **Follow-up response**
> >
> > Thank you for your response and for raising the soundness score in light of the new experimental results. We are pleased that these additions have strengthened the technical quality of the work. As per your suggestion, all new results have been integrated into the revised manuscript. **The revised manuscript has been uploaded.**
> >
> > ---
> >
> > We would like to further address your comments regarding the main paper balance (W5) and the novelty of SPEAR:
> >
> > ## [W5]: Main paper balance
> >
> > To address your concern that key mechanisms were relegated to the appendix, we have restructured the results section in the revised manuscript. Specifically, we added a new subsection, **Section 5.4: Ablation Studies** (Page 10), to the main text.
> >
> > This section now explicitly summarises the critical findings regarding:
> > - **Teacher model selection**: SPEAR performs better with stronger teacher models, and the performance of student models are not upper-bounded by their teachers.
> > - **Controlled comparison between MVQ and k-means**: Head-to-head comparison of MVQ and k-means as the pre-training target, attributing the performance gains specifically to the fine-grained nature of MVQ tokens.
> > - **Dual-domain training strategies**: Compare three dual-domain training strategies and justify the adopted asymmetrical design for handling imbalanced speech and audio data.
> >
> > While the secondary ablations (such as the weighting factor $\alpha$, codebook numbers, and encoder architecture) are not discussed in detail in the main text, the last paragraph of Section 5.4 provides clear pointers to them.
> >
> > **We believe this structural change ensures that the core mechanisms and design justifications are now fully accessible within the main narrative**, allowing readers to assess the correctness of SPEAR without relying on the appendix. We hope this direct inclusion resolves your concern regarding W5.
> >
> > ---
> >
> > ## Novelty of SPEAR
> >
> > We would like to emphasise that **SPEAR is the first to successfully learn unified speech and audio representations via a single SSL objective from a mixture of heterogeneous speech and general audio data.**
> >
> > To summarise, SPEAR's novelty is as follows:
> > - **A novel SSL objective for unified speech and audio representation learning**: We are the first to propose masked prediction of fine-grained tokens as a unified objective for both speech and audio domains. Prior to SPEAR, SSL approaches for speech or audio domain are highly domain specific. SPEAR bridges this gap by proposing a unified SSL pretext task of masked prediction of fine-grained MVQ tokens on both domains. The fine-grained MVQ tokens retain rich acoustic and semantic detail, making joint SSL pre-training on speech and audio feasible.
> > - **Multi-domain joint KD**: We propose to learn a unified speech-audio representation space by jointly distilling teachers from both domains. **Although USAD also adopts similar ideas, we note that USAD is a contemporaneous work to SPEAR according to the ICLR review policy [1]**. In fact, USAD has not yet been published/presented at a peer-reviewed venue (USAD will be presented at ASRU, Dec 2025). Furthermore, our work on SPEAR was mostly complete before the USAD preprint appeared on arXiv. **Therefore, SPEAR should be evaluated on its own merits and novelty at the time of submission**. In addition, we have designed a novel asymmetrical multi-domain training strategy (see Section 3.2.2) in SPEAR to cope with the imbalance of the available speech and general audio training data, which proves to be essential for SPEAR to achieve a high and balanced performance across both domains.
> > - **Novel combination of SSL and KD**. We are the first to formulate KD as a masked-token prediction SSL task using MVQ-quantised teacher embeddings. The fine-grained MVQ tokens derived from the teacher models transfer domain-specific knowledge to the student model, while the masked-token prediction objective encourages the student to learn contextualised representations and semantic structure from the data. This design makes SPEAR fundamentally different from other KD methods (e.g. USAD) that rely on feature-matching objectives (e.g. L1 loss and cosine similarity), which is the key to the stronger performance of SPEAR. In fact, SPEAR can learn representations that even outperform its teachers (e.g., SPEAR_s vs. WavLM in Table 5), which is inherently very difficult for traditional KD methods such as USAD.
> >
> > In conclusion, we argue that SPEAR represents a foundational step towards universal auditory perception, as it provides a single, versatile model with a unified representation space for speech and audio.
> >
> > **References**
> >
> > [1]: [ICLR reviewer guide](https://iclr.cc/Conferences/2026/ReviewerGuide) (Q: Are authors expected to cite and compare with very recent work? What about non peer-reviewed (e.g., ArXiv) papers?)

---

> ### Author Response · Authors · 2025-11-28
> **Thank you for your feedback!**
>
> Thank you again for your quick response and positive feedback, and for increasing the soundness score from 3 to 4. We’re glad to hear that our responses have addressed your concerns. We have already incorporated the additional results and discussions into our revised manuscript.

---

### Official Review · Reviewer_UWLL · 2025-10-27

**Soundness:** 3
**Presentation:** 4
**Contribution:** 3
**Rating:** 8
**Confidence:** 4

**Summary:**

This paper proposes a new self-supervised distillation technique to unify speech and audio SSL models.
Using MVQ representations from two well-known SSL models (WavLM Large for speech and Dasheng 1.2B for audio), the authors train a ZipFormer on filterbanks to produce representations H that can effectively reconstruct the quantized representations from WavLM and Dasheng.
The SPEAR model, obtained by distilling SSL models in an SSL manner, is therefore an SSL model itself.
More than preserving the capabilities of both models, SPEAR outperforms them in most tasks from the SUPERB and HEAR benchmarks, respectively, for speech and audio.

**Strengths:**

The article is very clear, shows an interesting way of merging speech and audio representations.
The structure is linear, and the experiments show in great detail the comparison to existing models through known benchmarks.

**Weaknesses:**

A few weak points can be identified still:
This article proposes multiple factors of improvement: the combination of speech and audio teacher models, the use of MVQ representations to jointly represent them, and the use of WavLM and Dasheng as teachers, on an architecture different than the Wav2Vec2.0 framework (namely filterbanks + Zipformer).
First: One or multiple ablation studies would have been interesting, to better judge which aspects of the pipeline have the greatest impact.
Second: One of the main points of the article is to propose a new SSL model that outperforms WavLM-Large (which has been the top model on the SUPERB benchmark for some time now). However, it could be clarified that this model is actually distilling WavLM-Large during its SSL training.

**Questions:**

About the ablation studies previously mentioned:
Did you try to replace your model by a Wav2Vec2.0 architecture?
Or did you tried to use the same L1 loss as in USAD, removing the MVQ, while keeping everything else similar?

---

> ### Author Response · Authors · 2025-11-20
> **Response to Reviewer UWLL (1/2)**
>
> We thank you for the positive and insightful feedback. We appreciate your clear summary of our core contribution, namely that SPEAR is “an SSL model itself” trained “in an SSL manner” from other SSL models. We are glad that the clarity, presentation, and experimental details were well received. We also appreciate your constructive suggestions for further improvement. Below, we address the specific points you raised.
>
> ---
>
> ## [W1]: More ablation studies
>
> > This article proposes multiple factors of improvement: the combination of speech and audio teacher models, the use of MVQ representations to jointly represent them, and the use of WavLM and Dasheng as teachers, on an architecture different than the Wav2Vec2.0 framework (namely filterbanks + Zipformer) ... One or multiple ablation studies would have been interesting, to better judge which aspects of the pipeline have the greatest impact
>
> Thank you for your point. Following your suggestion, we have performed extra ablation studies on the following components of SPEAR, which we believe should be useful for a better understanding of SPEAR.
>
> ---
>
> ### [W1.1] Encoder architecture
>
> We have compared the encoder architecture of SPEAR (Zipformer + fbank) with the commonly used wav2vec 2.0 architecture (transformer + waveform). The transformer implementation follows the WavLM paper, which is an improved version of wav2vec 2.0. We pre-trained the two models using the same pre-training loss under the SPEAR framework on LS-960 data for 300k updates. Below the following results are reported: ASR word error rates with LS-100 ASR fine-tuning and the SUPERB benchmark.
>
> | Architecture | LS-100 $\downarrow$ | PR $\downarrow$| IC $\uparrow$ | KWS $\uparrow$ |  SID $\uparrow$ | ER $\uparrow$ |
> | ---- | ---- | ---- | ---- | ---- | --- | --- |
> | Transformer + waveform | 4.1/9.0 | **3.4** | 97.79 | **97.27** | **89.8** | 66.27 |
> |  Zipformer + fbank | **3.1**/**7.0** | **3.4** | **98.37** | 96.83 | 88.4 |**68.29** |
>
> On the SUPERB tasks which do not allow fine-tuning the encoder parameters, both architectures show similar performance, with Transformer+waveform being stronger on SID, while Zipformer+fbank is stronger on ER. However, Zipformer+fbank is particularly strong on the ASR downstream fine-tuning task, since Zipformer is an architecture specifically designed for ASR.
>
> In addition, Zipformer also requires less computation than Transformer due to the downsampling operation in its intermediate layers. The GPU hours required for pre-training the two models above are as follows:
> - Zipformer + fbank: 8 GPUs, approximately ~350 GPU hours
> - WavLM + waveform: 8 GPUs, approximately ~600 GPU hours
>
> In conclustion, the better ASR fine-tuning performance and computation efficiency motivates us to select Zipformer + fbank as the backbone architecture for SPEAR. We will include this experiment in our revised manuscript.
>
> ---
>
> ### [W1.2] Teacher model selection
>
> We would like to direct the reviewer to the ablation experiments in Appendix G.1, where we used another two teacher models: HuBERT Large (317M, 60k hours) for speech and ATST-Frame (88M, 5k hours) for audio, as examples of teacher models with smaller amounts of pre-training data and smaller capacity. Below, we summarise the findings from the experiments:
>
> - **Stronger teacher model generally leads to a stronger student model**. As shown in Tables 16 and 17, SPEAR_s Large (with WavLM Large teacher) outperforms Large-H-2 (with HuBERT Large teacher) on SUPERB, and BASE-AUDIO-2 (with Dasheng teacher) outperforms BASE-AUDIO-1 (with ATST-frame teacher) on HEAR. This suggests that it is necessary to select a powerful teacher model for the optimal performance of SPEAR.
> - **Student models are not upperbouned by their teacher models**. Despite relying on the teacher model for generating pre-training targets, we discovered that SPEAR models can surpass its teacher without using extra data or bigger student model:
>     - Table 16: LARGE-H-1 outperforms its teacher HuBERT Large on all SUPERB tasks.
>     - Table 17: BASE-AUDIO-1 outperforms its teacher ATST-frame on all tasks except NS-5.
>
> *(The remaining comments will be addressed in subsequent responses).*

---

> ### Author Response · Authors · 2025-11-20
> **Response to Reviewer UWLL (2/2)**
>
> ## [W1] More ablation studies (Cont.)
>
> ### [W1.3] MVQ vs k-means
>
> In this ablation, we compared using MVQ-tokens and k-means token for pre-training. The same embedding layer (21st layer of WavLM Large) was used to extracted k-means and MVQ tokens, and we used 2000 clusters for k-means. We used the Base architecture (Zipformer + fbank) and pre-trained the models for 300k udpates on LS-960. The same input augmentations are adopted, including the noise mixing, utterance mixing and masking strategy. For the k-means experiment, we only computed the loss on the masked position following HuBERT[1]. The following results are given below: ASR word error rates with LS-100 ASR fine-tuning and the SUPERB benchmark.
>
> | Discrete Target | LS-100 $\downarrow$ | PR $\downarrow$| IC $\uparrow$ | KWS $\uparrow$ |  SID $\uparrow$ | ER $\uparrow$ |
> | --------------- | -----------| --- | ---| --- | --| --|
> | k-means, 2000 cluster | 3.5/7.2 | 4.0 | 97.92 |  96.79 | 86.6 | 67.56 |
> | Ours, MVQ | **3.1**/**7.0** | **3.4** | **98.37** | **96.83** | **88.4** |**68.29** |
>
> We observe that the model trained with MVQ tokens achieves better performance on all tasks, especially two paralinguistic tasks: SID and ER. This aligns with our findings in Appendix G.4, where it is shown that the fine-grained MVQ tokens retain richer paralinguistic information than the coarse k-means token. We thank the reviewer for proposing this experiment, which helps further verify the advantages of using fine-grained MVQ tokens for pre-training in SPEAR. **We will add this experiment in the revised manuscript.**
>
> ---
>
> ## [W2]: Usage of WavLM as teacher
>
> > One of the main points of the article is to propose a new SSL model that outperforms WavLM-Large (which has been the top model on the SUPERB benchmark for some time now). However, it could be clarified that this model is actually distilling WavLM-Large during its SSL training.
>
> Thank you for pointing out this. We agree that the high performance of SPEAR is achieved by leveraging the powerful WavLM model for extracting pre-training labels. We would like to direct the reviewer to Appendix G.1 in the original manuscript, where we compare different teacher models for SPEAR and found better teacher models lead to stronger student model in general. We will make sure that this is hightighted in our revised manuscript.
>
> ---
>
> ## [Q1]: Different encoder architectures:
>
> > Did you try to replace your model by a Wav2Vec2.0 architecture?
>
> Yes, we compared Zipformer with wav2vec 2.0 architecture. Please refer to the results presented in Response 1 (see [W1.2] Teacher model selection).
>
> ---
>
> ## [Q2]: USAD-style pre-training
>
> > Or did you tried to use the same L1 loss as in USAD, removing the MVQ, while keeping everything else similar?
>
> Thank you for the question. We did not try using the L1 loss within the SPEAR framework, mainly due to the following two reasons:
> - **Using USAD-style L1 loss constrains the scalability of SPEAR**. In the USAD paper, the authors use L1 and cosine similarity loss over multiple teacher-student layer pairs. This requires the teacher model to be evaluated in an online fasion (otherwise storing the frame-level intermediate layer's feature to disk is too expensive). However, online KD does not scale well to large teacher models in terms of computation and GPU memory, since the teacher model has to be evaluated in every forward pass, especially when we use large teacher models such as Dasheng 1.2B.
> - **KD loss based on distance metric like L1 loss enforces a strict imitation of the teacher representation, which could limit the performance of the student model.** Theoretically, a student trained with the L1 loss against the teacher model will try to approximate the representations of the teacher model, and its performance will be upperbounded by its teacher (given the same model size and data). In SPEAR, we predict the quantised labels of the teacher representations instead of matching the continuous representations in a masked token prediction manner. **This combination of KD and masked token prediction enables SPEAR to outperform its teacher model without using extra data or bigger model** (e.g SPEAR_s Large outperforms WavLM Large, Base-Audio-1 (see Table 17) outperforms ATST-Frame).

---

> ### Comment · Reviewer_UWLL · 2025-11-20
> **Thank you for your comments.**
>
> Thank you for your comment, the ablation studies you have shown, as well as the comparison on the SUPERB benchmark saw on another reviewer's answer really help building the soundness of the article.
>
> About the USAD-style pre-training:
> My understanding was that your proposal was an increment from USAD by the addition of MVQ representations and KD Loss, your explanation of why the KD loss is a necessary replacement of the L1 loss make sense, and I guess the ablation study on K-means clustering vs MVQ would be a way to justify the use of MVQ in comparison to the techniques used in USAD.
>
> Thanks again for your answers, I have no further inquiries.

---

> ### Author Response · Authors · 2025-11-28
> **Thank you for your feedback!**
>
> Thank you for your quick response and positive feedback, and for maintaining the score at 8. We’re glad to hear that our additional experiments and theoretical explanations have addressed all your concerns. We have already incorporated the additional results into our revised manuscript.

---

### Official Review · Reviewer_orCT · 2025-11-01

**Soundness:** 3
**Presentation:** 3
**Contribution:** 2
**Rating:** 4
**Confidence:** 4

**Summary:**

The paper proposes SPEAR, a self-supervised model to unify speech and audio representations. This is achieved by using a masked token prediction loss between the student model and domain-specific teacher models. The paper achieves strong performance across speech and audio benchmarks such as SUPERB and HEAR.

**Strengths:**

The paper is written well and contains detailed experiments. The experimental analysis is comprehensive. The student model achieves strong performance on many tasks, even outperforming the teacher model, such as WavLM, on some tasks.

**Weaknesses:**

The paper has limited novelty. Multi-codebook vector quantization was proposed for knowledge distillation, and distilling from modality-specific teachers to build a unified model has been done before, e.g., USAD.

**Questions:**

Line 15: “the first SSL framework to successfully learn unified speech and audio representations from a mixture of speech and audio data.” What makes it the first SSL framework when other methods, such as USAD (one of the main baselines of the paper), have proposed unified models to combine the audio and speech processing capabilities into a single model?

On the HEAR benchmark, Table 6, the Dasheng 0.6B model outperforms the SPEAR XLarge model in average performance across env, speech, and music tasks. For Env and music tasks, the Dasheng teacher-model is better, and on speech, SPEAR performs better. On the SUPERB benchmark, Table 5, the SPEAR-Large model matches the performance on the WavLM teacher.
Only after scaling the SPEAR model to 600M, the SPEAR model outperform the teacher WavLM model. For Audio tasks, even with scaling, the model does not seem to outperform the teacher Dasheng. Why does this discrepancy exist between the audio and speech domains?

In Table 5, why is the Dasheng model not included? It would be interesting to see how the representations extracted from the Dasheng perform on the SUPERB benchmark.

Given the strong performance of the Dasheng teacher model on the HEAR benchmark, do we really need to distill from domain-specific teacher models to build unified models, or is directly training a model with an SSL objective using both modalities sufficient?

---

> ### Author Response · Authors · 2025-11-19
> **Response to Reviewer orCT (1/4)**
>
> We thank the reviewer for the detailed review and valuable comments, which have helped improve the clarity and quality of our work. Please find below the responses to each comment.
>
> ---
>
> ## Additional Experiments
> Before we address the questions raised, we would like to first present the results of two additional experiments, which we believe could better support our arguments. Specifically, we performed more evaluation on the audio-SSL Dasheng model (as requested by the reviewer), which is used as the audio teacher models in SPEAR.
>
> ---
>
> ### [EXP1]: Dasheng Evaluation on SUPERB
>
> > In Table 5, why is the Dasheng model not included? It would be interesting to see how the representations extracted from the Dasheng perform on the SUPERB benchmark.
>
> Thank you for pointing out this missing result in Table 5. Following your suggestion, we have evaluated Dasheng on SUPERB and give the results in the following table.
>
> | Model | # Params | Pre-train Hours | PR $\downarrow$ | ASR $\downarrow$ | IC $\uparrow$ | KS $\uparrow$ | SID $\uparrow$ | ER $\uparrow$ |
> | ---- | --- | --- | --- | --- | --- | --- | --- | ---|
> | Dasheng-1.2B | 1.2B | 272k | 13.8  | 14.3 | 98.13 | 97.73 | 92.4 | 68.7 |
> | SPEAR_s Large | 327M | 84k | **2.6** | 3.3 | 99.47 | 97.89 | 95.5 | 72.1 |
> | SPEAR_{s+a} XLarge | 600M | 197k |  2.9 | **3.2** | **99.61** | **98.12** | **96.3** | **73.3** |
>
>
> It can be seen that Dasheng-1.2B performs significantly worse than SPEAR_{s+a} XLarge. **This indicates that the features learned by audio-SSL models like Dasheng do not generalise well to speech processing tasks**, especially tasks requiring semantic/phonetic (ASR, PR) and paralinguistic information (SID, ER).
>
> **We will add this result to Table 5 in the revised manuscript.**
>
> ---
> ### [EXP2]: Fine-tune Dasheng for ASR
>
> Additionally, we performed an experiment to fine-tune Dasheng model on LibriSpeech data for the ASR task, with the same fine-tuning configuration as described in Section 5.1. The WERs on test-clean and test-other are reported in the Table below with LS-100 and LS-960 as fine-tuning sets. We see that **Dasheng-1.2B model yields much higher WERs comared to SPEAR_{s+a} XLarge**, despite of its massive pre-training corpora (272k hours) and large model size (1.2B).
>
> | Model | # Params | Pre-train Hours | LS-100 $\downarrow$ | LS-960 $\downarrow$|
> | ------- | ------|----- | --- | ------ |
> | Dasheng-1.2B | 1.2B | 272k | 7.7/20.0 | 3.4/8.7 |
> | SPEAR_{s+a} XLarge | 600M |197k |  **2.5**/**4.6** | **1.6**/**2.9** |
>
> **We will add this result to Table 4 in the revised manuscript.**
>
> ---
>
> From the two experiments above, we can see that the large-scale audio-SSL Dasheng model (1.2B, 272k hours) shows weak generalisation capability to speech tasks (especially tasks requiring high-level semantic understanding), despite of giving good performance in the general audio domain. In contrast, the dual-domain SPEAR_{s+a} models demonstrate strong and balanced capabilities across both domains as shown in our paper.
>
> *(The remaining comments will be addressed in subsequent responses).*

---

> ### Author Response · Authors · 2025-11-19
> **Response to Reviewer orCT (2/4)**
>
> ### [W1]: Novelty of SPEAR
>
> > The paper has limited novelty. Multi-codebook vector quantization was proposed for knowledge distillation, and distilling from modality-specific teachers to build a unified model has been done before, e.g., USAD.
>
> Thank you for the comment. Indeed, SPEAR is built on existing technolgies, such as Multi-codebook Vector Quantisation (MVQ), and multi-modality knowledge distillation (KD).
> Despite of this, we would like to emphasise that **SPEAR is the first to propose a unified SSL objective to successfully learn unified speech and audio representations from a mixture of heterogeneous speech and general audio data.**
>
> Existing SSL approaches for speech or audio domain are highly domain specific. For instance, speech SOTA SSL models (e.g. WavLM) use masked prediction of coarse tokens, which are unsuitable for general audio (see Table 6 for WavLM results on HEAR). Audio SOTA SSL models (like Dasheng) use objectives like masked autoencoders (MAE), which has been shown to learn limited semantic information essential for speech tasks (see the supplemented experiment of Dasheng on SUPERB and ASR).
> **Our proposed SPEAR framework addresses this limitation by proposing a unified masked token prediction pretext task for data from both domains.**
>
> To summarise, SPEAR's novelty is as follows:
> - **A novel SSL objective for unified speech and audio representation learning**: We are the first to propose masked prediction of fine-grained tokens as a unified objective for both domains. This is enabled by our novel application of MVQ, a method originally proposed for KD, for generating fine-grained discrete pre-training labels for SSL. This specific design choice makes joint SSL pre-training on speech and audio feasible, which is validated by our strong experimental results on SUPERB and HEAR.
> - **Formulation of KD as an SSL objective**: While other unified models like USAD and MT2KD exist (as referenced in Section 2 related works), they are primarily KD frameworks that use feature-matching objectives (e.g., L1 loss and cosine similarity) to mimic teacher representations. SPEAR is fundamentally different by formulating the KD task as an SSL objective. This enables the student model to learn representations that can even outperform its teachers (e.g., SPEAR_s vs. WavLM in Table 5), which is difficult for traditional KD methods like USAD.
> - To cope with the imbalance of the available speech and general audio training data, we design an asymmetrical multi-domain training strategy (see Section 3.2.2), which proves to be essential for SPEAR to achieve a balanced performance across both domains.
>
> Hence, we argue that this unified SSL framework is a significant contribution, as it provides a single, versatile model that excels in both speech and audio understanding. Such a model could be used as a unified frontend for multi-modal speech LLMs by replacing the existing speech-audio dual-encoder architecture [1].
>
> ---
>
> ### [Q1]: Relationship with USAD
>
> > Line 15: “the first SSL framework to successfully learn unified speech and audio representations from a mixture of speech and audio data.” What makes it the first SSL framework when other methods, such as USAD (one of the main baselines of the paper), have proposed unified models to combine the audio and speech processing capabilities into a single model?
>
> Thank you for raising this important point. While both SPEAR and USAD aim to build a unified model with comprehensive speech and audio processing capability, we would like to highlight that the two approaches differ fundamentally in their pre-training objectives, which is precisely what defines SPEAR as an SSL framework rather than a standard knowledge distillation (KD) approach.
>
> - **USAD** is a multi-teacher KD framework that trains a student to directly match the continuous feature representations of its teachers. As we note in the related work (last paragraph Section 2), USAD uses L1 loss and cosine similarity to force student to mimic the teachers' embedding space explicitly.
> - **SPEAR**, in contrast, does not force the student to explicitly mimic the teacher embedding space. Instead, SPEAR leverages MVQ to quantise the teacher embeddings into fine-grained discrete tokens, and trains the student to predict them given a masked input. This masked token prediction task is a common pretext task in SSL. Since the teacher models are SSL models, SPEAR itself can be viewed as an SSL framework with a fine-grained masked token prediction pretext task.
>
> We believe this is why SPEAR learns more powerful and generic representations that it can even outperform its teacher model (e.g., SPEAR_s Large vs. WavLM Large), demonstrating that it is not limited in the same way as a direct feature-matching objective. **We will make sure to further clarify this difference in the revised manuscript.**
>
> [1]: Tang, Changli, et al. "Salmonn: Towards generic hearing abilities for large language models." ICLR 2024

---

> ### Author Response · Authors · 2025-11-19
> **Response to Reviewer orCT (3/4)**
>
> ### [Q2.1]: Performance comparison of SPEAR and WavLM Large on SUPERB
> > On the SUPERB benchmark, Table 5, the SPEAR-Large model matches the performance on the WavLM teacher. Only after scaling the SPEAR model to 600M, the SPEAR model outperform the teacher WavLM model.
>
> Thank you for raising this question. First, we would like to clarify that we provide two Large versions of SPEAR in Table 5, a speech-domain model and a dual-domain model.
> - The **speech-domain SPEAR_s Large** is compared to WavLM Large under fair conditions—similar model size (327M vs. 317M) and pre-training data (84k vs. 94k hours), and **SPEAR_s Large matches or outperforms WavLM Large on 12 out of 15 SUPERB tasks.**
> - The **dual-domain SPEAR_{s+a} Large** model is slightly less well than SPEAR_s Large on SUPERB, but it still matches its speech teacher WavLM Large, while also being able to yield high performance on the audio-centric HEAR benchmark. This extra capability makes SPEAR_{s+a} more versatile than the single-domain speech models like SPEAR_s or WavLM.
>
> To summarise, we observe:
> - SPEAR can produce strong speech-domain models, even outperforming its teacher WavLM without using bigger model or more data.
> - SPEAR can produce dual-domain models with the capability to match its speech-domain teacher WavLM Large, but possesses the extra capability to handle audio at the same time.
>
> Finally, by scaling to XLarge (600M, 197k hours), we are able to further improve the dual-domain model (as shown in Table 5 and 6).
>
> ---
>
> ### [Q2.2]: Performance comparison of SPEAR and Dasheng on HEAR
>
> > On the HEAR benchmark, Table 6, the Dasheng 0.6B model outperforms the SPEAR XLarge model in average performance across env, speech, and music tasks. For Env and music tasks, the Dasheng teacher-model is better, and on speech, SPEAR performs better ...
>
> We agree with your observation regarding performance comparison between Dasheng 0.6B and SPEAR XLarge. Please find below our explanation:
>
> - **The teacher Dasheng uses more general audio data for pre-training.** HEAR is an audio-centric benchmark, with 12 out of 18 tasks related to non-speech audio. Therefore, incorporating more general audio data (non-speech) is critical to the performance on HEAR. Our teacher model, Dasheng, is pre-trained on over 272k hours of general audio data (over 50% are non-speech audio) covering both speech and non-speech audio, whereas SPEAR_{s+a} XLarge only used 13k hours of general audio out of the Mix-197k dataset, constituting only 7% of the total pre-training data. It is therefore understandable that SPEAR models perform less well than Dasheng on the HEAR benchmark due to a far smaller amount of general audio data used during pre-training.
> - Despite this huge data limitation, the effectiveness of the SPEAR framework for audio is still evident: **Our audio-specialist SPEAR_a Large (327M, 13k hours, with Dasheng 1.2B as teacher) is able to outperform its teacher on the HEAR environment sub-category** (83.58 vs. 83.20). Note that this category is highly related to general audio.
>
> In addition, we anticipate an improvement of audio performance if more general-audio data is provided. To validate this trend, we conducted a scaling experiment for the SPEAR_a Base and Large models by increasing the pre-training data from 5k (AudioSet) to 13k hours (Audio 13k). **The average score on HEAR shows a positive scaling trend**:
>
> | Model | 5k hours |  13k hours |
> | ---- | --- | ---- |
> | Base | 76.4 | 77.0 |
> | Large | 78.1 | 79.2 |
>
> This trend motivates us to collect more general audio data to further enhance the audio performance of SPEAR, which we leave as important future work. We will add this result to Table 6 in the revised manuscript.
>
> > For Audio tasks, even with scaling, the model does not seem to outperform the teacher Dasheng. Why does this discrepancy exist between the audio and speech domains?
>
> We thank the reviewer for pointing out the discrepancy between speech and audio domains. For the speech experiments, we were able to replicate the data usage of WavLM, and the results show that SPEAR_s models outperform their WavLM counterparts consistently. However, we were not able to collect the same general audio dataset used by Dasheng, since the ACAV100M dataset (263k hours) used by Dasheng are difficult to acquire (need to be crawled from YouTube), making comparison between SPEAR and Dasheng not strictly fair since Dasheng utilises much more general audio data.
>
> Additionally, we would like to direct you to the ablation experiment in Appendix G.1 in the original manuscript, where we provided a controlled audio experiment with strictly fair comparison. We use another highly competitive audio SSL teacher ATST-frame (88M, 5k hours) and train a student (Base-Audio-1) on the same 5k hours. Base-Audio-1 outperforms its teacher ATST-frame on almost every task (see Table 17)**. We believe this experiment could help further clarify your concern on the audio performance of SPEAR.

---

> ### Author Response · Authors · 2025-11-19
> **Response to Reviewer orCT (4/4)**
>
> ### [Q3]: Dasheng result on SUPERB
>
> > In Table 5, why is the Dasheng model not included? It would be interesting to see how the representations extracted from the Dasheng perform on the SUPERB benchmark.
>
> Thank you for pointing out this missing result in Table 5. Please find the result and discussion of Dasheng evaluation on SUPERB in the additional experiments (see [EXP1] in Response 1). In addition, we also perform an ASR fine-tuning task to further test the speech capability of Dasheng (see [EXP2] in Response 1), where we find that Dasheng exhibits limited ASR capability.
>
> ---
>
> ### [Q4]: The necessity of using domain-specific teacher
> > Given the strong performance of the Dasheng teacher model on the HEAR benchmark, do we really need to distill from domain-specific teacher models to build unified models, or is directly training a model with an SSL objective using both modalities sufficient?
>
> Thank you for raising this interesting question. First of all, we agree that Dasheng is a very strong SSL model specialised in general audio given its very strong performance on HEAR benchmark. However, it should be noted that Dasheng exhibits limited capabilities on speech tasks, as shown by the weak performance on the speech-centric SUPERB benchmark and high WERs in the LibriSpeech fine-tuning task (see [EXP1] and [EXP2]). Note that over 40% of the pre-training data used by Dasheng are human sound related (statistics reported in ACAV100 [1]). This suggests that the MAE objective alone is not capable of learning unified speech and audio representations through a mixture of speech and audio data.
>
> Similarly, the masked token prediction objective commonly used in speech is also not suitable for learning high-quality audio representations, which is validated by the weak performance of WavLM on HEAR benchmark.
>
> In contrast, SPEAR provides a unified SSL framework for joint representation learning on mixture of speech and audio data by leveraging domain-specific teacher models. **The resulting SPEAR dual-domain models show competitive or even SOTA results on SUPERB and HEAR benchmark simultaneously, which validates the necessity of using domain-specific teachers in SPEAR.**
>
> [1]: Lee, Sangho, et al. "Acav100m: Automatic curation of large-scale datasets for audio-visual video representation learning." Proceedings of the IEEE/CVF International Conference on Computer Vision. 2021.

---

> ### Author Response · Authors · 2025-11-26
> **Looking forward to your feedback!**
>
> Dear Reviewer orCT,
>
> Thank you once again for your valuable feedback. We have conducted additional experiments and made revisions to the paper accordingly based on your suggestions. As the discussion phase is nearing its conclusion, we would like to know if our responses have addressed your concerns. We look forward to hearing from you.
>
> Best, Authors

---

### Author Response · Authors · 2025-11-25
**Summary of Revised Manusciprt (version 1, Nov 25th)**

We thank the reviewers for their time and the thoughtful, constructive feedback on our manuscript. We have carefully addressed all reviewers' comments and have revised the paper accordingly. The changes are summarised below:

### Content Revisions
- Enhanced the **motivation for developing a unified speech and audio encoder** in Section 1 (addressing reviewer y6wJ).
- Enhanced the **motivation for adopting masked token prediction** in Section 1 (addressing reviewer y6wJ).
- Revised the **description of BEATs regarding complexity** in Section 1 (addressing reviewer y6wJ)
- Added **missing reference of EncodecMAE** in Section 1&2, mentioning its application of predicting fine-grained discrete tokens for audio SSL (addressing reviewer y6wJ).
- Added **discussion on speech capabilities of audio-domain SSL models** in Section 5 (addressing reviewer orCT).
- Introduced a **new section, "Section 5.4 Ablation Studies", in the main text**. This section highlights key findings that were previously in the appendix or are newly added, ensuring they are more visible to the reader (addressing reviewer E2Pw).
- Added **more details on compuational resources** (addressing reviewer E2Pw).

### Additional Experimental Results
- Added **evaluation of Dasheng on speech tasks**: ASR fine-tuning in Table 5 and SUPERB evaluation in Table 5, which shows that single-domain audio-SSL models exhibit weak speech capabilities and reinforces the need for the unified approach proposed in SPEAR (addressing reviewer orCT).
- **Added two SPEAR_a models trained with fewer audio data** in Table 6 to show the positive scaling trend of SPEAR_a on audio SSL task, suggesting potential for further improvement with more data.
- Added **a controlled comparison between SPEAR and USAD** (same teacher, similar model size and training data) in Appendix H. Results confirming that SPEAR is more effective in learning unified speech and audio representations (addressing reviewer y6wJ).

### New Ablation Studies
- Added a **direct comparison between MVQ tokens and k-means tokens** as pre-training target, which confirms the advantage of MVQ tokens. (addressing reviewers UWLL, E2Pw).
- Added a **direct comparison between Zipformer+fbank and Transformer+waveform** as encoder backbone. Results justify the choice of Zipformer based on better ASR fine-tuning performance and lower computational cost (addressing reviewers UWLL, E2Pw).
- Added results on **varying $\lambda$ in dual-domain SPEAR training** (addressing reviewer E2Pw).
- Added results of **varying teacher layers for generating pre-training targets** (addressing reviewer E2Pw).

We hope these revisions address all concerns and further strengthen the paper.

Thank you again for your time and helpful feedback.

---

> ### Author Response · Authors · 2025-12-02
> **Summary of Revised Manusciprt (version 2, Dec 2nd)**
>
> We thank the reviewers again for their time and the thoughtful, constructive feedback on our manuscript. Following the first revision, we have incorporated additional content to the first revised manuscript to address further questions raised during the rebuttal. We believe these updates further strengthen the quality and clarity of our work. Below we summarise all changes we made compared to the original manuscript. **The newly added content (compared to the original manuscript) is marked in blue**.
>
>
> ### Content Revisions Relative to Original Manuscript
> - Enhanced the **motivation for developing a unified speech and audio encoder** in Section 1 (addressing reviewer y6wJ).
> - Enhanced the **motivation for adopting masked token prediction** in Section 1 (addressing reviewer y6wJ).
> - Revised the **description of BEATs regarding complexity** in Section 1 (addressing reviewer y6wJ)
> - Added **missing reference of EncodecMAE** in Section 1&2, mentioning its application of predicting fine-grained discrete tokens for audio SSL (addressing reviewer y6wJ).
> - Added **discussion on speech capabilities of audio-domain SSL models** in Section 5 (addressing reviewer orCT).
> - Introduced a **new section, "Section 5.4 Ablation Studies", in the main text**. This section highlights key findings that were previously in the appendix or are newly added, ensuring they are more visible to the reader (addressing reviewer E2Pw).
> - Added **more details on compuational resources** (addressing reviewer E2Pw).
> - Added **discussion on the limited performance of Dasheng** on SUPERB in Section 5.2 (addressing reviewer y6wJ)
>
> ### Additional Experimental Results Relative to Original Manuscript
> - Added **evaluation of Dasheng on speech tasks**: ASR fine-tuning in Table 5 and SUPERB evaluation in Table 5, which shows that single-domain audio-SSL models exhibit weak speech capabilities and reinforces the need for the unified approach proposed in SPEAR (addressing reviewer orCT)
> - **Added two SPEAR_a models trained with fewer audio data** in Table 6 to show the positive scaling trend of SPEAR_a on audio SSL task, suggesting potential for further improvement with more data.
> - Added **a controlled comparison between SPEAR and USAD** (same teacher, similar model size and training data) in Appendix H. Results confirming that SPEAR is more effective in learning unified speech and audio representations.
> - Added **a controlled comparison between SPEAR and Dasheng** (same 5k hours training data) in Appendix I. Results indicate that SPEAR outperforms Dasheng using the same training set (addressing reviewer y6wJ).
>
> ### New Ablation Studies Relative to Original Manuscript
> - Added a **direct comparison between MVQ tokens and k-means tokens** as pre-training target, which confirms the advantage of MVQ tokens. (addressing reviewers UWLL, E2Pw).
> - Added a **direct comparison between Zipformer+fbank and Transformer+waveform** as encoder backbone. Results justify the choice of Zipformer based on better ASR fine-tuning performance and lower computational cost (addressing reviewers UWLL, E2Pw).
> - Added results on **varying $\lambda$ in dual-domain SPEAR training** (addressing reviewer E2Pw).
> - Added results of **varying teacher layers for generating pre-training targets** (addressing reviewer E2Pw).

---

### Author Response · Authors · 2025-12-02
**Summary of Review and Rebuttal (Part 1/3)**

We thank all reviewers, the AC, and the SAC for their thoughtful feedback. In light of the recent decision to rollback review scores, below we summarise the overall status of the rebuttal process. We summarise the strengths and weaknesses pointed out by the reviewers, and elaborate on how we addressed all the weaknesses/questions raised by the reviewers.

---

## Overall Status before Rollback

SPEAR is the **first self-supervised learning (SSL) framework** to learn **unified speech and audio representations** from a heterogeneous mixture of speech and general audio data. To bridge the gap between speech and general audio domains, SPEAR proposes a unified masked-token prediction task leveraging **fine-grained MVQ tokens** extracted from domain-specific teachers, achieving either state-of-the-art or competitive performance simultaneously on **both** the SUPERB and HEAR benchmarks **with a single model**.

The reviewers acknowledged the novelty (UWLL, E2Pw), the clear presentation and writing (orCT, UWLL, E2Pw, y6wJ), the comprehensive evaluation (orCT, UWLL, E2Pw) and the strong performance (orCT, UWLL, E2Pw), while raising concerns about novelty/difference to a contemporaneous work (orCT, y6wJ), and audio performance compared to Dasheng (orCT, y6wJ).

After our rebuttal and the inclusion of extensive additional experiments, three of our four reviewers (UWLL, E2Pw, y6wJ) explicitly acknowledged that their concerns had been mostly resolved, while reviewer orCT (originally 4) did not respond. Most notably, Reviewer y6wJ (originally 2) explicitly confirmed (on Nov 26) that the rebuttal "**considerably improved the paper**" and **raised their overall score to 6** towards acceptance, while reviewers UWLL and E2Pw maintained their positive ratings of 8 and 6. Consequently, the consensus reached prior to the rollback was 4 (orCT), 8 (UWLL), 6 (E2Pw), 6 (y6wJ), **yielding an average score of 6 before the decision to rollback the scores**.

---

## Summary of weaknesses/questions raised and how we addressed them

For simplicity, **we group common concerns and questions, and provide a summary of responses for each of them**. The detailed full response (including additional experimental results) can be found in the discussion with individual reviewers.

### 1. Novelty and difference of SPEAR compared to USAD (orCT, y6wJ)

The reviewers questioned the novelty of SPEAR, mainly against USAD, a contemporaneous work for building a unified model for speech and audio. Reviewer y6wJ also pointed out a missing reference.

> orCT: Distilling from modality-specific teachers to build a unified model has been done before, e.g., USAD

> y6wJ: There is no mention of EncodecMAE in the submitted paper... Knowledge distillation in this realm has been done before in USAD


**Our response**:
- **Contemporaneous Nature of USAD**: We noted that USAD is contemporaneous work according to ICLR policy (it had not appeared in a peer-reviewed venue at the time of submission, it will appear at ASRU in December 2025). Nevertheless, we provided a detailed comparison in our original manuscript.
- **Fundamental Difference to USAD (feature matching vs masked-token prediction)**: We clarified that USAD relies on the feature-matching loss (L1/cosine), which forces the student to strictly mimic the teachers. In contrast, SPEAR formulates the problem as an SSL pretext task via masked prediction of fine-grained MVQ tokens extracted from teachers. This design enables SPEAR to learn both high-level semantic structures and retain low-level acoustic details simultaneously, even outperforming its teachers, which is inherently difficult for feature-matching methods.
- **Controlled Comparison**: To fully respond to the reviewers on the advantages of SPEAR, we added a controlled experiment (Appendix H) to compare SPEAR and USAD with identical teachers and data constraints. **Results showed that SPEAR consistently outperformed USAD on all tasks**. We also provided theoretical explanations as to why SPEAR outperforms USAD.
- We added the reference on EncodecMAE and clarified that it is an audio-only model with limited speech capabilities, whereas SPEAR excels both domains.

**Reviewer response**:
Reviewer y6wJ on Nov 26: "***SPEAR works better than USAD, which is crystal clear**...This commentary is necessary and actually makes your paper stronger*"

---

> ### Author Response · Authors · 2025-12-02
> **Summary of Review and Rebuttal (Part 2/3)**
>
> ### 2. Request for more ablation studies (UWLL, E2Pw)
>
> The reviewers suggested additional ablations to better isolate the impact of specific components in SPEAR.
>
> > One or multiple ablation studies would have been interesting, to better judge which aspects of the pipeline have the greatest impact.
>
> **Our response**:
> We highlighted the existing ablations in Appendix G, including teacher model selection, quantisation configuration and dual-domain training strategy. Following the reviewers' request, we added new ablation studies on the following components in the revised manuscript:
> - Pre-training target (Appendix G.5): MVQ tokens (adopted) vs. k-means tokens. Results confirm MVQ tokens consistently outperform coarse k-means tokens.
> - Encoder architecture (Appendix G.6): Zipformer + fbank (adopted) vs. Transformer + waveform. Results justify the choice of Zipformer due to better downstream ASR performance and higher training efficiency.
> - Teacher layer selection (Appendix G.1.2)
> - Loss weighting (Appendix G.7.2)
>
> **Reviewers response**:
> Reviewer E2Pw on Nov 20 regarding the new ablations: "*including them will strengthen the paper's quality and technical soundness. **I will update the soundness score accordingly**.*"
> Reviewer UWLL on Nov 20: "*the ablation studies you have shown, ... **really help building the soundness of the article***"
>
> ---
>
> ### 3. Performance comparison between SPEAR and Dasheng (orCT, y6wJ)
>
> The reviewers asked why Dasheng outperforms SPEAR models on the audio-centric HEAR benchmark and questioned the benefit of the SPEAR framework.
>
> > orCT: For Audio tasks, even with scaling, the model does not seem to outperform the teacher Dasheng
>
> > y6wJ: the Dasheng teacher still outperforms the proposed SPEAR-{a} and SPEAR-{a+s} models
>
> **Our response**:
> - **Data Disparity**: We clarified the massive difference in general audio training data (Dasheng uses ~10x more general audio data than SPEAR).
> - **Data Scaling**: We demonstrated a positive scaling trend: increasing audio data from 5k to 13k hours yielded notable improvements on HEAR benchmark for SPEAR.
> - **Controlled Comparison**: We provided a head-to-head comparison between SPEAR and Dasheng in Appendix I using the same 5k-hour training set (AudioSet) and showed that **SPEAR outperformed Dasheng on HEAR (78.08 vs. 74.87) despite having only a quarter of the parameters.**
>
> **Reviewers response**:
> Reviewer y6wJ acknowledged this on Nov 26: "Thanks for adding these comparisons... **The provided additional comparison further strengthens the paper**."
>
> ---
>
> ### 4. Lack of motivation for unified speech&audio model and for using masked-token prediction as objective (y6wJ)
>
> > Why do we need a unified front against speech and audio, and is masked discretised token prediction actually the correct way to go about it?
>
> **Our response**:
> - **Enhanced Motivation**: We expanded Section 1 (first paragraph) to articulate the practical necessity of unified models for real-world mixed-audio scenarios.
> - **Theoretical Justification**: We provided empirical evidence and theoretical analysis by contrasting our masked-token prediction objective in SPEAR with Dasheng's MAE objective and USAD's feature matching objective, explaining why joint-domain masked-token prediction effectively captures both high-level semantics and fine-grained acoustic details.
>
> **Reviewers response**:
> Reviewer y6wJ on Nov 26: "**Something quite constructive emerged from the author's response** ... This commentary is necessary and actually makes your paper stronger."
>
> ---
>
> ### 5. Table 6 hard to compare (y6wJ)
>
> > models presented in Table 6 are trained on different amounts of data. This only obfuscates results.
>
> **Our response**:
> We included SPEAR models trained on 5k hours for fair comparison with BEATs in the first revised manuscript. Additionally, we have now provided another comparison with Dasheng (Appendix I) using the same training data in the latest version of the manuscript.
>
> ---
>
> ###  6. Missing result of Dasheng on SUPERB (orCT)
>
> > In Table 5, why is the Dasheng model not included? It would be interesting to see how the representations extracted from the Dasheng perform on the SUPERB benchmark.
>
> **Our response**:
> We gave the results of Dasheng on SUPERB and on the ASR fine-tuning task in the discussion and added these results to the first revised manuscript. The results show that Dasheng significantly underperforms SPEAR on these tasks, confirming that large-scale audio SSL models (e.g. Dasheng) lack the capability of semantic understanding necessary for speech.
>
> **Reviewers response**:
> Although Reviewer orCT did not respond, Reviewer UWLL responded (Nov 20) concerning this experiment that: "the comparison on the SUPERB benchmark saw on another reviewer's answer really help building the soundness of the article"

---

> ### Author Response · Authors · 2025-12-02
> **Summary of Review and Rebuttal (Part 3/3)**
>
> ### 7. The necessity of distillation from domain-specific teachers in SPEAR (orCT)
>
> > Given the strong performance of the Dasheng teacher model on the HEAR benchmark, do we really need to distill from domain-specific teacher models to build unified models, or is directly training a model with an SSL objective using both modalities sufficient?
>
> **Our response**:
> We justified our joint distillation approach in SPEAR by highlighting Dasheng's failure to generalise to speech (as shown in the new SUPERB/ASR results), despite it being trained on a mixture containing >100k hours of speech. This proves that single-domain objectives (e.g. MAE) are insufficient for unified understanding of speech and audio, validating SPEAR's use of domain-specific teachers to ensure excellence in both domains.
>
> ---
>
> ### 8. Main paper balance (E2Pw)
>
> > Many key analyses ... are relegated to the appendix, making it difficult for readers to assess mechanisms from the main narrative
>
> **Our responses**:
> We added a new "Section 5.4: Ablation Studies" to the main body of the revised manuscript. This section explicitly summarises key findings (Teacher selection, MVQ vs. k-means, Dual-domain strategy) that were previously in the appendix
>
> **Reviewer response**:
> Although the reviewer noted (Nov 20) that they "cannot assume the revised version will fully resolve my concerns", we have since implemented the specific structural changes requested in the first manuscript revision, which we believe fully addresses this concern.
>
> ---
>
> ### 9. Missing details on computation resources (E2Pw)
>
> > The paper lists steps and batch sizes but provides no hardware or GPU hours accounting for pre-training and fine-tuning
>
> **Our responses**
> We provided the requested details on computation resources for pre-training in Table 8 in Appendix C.2 (included in the first manuscript revision).
>
> ---
>
> ## Rebuttal Status at Discussion Freeze
>
> - Reviewer UWLL (Score: 8): Maintained score 8, explicitly stating that the new ablation studies further "help building the soundness of the article" and confirming no further inquiries.
> - Reviewer orCT (Score: 4): Did not respond to the rebuttal before the rebuttal process was unexpectedly stopped.
> - Reviewer y6wJ (Score: **2 $\rightarrow$ 6**): Most notably, this reviewer explicitly confirmed the intention to raise their score to 6 (26 Nov), stating "I believe it merits acceptance" and that the rebuttal "considerably improved the paper". Crucially, Reviewer y6wJ also explicitly struck through the weaknesses listed in the original review **to signal that these concerns had been fully resolved**.
> - Reviewer E2Pw (Score: 6): Maintained a positive rating of 6 but withheld a higher score because at that time (i.e. before the first manuscript revision was uploaded) they could not "assume the revised version will fully resolve" the concern regarding main paper balance and novelty. We have since explicitly implemented the structural changes and clarified the novelty of SPEAR in first manuscript revision but the reviewer did not further respond before the rebuttal process was unexpectedly stopped.
>
> We are confident that our extensive rebuttal and the latest revised manuscript have resolved the reviewers' concerns. Given the explicit endorsement from Reviewer y6wJ and the strong positive feedback from Reviewer UWLL, we believe that had the discussion period continued, Reviewer E2Pw (upon seeing the implemented structural revisions and clarification on novelty) and Reviewer orCT (upon reviewing the new experiments) would have acknowledged these improvements, likely leading to score increases for both and hence an expected final average score above 6.

---

### Meta-Review · Area_Chair_jY8r · 2026-01-03

**Summary:**

The reviewer's concerns that informed my suggested decision for the paper can be summarized as follows: (i) limited novelty, (ii) lack of comparison against a unified model trained directly with a joint SSL objective on mixed data, and (iii) not convincing results on the HEAR benchmark.

**Reviewer Concerns:**

The authors have addressed some of the reviewers’ concerns in their rebuttal. For example, (i) they show that Dasheng-1.2B performs significantly worse than the proposed SPEAR_{s+a} XLarge, which implies that features learned by models such as Dasheng do not generalize well to speech processing tasks; (ii) they provide ablation studies that clarify the relative importance of combining speech and audio teachers, using MVQ representations, and adopting a non–Wav2Vec2.0 architecture; (iii) they explain why a single speech–audio foundational model is needed; and (iv) they address concerns about the complexity of the proposed model.

The following concerns remain outstanding: (i) The paper offers limited novelty, largely reiterating existing ideas. (ii) SPEAR is not the first unified speech and audio representation model. The proposed SPEAR model and USAD differ more in implementation details than in their overall objective. (iii) A direct performance comparison between SPEAR and WavLM-Large on the SUPERB benchmark is still missing, and the authors’ response is only partially convincing. (iv) On the HEAR benchmark, the Dasheng 0.6B model outperforms the SPEAR XLarge model in average performance across environmental, speech, and music tasks. This may indicate that the proposed solution is not yet fully mature and requires further development, particularly given that nothing prevents the authors from using more data. (vi) The necessity of domain-specific teachers (versus a unified model trained on mixed speech and audio data) remains unclear; notably, no ablation is provided in which a unified model is trained directly with a joint SSL objective on mixed data.

**Reviewer Scores:**

orCT would not have changed the score because the authors failed to address the reviewer’s concerns. The reviewer’s assessment is robust and can be considered the most reliable, as it does not appear to be influenced by any potential author-identification bias.

UWLL would not have changed their score, given its very high original value (namely, 8). The questions raised by the reviewer are not challenging and are very obliging in nature.

E2Pw increased the soundness score but kept the overall recommendation score unchanged (namely, 4), due to concerns regarding novelty.

y6wJ would have changed their original score, with the recommendation increasing from 2 to 6, which is a rather surprising jump given that the reviewer reported a confidence level of 5 in their assessment and originally considered the contribution to be poor and lacking in  novelty. This raises concerns about the technical reliability of this reviewer’s evaluation.

---

### Decision · Program_Chairs · 2026-01-26

Reject